# Spatial proteomics of ovarian cancer precursors delineates early disease changes and drug targets

Anuar Makhmut[1,2,8], Mihnea P Dragomir[3,4,5,8], Sonja Fritzsche [ID] [1,6], Markus Moebs [ID] [3], Wolfgang D Schmitt [ID] [3], Eliane T Taube [ID] [3,7] & Fabian Coscia [ID] [1✉]

## Abstract

**High-grade serous ovarian cancer (HGSOC) is often detected at an advanced stage, where curative treatment options are limited. Recent advances in ultrasensitive mass spectrometry-based spatial proteomics have provided a unique opportunity to uncover molecular drivers of early tumorigenesis and novel therapeutic targets. Here, we present a comprehensive proteomic analysis of serous tubal intraepithelial carcinoma (STIC), the HGSOC precursor lesion, and concurrent invasive carcinoma, covering more than 10,000 proteins from ultra-low input archival tissue. STIC and HGSOC showed highly similar proteomes, clustering into two subtypes with distinct tumor-immune microenvironments and common remodeling of the extracellular matrix. We discovered cell-of-origin signatures from secretory fallopian tube epithelial cells in STICs and identified early dysregulated pathways of therapeutic relevance. Targeting cholesterol biosynthesis by inhibiting the terminal steps via DHCR7 showed therapeutic effects in ovarian cancer cell lines and synergized with standard-of-care carboplatin treatment. This study demonstrates the power of spatially resolved quantitative proteomics in understanding early carcinogenesis and provides a rich resource for biomarker and drug target research.**

**Keywords** Spatial Tissue Proteomics; High-Grade Serous Ovarian Cancer; Serous Tubal Intraepithelial Carcinoma; Cancer Proteomics
**Subject Categories** Cancer; Proteomics

## Introduction

Among ovarian cancer subtypes, high-grade serous ovarian carcinoma (HGSOC) stands out as the subtype that causes the highest number of fatalities related to this malignancy (Bowtell et al, 2015). It is often diagnosed at an advanced stage and is characterized by frequent peritoneal metastasis and tumor recurrence following platinum-based chemotherapy. Genomic analyses have identified key genetic features of HGSOC, including pronounced genomic instability and vast copy number alterations, few recurrent mutations other than ubiquitous *TP53* mutations (Bell et al, 2011), homologous recombination (HRD) deficiency in roughly half of the tumors, often caused by *BRCA1/2* germline and somatic mutations, and high inter- and intra-tumoral heterogeneity (Veneziani et al, 2023). Moreover, several studies have highlighted the polyclonal nature of HGSOC and its pronounced spatial and temporal heterogeneity (Geistlinger et al, 2021; Cunnea et al, 2023; Denisenko et al, 2024). Mass spectrometry-based proteomics studies have addressed how HGSOC genomic alterations translate to the protein level, identifying dysregulated pathways associated with patient survival (Zhang et al, 2016) and chemotherapy response (Chowdhury et al, 2023; Coscia et al, 2018). These studies also revealed that the four bulk transcriptomic subtypes (mesenchymal, proliferative, immunoreactive, and differentiated) are reflected in the proteome (Zhang et al, 2016).

Despite these major efforts to map the proteogenomic disease landscape, far less understood are HGSOC precursor lesions, which are histologically present in 20–60% of all HGSOC patients (Kindelberger et al, 2007; Crum et al, 2007). Termed serous tubal intraepithelial carcinomas (STICs), these lesions share histological, molecular and genetic features with advanced HGSOC. Although it was first somewhat surprising to find the precursor for ovarian carcinoma in the tubal epithelium, the concept of STICs is widely acknowledged (Auersperg et al, 2008; Folkins et al, 2008; Labidi-Galy et al, 2017). STICs were previously profiled by whole-exome sequencing, which showed that they already present high genome instability and pronounced copy number alterations (Eckert et al, 2016; Labidi-Galy et al, 2017). However, very little is known about their proteomic makeup. In particular, we lack a detailed understanding of the extent to which precursor lesions molecularly diverge from normal fallopian tube epithelial cells to ultimately form morphologically distinct invasive tumors. As protein abundance is directly related to cellular phenotype and function (Aebersold and Mann, 2016), unraveling precancerous proteome states can provide important insights into the earliest stages of

[1]Max-Delbrück-Center for Molecular Medicine in the Helmholtz Association (MDC), Spatial Proteomics Group, Berlin, Germany. [2]Charité – Universitätsmedizin Berlin, corporate member of Freie Universität Berlin and Humboldt- Universität zu Berlin, Berlin, Germany. [3]Charité – Universitätsmedizin Berlin, corporate member of Freie Universität Berlin and Humboldt-Universität zu Berlin, Institute of Pathology, Berlin, Germany. [4]Berlin Institute of Health at Charité – Universitätsmedizin Berlin, Berlin, Germany. [5]German Cancer Consortium (DKTK), Partner Site Berlin, German Cancer Research Center (DKFZ), Heidelberg, Germany. [6]Humboldt-Universität zu Berlin, Institute of Biology, Berlin, Germany. [7]Present address: Institute of Pathology, Diagnostik Ernst von Bermann, Potsdam, Germany. [8]These authors contributed equally: Anuar Makhmut, Mihnea P Dragomir. ✉E-mail: fabian.coscia@mdc-berlin.de

HGSOC development and progression. Such knowledge is of paramount importance for preventive medicine approaches and for identifying protein-based drug targets that are likely to be present in the majority of tumor clones. In addition, we lack a deeper understanding of the co-evolving tumor microenvironment (TME), comprising the extracellular matrix (ECM) and diverse stromal and immune cell types, which are critically involved in all phases of HGSOC development (Schoutrop et al, 2022). Several lines of evidence support the crucial role of the HGSOC TME in regulating disease progression (Nieman et al, 2011; Cheon et al, 2014) and immunosuppression (Ghisoni et al, 2024), which dictate therapeutic outcomes (Jordan et al, 2020). Our own work recently identified a stromal signature of ovarian cancer metastasis and identified nicotinamide *N*-methyltransferase (NNMT) as a promising new drug target against cancer-associated fibroblasts (Eckert et al, 2019).

Here, we performed deep spatial proteomics of laser-microdissected fallopian tube epithelial cells, STICs, and concurrent invasive carcinomas, as well as their adjacent stromal regions. We present the first comprehensive proteomic map of ovarian cancer precursor lesions, encompassing more than 10,000 proteins in 36 patients. Our data revealed strong cell-of-origin proteome signatures preserved in STICs and concurrent invasive tumors, nominated onco-metabolic adaptations as early events during HGSOC development, and dissected the progressive co-evolution of the HGSOC tumor microenvironment.

# Results

## Histopathology-guided ultra-low input proteomics of HGSOC precursor lesions

To study the cell-type- and compartment-resolved proteomic progression of HGSOC precursor lesions (Fig. 1A), we selected a cohort of 36 patients with histologically confirmed serous tubal intraepithelial carcinoma (STIC). Precursor lesions were classified according to the criteria proposed by Vang et al (Vang et al, 2012) using three tissue sections stained with H&E and immunohistochemically for p53 and Ki67 (Fig. 1B). Only samples for which full agreement was reached were included in our study, and those that received chemotherapy before resection were excluded. Most patients were diagnosed at an advanced stage (>70% stage T3c, Fig. 1C), characteristic of sporadic HGSOC (Bell et al, 2011), and had a mean age at diagnosis of 63 years (Table EV1). Importantly, all but one patient had associated invasive carcinoma (IC), which in most cases was sampled in the adnexal region on the same histological slide (Fig. EV1), allowing us to directly compare STICs with concurrent ICs. We used additional serial tissue sections to analyze homologous repair deficiency (HRD) in ICs using targeted next-generation sequencing. The HRD status was successfully determined in 29 samples, which classified 15 tumors (approx. half of our cohort) as HRD-positive and 14 tumors as HRD-negative, in excellent agreement with a previous large-scale study conducted by TCGA (Bell et al, 2011). Of the HRD-positive samples, three carried a *BRCA1* mutation and one had a *BRCA2* mutation, classifying them as HRD-positive based on both the genomic instability score (GIS) and the *BRCA1/2* mutations. For three other samples, the GIS could not be determined because of the low tumor purity. Pathogenic *TP53* mutations, a ubiquitous genetic feature of

HGSOC (Bell et al, 2011), were identified in all 32 sequenced samples, with missense and nonsense mutations being the most prevalent (Fig. 1C; Dataset EV1). Having identified a representative cohort of HGSOC with co-occurring STIC precursor lesions and invasive carcinoma, we next investigated their proteomic makeup.

## Spatially resolved proteomes reflect disease-specific alterations at the bulk level depth

We employed our recently developed ultra-low-input tissue proteomics workflow (Makhmut et al, 2023), optimized for the seamless integration of immunohistochemistry (IHC) and immunofluorescence (IF) staining and ultrasensitive LC-MS-based proteomics. This approach enables the deep profiling of FFPE tissue microregions of only 50–100 cells in size, which is characteristic of STICs. For laser microdissection (LMD), tissues were mounted on PPS metal frame slides and stained with antibodies targeting p53, Ki67, and PAX8 to guide LMD and proteome profiling. For 35 cases, we successfully sampled at least one STIC lesion and one adjacent normal fallopian tube epithelial (NFTE) region. Additionally, in three patients, a second STIC could be sampled from the contralateral fallopian tube. For most cases, a complete set of samples was obtained, including epithelial and stromal regions (NFTE and NFT-St), STICs and adjacent stroma (STIC-St), and invasive carcinoma (IC) and its connected stroma (IC-St) (Figs. 1B,C and EV1). IC regions were defined as small tumor 'nests' clearly separated from the surrounding stroma (Fig. 1B). All samples were processed in a single batch in 384-well low-binding plates using our loss-reduced sample processing protocol (Methods) and measured in high-sensitivity dia-PASEF mode (Meier et al, 2020) on a timsTOF Ultra mass spectrometer. Raw files were analyzed in DIA-NN (Demichev et al, 2020) using a predicted spectral library, resulting in 10,223 unique protein groups from 192 mass spectrometric injections. The median proteome coverage for epithelial samples exceeded 7500 protein groups and 6000 for stromal samples, spanning more than five and four orders of magnitude, respectively (Figs. 2A–C and EV2A,B). The fewer number of identified proteins in stromal samples can be explained by differences in protein abundance. For example, the top 50 STIC stroma proteins contributed to 37.16% of the total protein mass, compared to 22.96% in STIC epithelial samples (Fig. 1B and Dataset EV2, EV3). Epithelial and stromal proteomes exhibited high compartment specificity. For example, known HGSOC markers, such as p53, PAX8, and MUC16 (CA125), were specific to the epithelial samples, whereas stromal and immune markers (i.e., collagens, vimentin, CD3E, and CD20) were characteristic of the stromal sample groups (Figs. 2B,C and EV2A,B). Notably, our high proteome coverage from FFPE tissue regions of only 50–100 cells in size was on par with recent bulk-level studies that required hundreds of micrograms to milligrams of fresh frozen tissue (Zhang et al, 2016; Qian et al, 2024). To investigate this further, we compared our dataset with those two large-scale bulk proteomic studies focusing on advanced HGSOC. Comparing our dataset with that of Qian et al and Zhang et al (CPTAC), we found that 73% of all identified proteins were common in all three studies. Interestingly, our study featured the highest number of uniquely identified protein groups (1022) (Fig. 2D), which we attributed to our laser microdissection-based sampling strategy that results in a "biological fractionation" effect for improved proteome coverage. Proteins

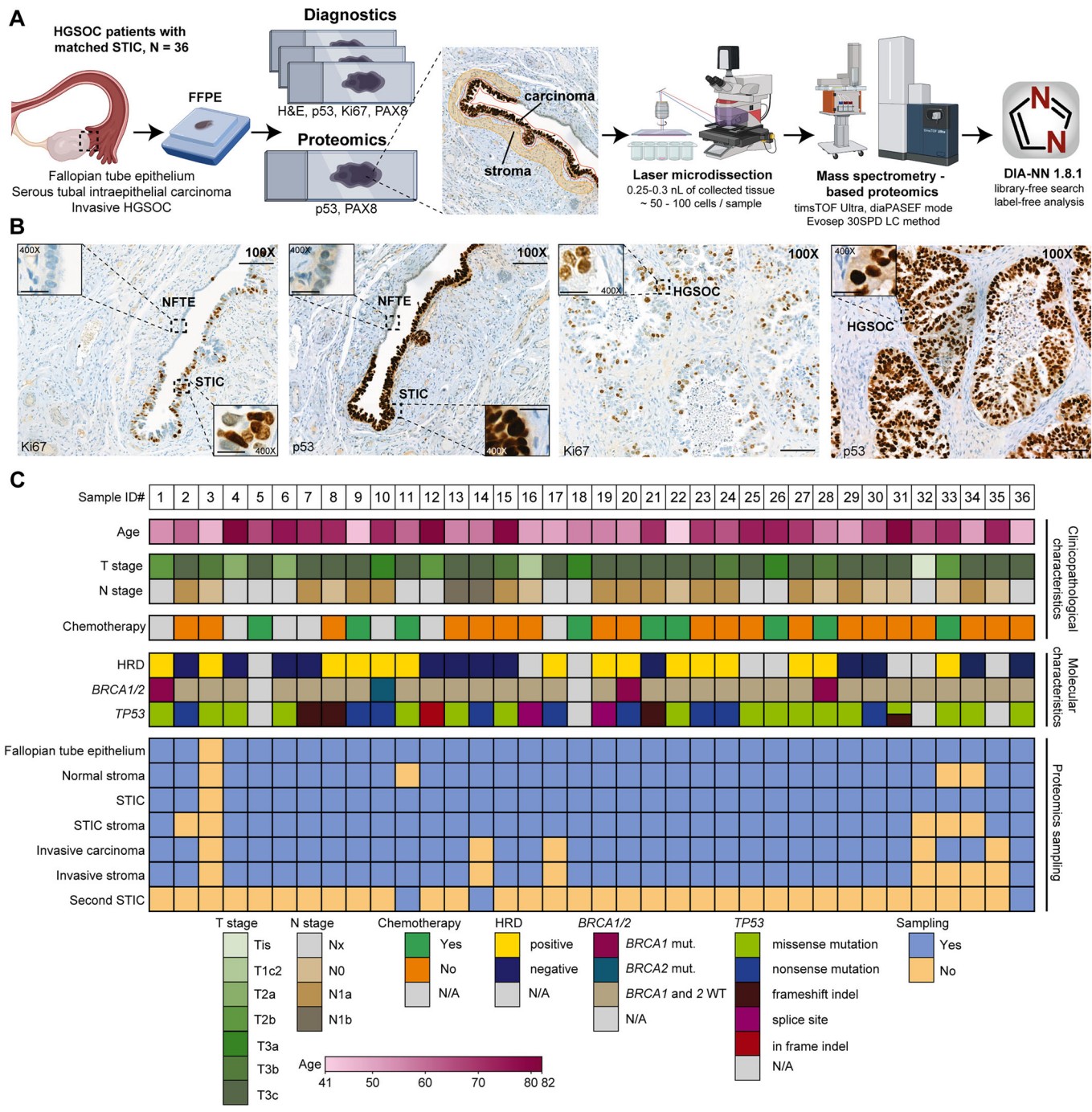

**Figure 1. Histology-guided ultra-low input proteomics of HGSOC precursor lesions.**

(A) Pathology-guided ultra-low input proteomics workflow for studying ovarian cancer precursor lesions at the global proteome level. Guided by immunohistochemistry (IHC), healthy and diseased epithelial and stromal regions were laser-microdissected and analyzed using ultrasensitive LC-MS-based proteomics. (B) Representative immunohistochemical staining (Ki67 and p53) of normal fallopian tube epithelium (NFTE), serous tubal intraepithelial carcinoma (STIC), and invasive carcinoma (IC) in the epithelial and stromal compartments. Scale bars: 100X: 100 μm, 400X: 20 μm. (C) Patient cohort (*n* = 36) with clinicopathological and molecular characteristics. Panel (A) was created with BioRender.com.

unique to our dataset were enriched for tumorigenesis-associated processes, such as immunity (interleukin-36 pathway), transcription (RNA polymerase I), and chromatin-related functions (PRC2 chromatin regulator complex, Fig. 2E). The biological richness of our proteomics data was also reflected in the hallmark gene sets, which showed a median pathway coverage of 73% (Fig. 2F). For example, pathways such as fatty acid metabolism (86%), DNA repair (88%) and oxidative phosphorylation (91%) were almost

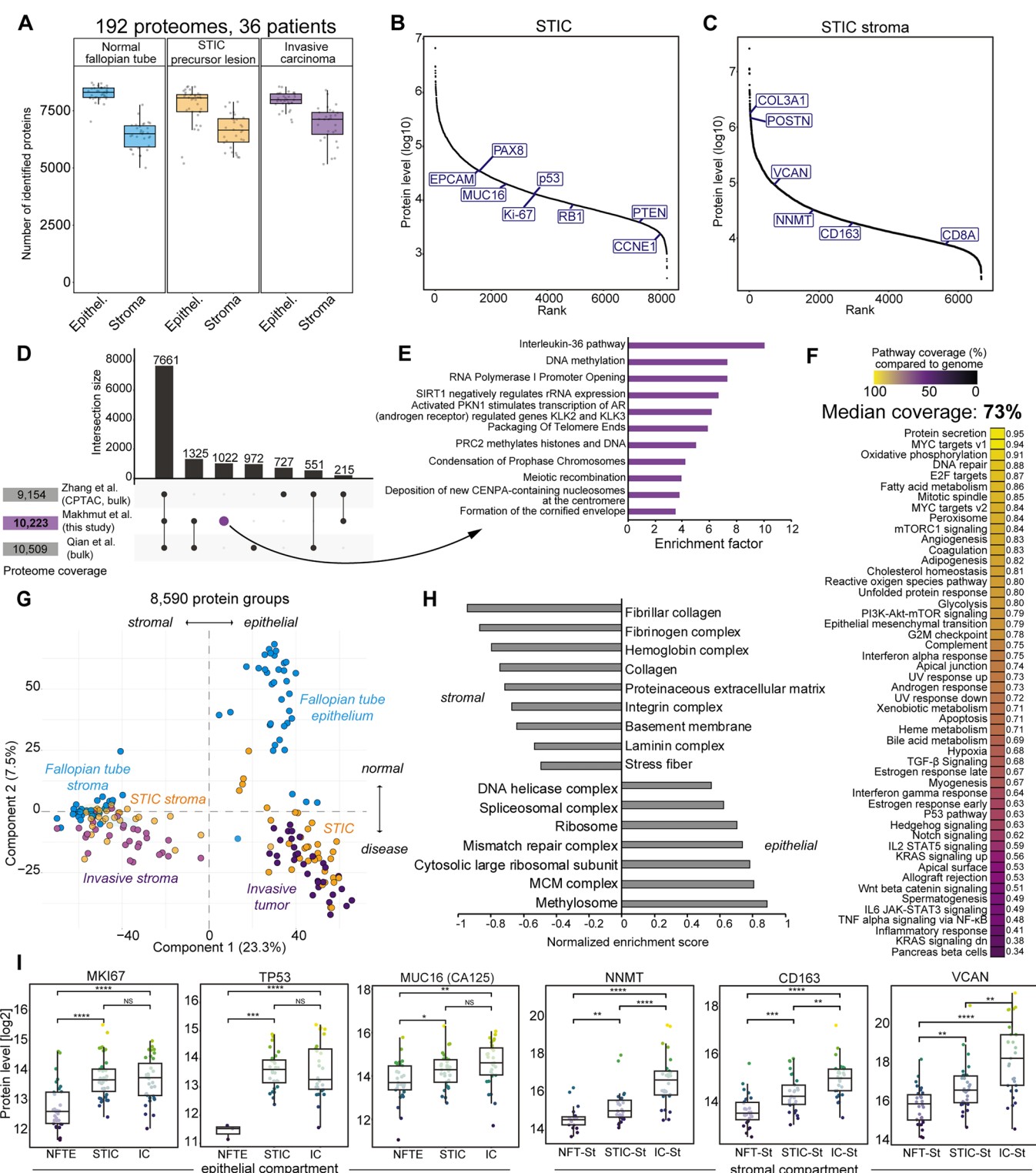

completely covered by the quantified proteins. Principal component analysis (PCA) of all 192 proteomes clearly delineated disease-associated changes in the epithelial and stromal compartments, separating healthy control regions from the STIC precursor and invasive regions (Fig. 2G). Principal component 1 (PC1, 23.3% total variability) separated epithelial cells from stromal proteomes (Fig. 2H), whereas PC2 displayed a transition from healthy to invasive regions in both compartments (7.5% total variability; Figs. 2G and EV2C). Dysregulated processes along the progression from healthy to invasive regions included known HGSOC-associated processes, such as increased DNA replication, DNA repair, and inflammation signatures (Fig. EV2C). Known disease drivers and

**Figure 2. Spatially resolved proteomes reflect disease-specific alterations at bulk level depth.**

(A) Boxplots showing the number of proteins identified in the epithelial and stromal sample groups. Boxplots define the range of the data (whiskers), 25th and 75th percentiles (box), and medians (solid line). Number of samples per group: NFTE – 35, STIC – 35, IC – 31, NFT stroma – 31, STIC stroma – 30, IC stroma-29. (B, C) Dynamic range plots of median protein abundance for epithelial (B) and stromal (C) STIC samples. Known ovarian cancer, cell type, and stromal markers are highlighted. Proteins with at least 50% valid values for each group are shown. (D) Upset plot comparing common and unique proteins identified in this study and in Qian et al (Qian et al, 2024) and Zhang et al (Zhang et al, 2016) bulk proteome datasets. (E) Overrepresented pathways (Reactome) for the 1022 proteins uniquely identified in this study (Benjamini–Hochberg FDR <0.05). (F) Median pathway coverage of hallmark gene sets based on the 10,223 proteins identified in this study. (G) Principal component analysis (PCA) of all 191 samples based on 8590 protein groups. (H) Pathway enrichment analysis (gene ontology terms) for PC1, separating all epithelial and stromal samples. Representative terms are shown with a Benjamini–Hochberg FDR <0.05. (I) Boxplots of the relative protein levels (log2) of the selected epithelial and stromal HGSOC markers. Boxplots define the range of the data (whiskers), 25th and 75th percentiles (box), and medians (solid line). Point colors reflect protein levels and are for visual guidance, where brighter colors indicate higher protein level and darker colors indicate lower protein levels. Asterisks indicate two-sided $t$-test $p$ values ($p > 0.05$, NS; $p < 0.05$, *; $p < 0.01$, **; $p < 0.001$, ***; $p < 0.0001$, ****. NFT-St NFT stroma, STIC-St STIC stroma, IC-St invasive stroma). Number of samples per group: NFTE – 35, STIC – 35, IC – 31, NFT stroma – 31, STIC stroma – 30, IC stroma-29. Exact $p$ values are provided in the Appendix Table S2.

functional markers were upregulated with disease progression. For example, STICs and ICs were characterized by significantly higher Ki67 levels, reflecting their high proliferative state, and strongly elevated p53 levels compared to NFTE (Fig. 2I, $p < 0.001$). Interestingly, the strong increase in p53 protein levels was not only specific to tumors with stabilizing missense mutations (compared to p53 null tumors, Fig. EV2D), but also marked the only significantly expressed protein between these two sample groups (Fig. EV2D). The stromal compartment featured strong extracellular matrix and cell-type-related changes associated with disease progression. For example, the cancer-associated fibroblast regulator NNMT (Eckert et al, 2019), CD163, a marker of tumor-associated macrophages (TAMs) (Lecker et al, 2021), and oncogenic proteoglycan VCAN (Cheon et al, 2014) were significantly upregulated from the NFT stroma to the invasive stroma (Fig. 2I). Taken together, our deep and compartment-resolved proteomic dataset provides a rich resource for studying the early steps of HGSOC development and progression by incorporating tissue-matched healthy control regions, STIC precursor lesions, and IC.

## Precursor lesions feature histological markers and cell-of-origin signatures

Although there is accumulating evidence that HGSOC originates from fallopian tube epithelial cells (Perets et al, 2013; Kim et al, 2012), a global and cell-type-resolved proteomic comparison of these putative precursor cells with concurrent STICs and ICs remains elusive. Furthermore, little is known about the proteomic landscape and heterogeneity of STICs and their molecular deviation from NFTE to ultimately form invasive tumors. To address this, we isolated small stretches of NFTE guided by PAX8 IHC and isolated concurrent STIC regions and IC 'nests' of the same area (total 50,000 µm³). Principal component analysis revealed strong proteome differences between the three sample groups, with NFTE clearly grouping apart from the disease states (Fig. 3A). Interestingly, four NFTE samples grouped much closer to STICs and ICs, indicating that they likely did not represent NFTE (here referred to as atypical). Proteins upregulated in these samples were strongly enriched for cancer-associated pathways, such as higher DNA replication, cell cycle, and DNA double-strand break repair signatures (Fig. 3B,D). The most upregulated protein was the cell cycle regulator CDKN2A (p16/INK4A, Dataset EV4), an established histological marker for the diagnosis of STIC with a p53 null

phenotype (Novak et al, 2015). Informed by proteomics, we therefore re-stained these tissues against p16 and evaluated whether they were normal-like or indeed malignant lesions. Indeed, we observed strong and uniform p16 staining (Fig. 3C) and a morphology suspicious for an STIC for two of them, leading to the re-diagnosis of two of these samples as serous tubal intraepithelial lesions (STIL) (Kuhn et al, 2012; Vang and Shih, 2022) and STIC for the other two. The absence of p53 positivity (p53 null phenotype) and only marginal morphological changes in the epithelium potentially led to their misclassification. We also noticed that two samples initially labeled as STIC grouped closer to the NFTE. Reanalysis of these samples revealed that one was the only STIC sample with no associated IC, likely representing an earlier disease state captured by proteomics. The second one represented an STIC with a stabilizing p53 missense mutation but a p53 null phenotype in the invasive component. Pairwise comparison between the four atypical fallopian tube lesions and the NFTE also revealed the presence of ciliated cell signatures in the NFTE samples (Fig. 3D; Dataset EV5), potentially due to the co-isolation of ciliated cells. To investigate this further, we integrated data from a recent single-cell transcriptomic atlas of the healthy human fallopian tube (Dinh et al, 2021) to assess which cell type and cell states were present in our dataset. This resulted in a binary clustering, marking roughly half of our NFTE samples as secretory cell-enriched and the other half as more ciliated (Fig. EV3B; Table EV2). As expected, the cell type marker PAX8 (secretory cells), expressed in 96% of HGSOCs (Nonaka et al, 2008), and the transcription factor FOXJ1 (ciliated cells) were among the most significantly regulated proteins in these two clusters (Fig. 3E; Dataset EV6). We therefore next asked whether these two distinct cell-type signatures present in our dataset could help to unmask cell-of-origin signatures preserved in STICs and invasive tumors. We found that the secretory-like NFTE cluster grouped closely with all malignant stages (STIC and IC) and was clearly apart from the ciliated-like cluster (Fig. 3F). Co-expressed markers in the carcinoma/secretory NFTE group included known histological markers, such as PAX8 and STMN1, a proposed ancillary marker for detecting STIC lesions with a p53 null phenotype (Fig. 3G, Novak et al, 2015). Other proteins with similar cell-type specific and disease-associated expression included DHCR24, an oxidoreductase important in cholesterol biosynthesis (Waterham et al, 2001), the collagen-specific chaperone SERPINH1, the calcium-binding protein S100A4, as well as the basal cell adhesion protein BCAM, and 27 other proteins with similar abundance profiles (Figs.

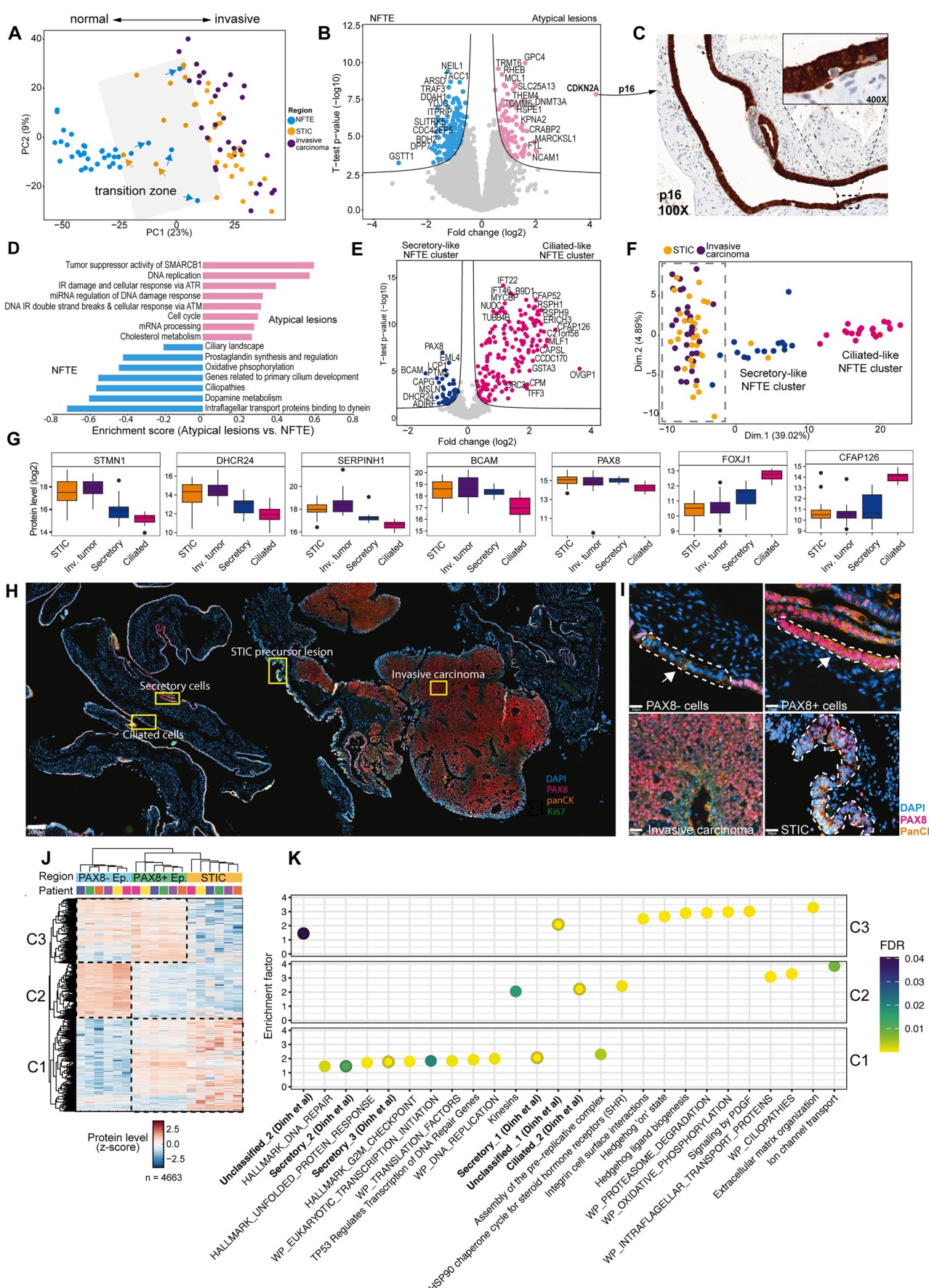

Figure 3. Precursor lesions feature histological markers and cell-of-origin signatures.

(A) Principal component analysis (PCA) of 101 NFTE, STIC, and invasive carcinoma proteomes based on ANOVA-significant 3564 proteins. (B) Volcano plot of pairwise proteomic comparison between atypical (light-pink, four samples) and NFTE samples (light blue, 31 samples). Markers with the highest fold change are highlighted (two-sided *t*-test, permutation-based FDR <0.05). (C) Representative immunohistochemical staining of CDKN2A (p16) in one atypical fallopian tube epithelial region. (D) Pathway enrichment analysis based on the *t*-test difference between premalignant and normal fallopian tube epithelial samples. Selected significantly enriched pathways are shown for pathways higher in atypical samples (light-pink) and in NFTE samples (light blue) (Benjamin–Hochberg FDR <0.05). (E) Volcano plot of the pairwise proteomic comparison between ciliated-like epithelial (dark pink, 20 samples) and secretory-like epithelial (dark blue, 15 samples) samples. Known secretory (e.g., PAX8) and ciliated (e.g., FOXJ1) markers are highlighted (two-sided *t*-test, permutation-based FDR <0.05). (F) Principal component analysis of all epithelial samples based on 632 proteins overlapping with cell type-specific markers identified by single-cell transcriptomics(Dinh et al, 2021). (G) Boxplots of STMN1, DHCR24, SERPINH1, BCAM, PAX8, FOXJ1, and CFAP126 relative protein levels (log2) in STIC (yellow, 35 samples), invasive cancer (purple, 31 samples), secretory NFTE cluster (dark blue, 15 samples), and ciliated NFTE cluster (dark pink, 20 samples) samples. Boxplots define the range of the data (whiskers), 25th and 75th percentiles (box), and medians (solid line). (H) Immunofluorescence image of one representative HGSOC tissue section stained for PAX8, Ki67, panCK, and DAPI (DNA). Yellow boxes highlight regions of invasive carcinoma, STIC precursor lesion, and normal fallopian tube ciliated and secretory cells, which were sampled for proteomic profiling. (I) Magnified tissue regions from immunofluorescence images in panel (H). Arrows show exemplary epithelial regions of the PAX8+ and PAX8- regions used for proteomic profiling. Scale bar = 20 μm. (J) Unsupervised hierarchical clustering of all ANOVA-significant proteins between STICs, secretory, and ciliated fallopian tube epithelial cells obtained from six patients. Relative protein levels (z-scores) are shown. Three distinct clusters, C1, C2, and C3, are highlighted. PAX8+ secretory cells clustered together with STICs and were separated from ciliated PAX8−samples. (K) Pathway enrichment analysis (Reactome, Hallmark, WikiPathways, and cell type signatures from Dinh et al (Dinh et al, 2021)) showed significantly overrepresented pathways for clusters C1, C2, and C3 with a Benjamini–Hochberg FDR <0.05.

3G and EV3D). To further validate that STICs are globally related to secretory cells, we performed additional immunofluorescence whole-slide imaging of six tissue sections and precisely isolated PAX8+ and PAX8- cells for direct proteomic comparison (Fig. 3H,I and EV3E–G). PAX8+ secretory cell proteomes showed strong overlapping signatures with STICs and clustered apart from all PAX8- epithelial samples (Figs. 3J,K and EV3H). Secretory cell-specific signatures were strongly enriched in the STIC/PAX8+ cluster with high p16, p53, and PAX8 protein levels and pathways related to cell cycle, unfolded protein response and DNA repair. In contrast, PAX8- cells expressed high levels of ciliated cell-specific cilium and intra-flagellar transport proteins, as well as FOXJ1 (Figs. 3K and EV3F–I; Dataset EV7–10). The NFTE cluster, which comprised PAX8+ and PAX8- cells, was enriched for previously identified transition signatures between secretory and ciliated cells (unclassified clusters 2 and 3) (Dinh et al, 2021), consistent with the expected tissue composition and possibly capturing the previously proposed secretory-to-ciliated cell differentiation programs in the fallopian tube epithelium. In summary, this cell-type resolved proteomic analysis revealed strong cell-of-origin signatures present in STIC and concurrent carcinomas, supporting the view that HGSOC originates in secretory cells of the distal FT. Moreover, our analysis highlighted several new proteins as potential ancillary markers for the detection of p53 null phenotype STIL and STIC lesions, independent of HRD and *TP53* mutation status.

## Refined molecular subtyping from spatial proteomics data

Next, we compared all STICs and ICs. Similar to principal component analysis (PCA) (Fig. 3A), unsupervised hierarchical clustering of the 2000 most variably expressed proteins in our dataset confirmed that STICs and ICs were highly related, resulting in two distinct clusters, irrespective of sample type (STIC or IC) (Figs. 4A,B and S4A). Clustering was also independent of *TP53* mutation type, histological stage, and HRD status, but showed an association of cluster 2 samples with higher patient age and shorter overall survival (Figs. 4A and EV4B; Appendix Fig. S1). Nearly all (29/30) patient-matched STIC-IC pairs clustered together (Fig. 4A,B), underlining the strong patient-specific proteome profiles.

The high similarity between tissue-matched STICs and ICs was also reflected in their higher global proteome correlation (median Pearson $r = 0.94$, Fig. EV4C) compared to interpatient comparisons (median Pearson $r = 0.91$). Interestingly, bilateral STICs obtained from both ovaries of the same patient featured exceptionally high proteome correlations (Pearson $r = 0.97$), despite being spatially unrelated. While this pointed towards a possible clonal relatedness of these two premalignant lesions, additional genetic analyses are needed to confirm or reject this hypothesis.

Cluster 1 tumors were primarily enriched for immune-associated processes (e.g., interferon response, TNFα signaling, antigen processing, and presentation), whereas cluster 2 samples showed elevated levels of cell cycle, EMT, and DNA repair-associated proteins (Fig. 4C,D; Dataset EV11–EV12). To integrate our findings with previously identified HGSOC molecular subtypes (Zhang et al, 2016; Bell et al, 2011), we employed the consensusOV classifier (Chen et al, 2018) and assessed whether known subtypes were present in our two main clusters. 36 proteomes could be assigned to one subtype based on a margin score of greater than 0.2 (difference between the probabilities of the most-probable and second most-probable subtype, mean score = 0.54). Notably, most cluster 1 samples (12 of 17, 71%) were classified as the 'immunoreactive' (IMR) subtype, whereas cluster two samples were classified as "proliferative" (PRO) tumors, independent of HRD status (Fig. 4E). Only a few samples were classified as mesenchymal ($n = 4$) or differentiated ($n = 5$). We mainly attribute this discrepancy to our compartment-resolved strategy that contains minimal stromal admixing, thereby enabling more accurate tumor subtyping. Using single-cell RNA sequencing, previous studies have contextualized HGSOC bulk subtypes and identified "differentiated" (DIF) and PRO subtypes as tumor cell-specific signatures (Geistlinger et al, 2021). In contrast, stromal and immune cells are associated with the "mesenchymal" (MES) and IMR subtypes, respectively (Izar et al, 2020). The study by Geistlinger et al also revealed that the DIF subtype is associated with the strongest lymphocyte infiltration, letting us speculate that cluster 1 tumors could likely represent DIF tumors whose proteomes were strongly masked by pronounced intra-tumoral immune infiltration. Indeed, not only was the second-best subtype assignment for IMR tumors the DIF subtype (Fig. EV4G), but cyclic

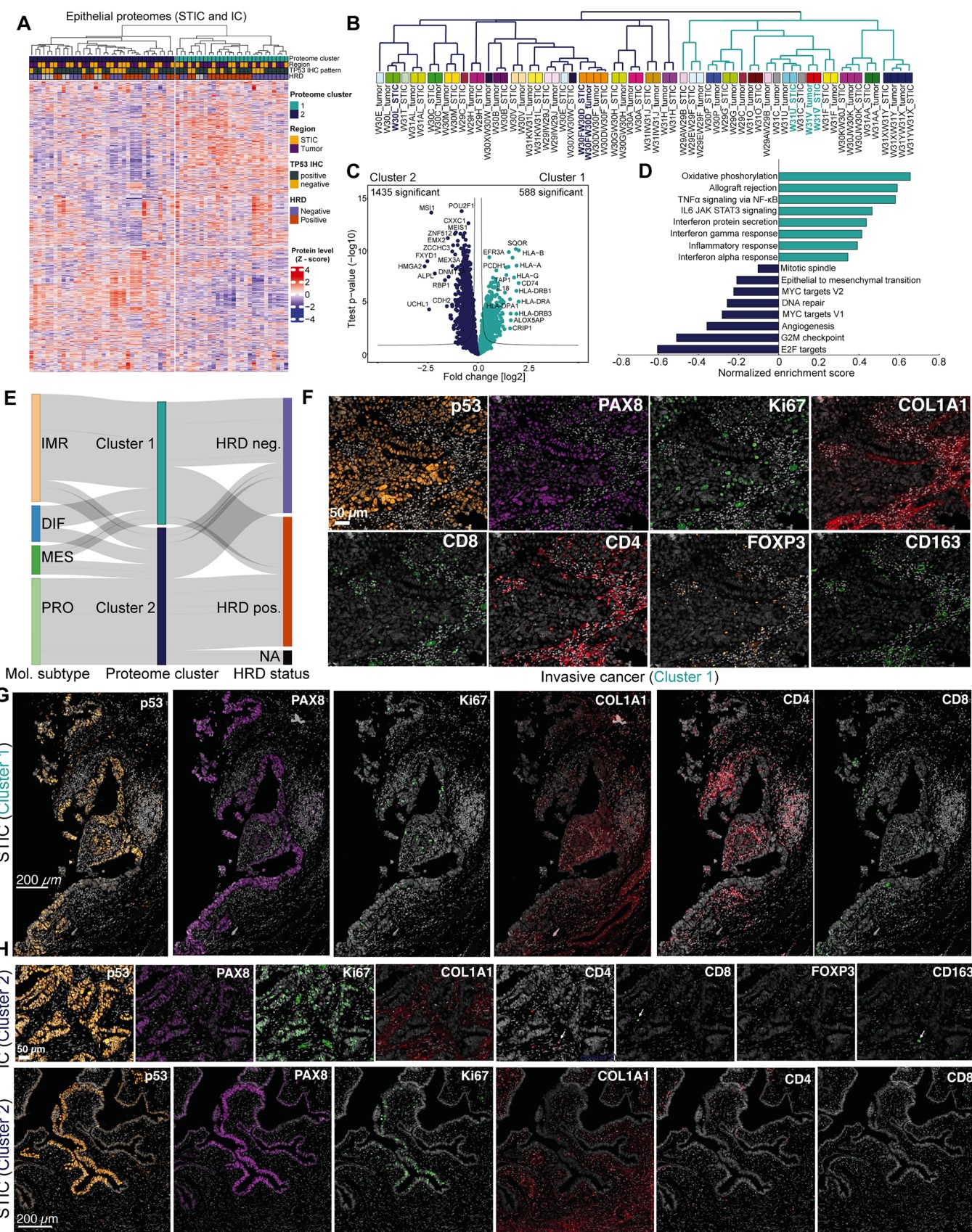

**Figure 4. Refined molecular subtyping from spatial proteomics data.**

(A) Unsupervised hierarchical clustering of all 64 STIC and IC samples based on the 2000 most variable proteins (highest median absolute deviation), showing two clusters. Relative protein levels (z-score) are shown with clinicopathological information as color bars. (B) Cluster dendrogram related to panel (A). Selected cluster 1 and 2 patients shown in panels 4F-H are highlighted in bold. (C) Volcano plot of pairwise proteomic comparison between cluster 1 (turquoise, 28 samples) and cluster 2 (dark blue, 36 samples) samples. Proteins with the highest fold change are highlighted (two-sided Student's *t*-test, false discovery rate [FDR] <0.05). (D) Pathway enrichment analysis (WikiPathways, Hallmarks) based on the *t*-test difference between cluster 1 and cluster 2 epithelial samples. Selected pathways with a Benjamin–Hochberg FDR <0.05 are shown. (E) Sankey plot illustrating the distribution of STIC & IC samples across consensusOV molecular subtypes, cluster 1 or cluster 2, and HRD statuses. The width of each flow corresponds to the proportion of the samples within each category, highlighting the relationship between molecular subtype, cluster assignment, and HRD status. (F–H) CyCIF of one representative immune-enriched cluster 1 tumor and STIC (patient W31V, (F) – tumor, (G) - STIC) and one immune-deserted cluster 2 tumor and STIC (patient W30FW30D, (H), top – tumor, bottom - STIC) confirms pronounced differences in Ki67+ tumor cells and intra-tumoral immune cell infiltration. Scale bar = 50 μm (STIC), 200 μm (tumor).

immunofluorescence imaging (CyCIF) also confirmed strong immune cell infiltration in this subgroup (CD8+ cytotoxic T cells, CD4+ helper T cells, FOXP3+ regulatory T cells, CD163 + M2-like TAMs, and CD11c+ dendritic cells) (Fig. 4F,G; Appendix Figs. S4, S5). Conversely, cluster 2 tumors showed negligible immune infiltration based on proteomics and CyCIF but had a larger fraction of Ki67+ tumor cells, in line with their proliferative nature (Fig. 4H; Appendix Figs. S4, S5).

We next addressed whether STICs and ICs featured similar molecular subtypes and whether the progression from STIC to IC was associated with a change in molecular subtype identity. Ten matching STIC-IC pairs could be clearly assigned to one of the four subtypes. We found a similar subtype frequency at both stages (Fig. EV4F), suggesting a similar level of phenotypic and proteomic heterogeneity. For the majority of paired cases (70%), the STIC and IC subtypes were coherent (Fig. EV4H), underlining the absence of a distinct epithelial STIC proteotype distinguishable from ICs. This finding was supported by the global proteome comparison, which revealed only a few differentially regulated proteins (9 out of 6744) between all STIC and IC samples (Fig. EV4I) and exceptionally high proteome correlations of patient-matched STIC-IC pairs (Fig. EV4C).

## Mapping progressive ECM remodeling reveals stromal drug targets

We next focused our attention on the tumor microenvironment (TME). The TME comprises various stromal and immune cell types and the extracellular matrix (ECM) and is critically involved in all phases of tumorigenesis (Pickup et al, 2014; Winkler et al, 2020; Prakash and Shaked, 2024). As the TME is increasingly recognized as a promising therapeutic target across cancer entities (Bejarano et al, 2021), exploratory proteomics data comparing normal, precursor, and invasive stromal tissue regions offer a particularly important biomedical resource. While the broad dynamic range of protein abundance in the ECM generally poses a significant obstacle for deep proteome profiling (Naba, 2023), our analysis still yielded more than 6,000 proteins for each stromal microregion, distributed over four orders of magnitude (Fig. 2A,B). Principal component analysis revealed a disease gradient from NFT stroma over STIC stroma (STIC-St) to invasive stroma (IC-St) (Fig. 5A). Interestingly, in contrast to our epithelial findings that showed that STICs and ICs were proteomically highly related and distinct from the NFTE, STIC-St proteomes showed both normal-like and malignant signatures. This observation was supported by unsupervised hierarchical clustering of the 2888 differentially abundant

proteins, representing roughly one-third of our analyzed stromal proteome (Fig. 5B; Dataset EV13). We found that stromal clustering could generally not be explained by the two tumor clusters (Fig. 5B), emphasizing the importance of compartment and cell-type-resolved analyses to study spatially-defined changes during disease progression. Nevertheless, immune-related and inflammatory pathways were clearly higher in the stroma of immune-enriched cluster 1 tumors (Appendix Fig. S2). The invasive stroma cluster showed high inflammation, hypoxia and VEGF signaling, accompanied by strong lymphocyte and myeloid cell signatures (e.g., T-cell, B-cell, macrophage, and NK-cell). Conversely, the normal-like cluster, which included half of the STIC stroma group, was enriched for fibroblast and endothelial cell signatures, as well as core ECM functions, indicating a structurally different and more intact ECM. This prompted us to more systematically analyze the quantitative ECM changes during HGSOC development and to this end filtered our data for all quantified matrisome-associated proteins, as recently cataloged (Renner et al, 2022). From the total of 467 matrisome-related proteins in our dataset, we identified 95 (20%) as differentially abundant between NFT and invasive stroma (Figs. 5C and EV5A; Dataset EV14). Proteins with the highest significance included known ECM degraders, such as cathepsins (CTSA, CTSC, CTSS, and CTSZ), matrix metalloproteinases (MMP11 and MMP14), pro-inflammatory cytokines (TGFβ1, TGFβ2, IL16, and IL18), and insulin-like growth factor-binding proteins (IGFBP2, IGFBP7). We identified strong structural ECM changes, as evident from the downregulation of multiple collagen isoforms (e.g., collagen types 4, 6, 11, and 21) during HGSOC progression (Fig. 5D,E). Our collagen isoform-resolved data also enabled us to quantify a common decrease in the COL3/COL1 ratio in 27 of 32 tissues (84% of the cohort), a marker for fibrosis, ECM stiffening, and tumor progression (Brisson et al, 2023; Sinha et al, 2022) (Table EV3). An exception to the general decrease in collagen abundance with disease progression were collagens 8A1, 8A2, and 10A1, which showed higher levels in the invasive stroma (Fig. 5D). Notably, these three collagen isoforms were previously linked to myofibroblasts (myCAFs) in pancreatic cancer and were associated with unfavorable clinical outcomes (Thorlacius-Ussing et al, 2024). This underlines the central role of CAFs as key drivers for oncogenic ECM remodeling in our cohort, which was further supported by a consistent upregulation of several other CAF markers, such as NNMT, TNC and FAP (Dataset EV15). We also identified other collagen isoforms, such as COL9A3, COL9A1, COL23A1, and COL13A1, not previously reported in HGSOC and whose specific functions remain to be elucidated.

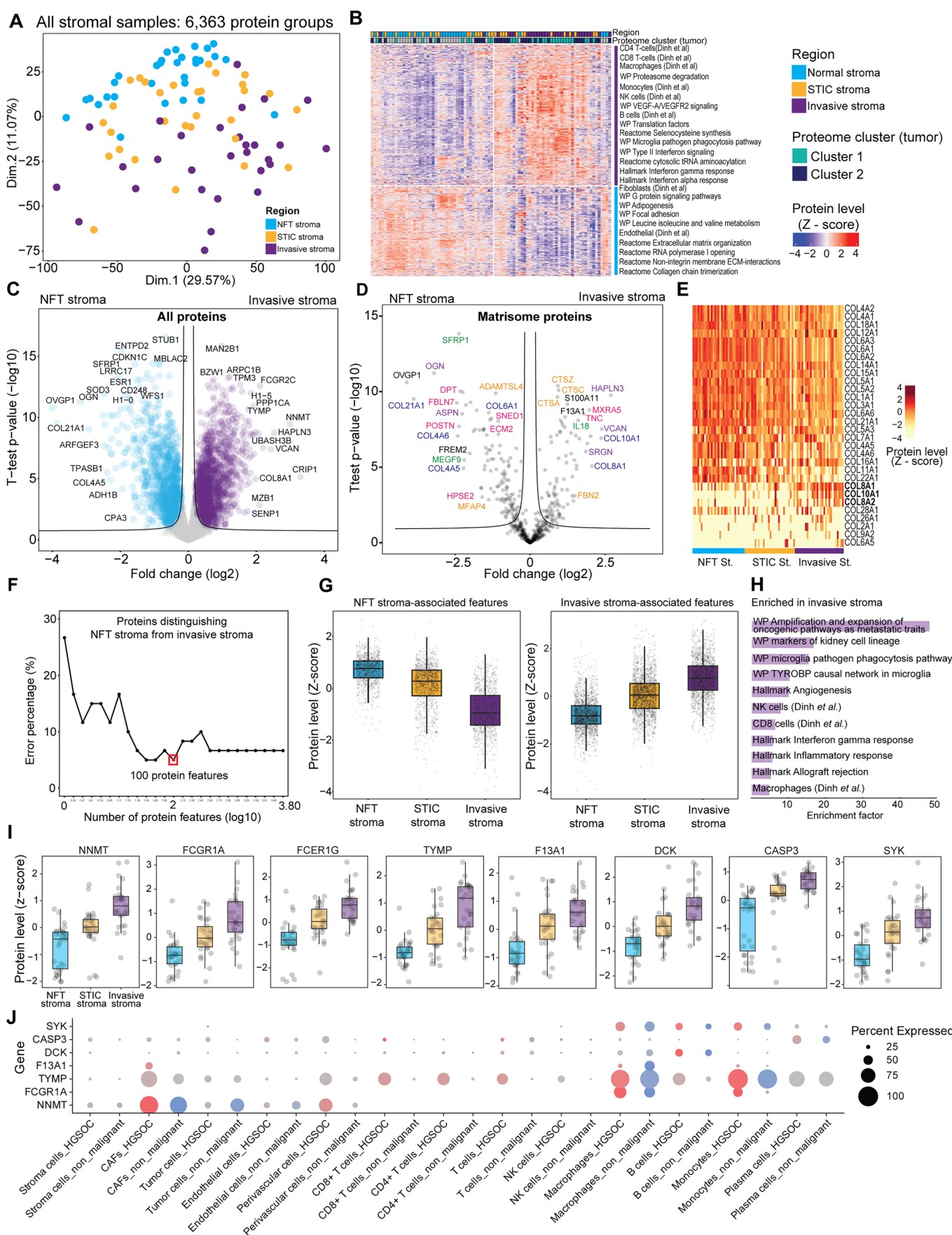

**Figure 5.  Mapping progressive ECM remodeling reveals stromal drug targets.**

(A) PCA of all 90 stromal proteomes based on 6363 protein groups. PC1 and PC2 accounted for 40.64% of the total data variability. (B) Unsupervised hierarchical clustering of all stromal proteomes based on 2888 ANOVA-significant proteins (permutation-based FDR <0.05). Relative protein levels (z-score) are shown with clinicopathological and tumor subtype information as color bars. A subset of enriched pathways (WikiPathways, cell-type signatures obtained from Dinh et al (Dinh et al, 2021)) for the two main row clusters marked in purple and green is shown. (C) Pairwise proteomic comparison between invasive (28 samples) and NFT stroma (30 samples) (two-sided *t*-test) for all proteins. Proteins with a permutation-based FDR <0.05 were considered significant. Orange: ECM regulators; green: secreted factors, pink: ECM glycoproteins; blue: core matrisome; purple: proteoglycans. (D) Pairwise proteomic comparison between invasive (28 samples) and NFT stroma (30 samples) (two-sided *t*-test) for matrisome-associated proteins. Proteins with a permutation-based FDR <0.05 were considered significant. Orange: ECM regulators; green: secreted factors, pink: ECM glycoproteins; blue: core matrisome; purple: proteoglycans. (E) Relative protein levels (z-score) of all quantified collagens across NFT stroma (light blue), STIC stroma (yellow), and invasive stroma (purple) samples. (F) Support vector machine-based identification of top-ranked protein features distinguishing NFT stroma from IC stroma. (G) Boxplots of the relative protein levels (z-score) of the top 100 proteins distinguishing NFT from invasive stroma. Boxplots define the range of the data (whiskers), 25th and 75th percentiles (box), and medians (solid line). Left: Proteins that were higher in the NFT stroma. Right: Proteins that were higher in the IC stroma. NFT stroma – 31 samples, STIC stroma – 30 samples, IC stroma – 29 samples. (H) Overrepresented pathways (FDR <0.05) of the top 100 proteins compared to all proteins in the dataset. (I) Boxplots of relative protein levels (z-score) of FDA-approved drug targets. Boxplots define the range of the data (whiskers), 25th and 75th percentiles (box), and medians (solid line). NFT stroma – 31 samples, STIC stroma – 30 samples, IC stroma – 29 samples. (J) Dotplot showing RNA expression of NNMT, FCGR1A, TYMP, F13A1, DCK, CASP3, and SYK across different cell types in HGSOC and non-malignant ovarian samples. Dot size represents the percentage of cells expressing each marker, and color indicates expression level (gray to red for HGSOC, gray to blue for non-malignant samples).

To prioritize protein drug targets significantly upregulated in the invasive stroma, we employed support vector machine (SVM) classification combined with feature ranking and retrieved a 100-protein stromal signature that robustly (5% error rate) separated all NFT and malignant stromal proteomes (Fig. 5E,F). Notably, protein levels of the STIC stroma samples were between the NFT and invasive stroma levels, indicating a gradual decrease or increase of the selected proteins (Fig. EV5B,C). Our signature included many known proteins of the desmoplastic stroma previously linked to poor patient outcome (Cheon et al, 2014) (e.g., VCAN, THBS2, and TNC), pro-inflammatory cytokines (IL18), immune response and complement system modulators (e.g., F13A1, FCGR1A, and FCER1G) and markers of tumor promoting cell-types (e.g., NNMT for CAFs and CD163 and MSR1 for M2-polarized TAMs)(Eckert et al, 2019; Gudgeon et al, 2022; Lecker et al, 2021) (Figs. 5G and S5B,C; Dataset EV16). Notably, eight proteins upregulated in the invasive stroma (NNMT, FCGR1A, FCER1G, TYMP, F13A1, DCK, CASP3, and SYK) represented FDA-approved drug targets outside of ovarian cancer (Fig. 5I,J), emphasizing their high relevance for future preclinical investigations. Integration with publicly available single-cell sequencing data revealed that these proteins were mostly macrophage and monocyte-associated with malignant features (TYMP, FCER1G, FCGR1A, SYK, and F13A1), except NNMT, which was highest in cancer-associated fibroblasts (Fig. 5J; Appendix Fig. S3; Table EV4).

## Identification of early dysregulated pathways of therapeutic relevance

To identify commonly dysregulated pathways during early HGSOC development, we compared all NFTE to STICs. Such data are of high translational importance, for example, for the development of new disease prevention and therapeutic strategies (Zhang et al, 2024). To this end, we compared all secretory-like NFTE samples with their corresponding STICs. Overall, we observed pronounced proteomic differences with 615 differentially abundant protein groups (Fig. 6A; Dataset EV17), in stark contrast to the STIC *versus* IC comparison that only yielded nine differentially expressed proteins (Fig. EV4I). Proteins of highest significance and fold change included DNA replication proteins (e.g., MCM2, 3, 4, 5, and 7), the cell cycle regulator CDKN2A (p16-INK4A), the microtubule

and cell cycle regulator STMN1, and the insulin growth factor-binding protein IGFBP2, which was previously reported to be upregulated in STICs through DNA hypomethylation (Wang et al, 2022). Notably, we also discovered several dysregulated metabolic pathways (Figs. 6B,C and EV6A; Dataset EV18). For instance, protein levels related to prostaglandin biosynthesis were higher in NFTE than in STIC and IC (Fig. 6C), reflecting the important physiological role of the fallopian tube in hormone regulation (Paik et al, 2012), and the loss of epithelial cell function during carcinogenesis. Conversely, the glycolysis and cholesterol biosynthesis/mevalonate pathways were prominently elevated in carcinomas, coinciding with a decrease in oxidative phosphorylation (OxPhos). Our data further showed that these metabolic pathway changes were independent of the molecular subtype (Figs. 6D and EV6B; Dataset EV19), suggesting that metabolic adaptation to a glycolytic and cholesterol-dependent state is a common and likely early event during HGSOC development. We found several key enzymes of the cholesterol biosynthesis pathway were upregulated in STICs and ICs compared to NFTE (Fig. 6E). Most prominently, the enzymes dehydrocholesterol-reductase 24 (DHCR24) and 7 (DHCR7) were upregulated by 5.2 and 3.3-fold, respectively, which catalyze the final steps in cholesterol biosynthesis. Integrating a large-scale pan-cancer study (Ben-David et al, 2018; Gao et al, 2013) comparing 1739 cell lines of different tumor origins confirmed high DHCR24 and DHCR7 mRNA levels in HGSOC (Fig. EV6C,D). Furthermore, we orthogonally validated the upregulation of DHCR7 and DHCR24 in STICs and invasive regions versus NFTE based on IHC in 36 samples (Fig. 6F,G).

## Cholesterol biosynthesis inhibition sensitizes ovarian cancer cells to carboplatin

Next, we asked whether blocking the cholesterol biosynthesis pathway via dehydrocholesterol-reductases could offer a novel and potent approach for HGSOC treatment. To test this, we selected four ovarian cancer cell lines (OAW-42, OVCAR-8, ES-2, and EFO-21) of high molecular similarity to HGSOC tumors (Sinha et al, 2021) based on mutation, copy number and gene expression profiles for treatment with the DHCR7 inhibitor AY9944 (Datasets EV20 and 21). While OVCAR-8, EFO-21 and ES-2 were *TP53* mutant lines, we also chose the *TP53* wildtype OAW-42 model as

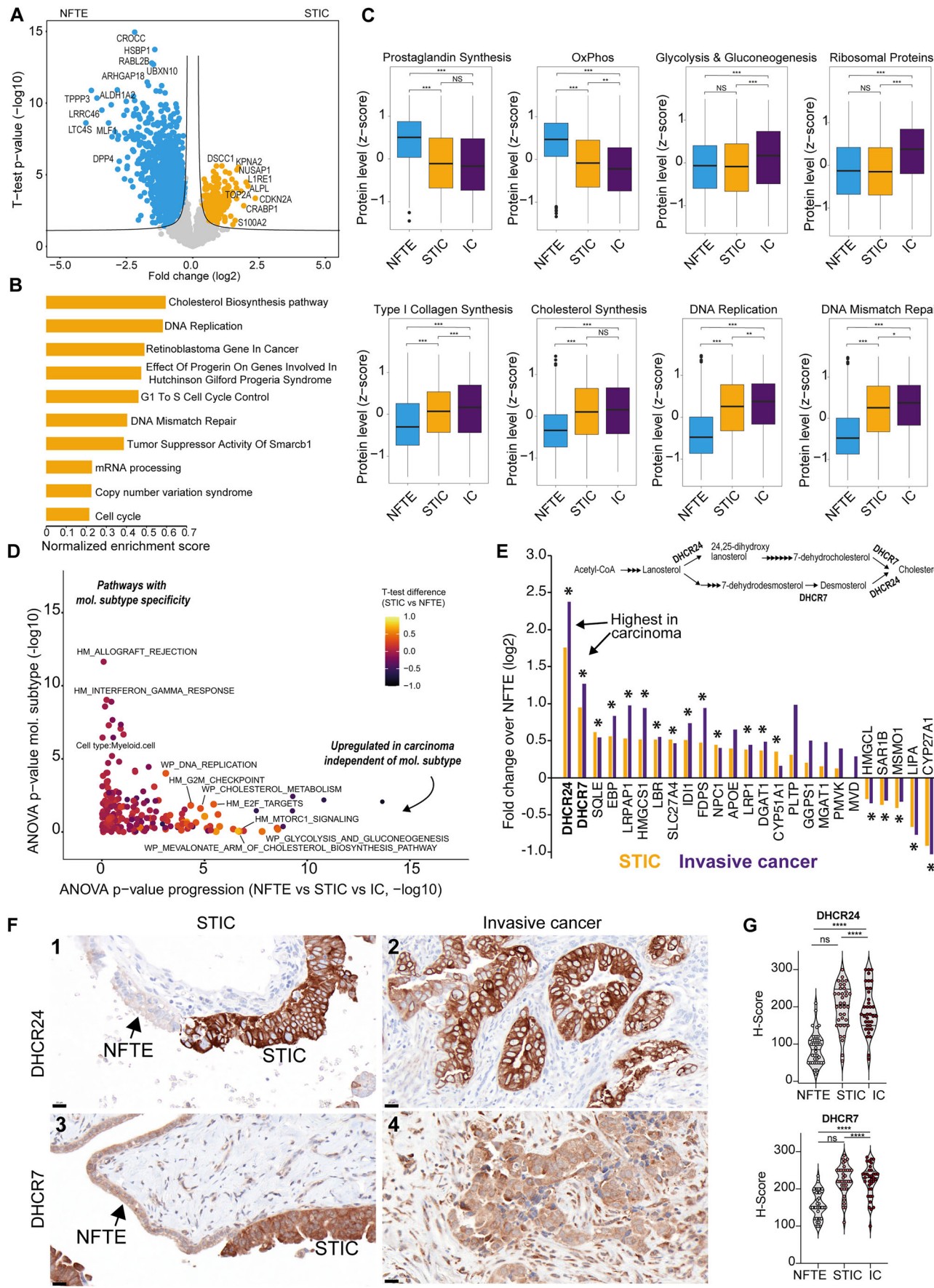

**Figure 6. Identification of commonly dysregulated pathways of therapeutic relevance.**

(A) Volcano plot of the pairwise proteomic comparison between STIC (yellow) and secretory cell-enriched NFTE samples (light blue). Proteins with the highest fold change were highlighted (two-sided *t*-test, permutation-based FDR <0.05). (B) Pathway enrichment analysis (WikiPathways, Benjamin–Hochberg FDR <0.05) revealed processes upregulated in STICs. (C) Boxplots of relative protein levels (group average, z-score) for selected pathways. Boxplots define the range of the data (whiskers), 25th and 75th percentiles (box), and medians (solid line). OxPhos: oxidative phosphorylation. Asterisks indicate two-sided *t*-test *p* values ($p > 0.05$, NS; $p < 0.05$, *; $p < 0.01$, **; $p < 0.001$, ***). Number of samples per group: NFTE – 35, STIC – 35, IC – 31. Exact *p* values are provided in the Appendix Table S3. (D) Two-dimensional ANOVA showing pathways significantly affected by disease progression (x-axis) and molecular subtype (y-axis). Note that metabolism-related terms such as the mevalonate and cholesterol biosynthesis pathways are upregulated in carcinoma, independent of molecular subtype. (E) Ranked protein fold changes (STIC and IC *vs*. NFTE, log2) of all quantified cholesterol biosynthesis/metabolism pathway proteins. Most proteins show higher expression in STICs and ICs. Proteins significantly changed in both comparisons (STIC vs. NFTE and IC vs. NFTE, FDR <0.05) are highlighted with asterisks. (F) Representative immunohistochemistry (IHC) staining for DHCR24 (upper panel: 1 – serous tubal intraepithelial carcinoma [STIC] and normal fallopian tube epithelium [NFTE]; 2 – invasive carcinoma) and DHCR7 (lower panel: 3 – STIC and NFTE; 4 – invasive carcinoma) at 20× magnification (scale bar = 20 μm). Images illustrate staining patterns across NFTE, STIC, and invasive carcinoma regions in 33 patients (36 samples, three patients displayed bilateral STIC and IC). Wilcoxon matched pairs signed rank test: ns (not significant), ****$p < 0.0001$. (G) Violin plots showing distribution of DHCR24 and DHCR7 immunohistochemical H-scores in NFTE, STIC, and invasive carcinoma regions in 33 patients (36 samples, three patients displayed bilateral STIC and IC). H-scores were determined by assessing the percentage of cells at each staining intensity (0 = none, 1 = weak, 2 = moderate, 3 = strong) and summing the products of each percentage multiplied by its corresponding intensity score, giving a total score ranging from 0 to 300. A paired Wilcoxon signed rank test was used to compare the median between the groups (NFTE, STICs, and ICs). Exact *p* values are provided in the Appendix Table S4.

previous studies revealed a causal link between p53 deficiency and elevated levels of the mevalonate/cholesterol synthesis pathway (Freed-Pastor et al, 2012; Moon et al, 2018). For comparison to DHCR7 inhibition, we included the clinical cholesterol inhibitor simvastatin and four doses of carboplatin as the main chemotherapeutic drug for HGSOC treatment (Fig. 7A). DHCR7 inhibition resulted in a strong reduction in cell viability in all four cell lines, but with notable differences in the approximated IC-50 values. ES-2 (IC-50 = 2.44 μM) and OVCAR-8 (IC-50 = 6.72 μM) were most sensitive to DHCR7 inhibition, consistent with their responses to simvastatin (Figs. 7B and EV7). EFO-21 responded to the DHCR7 inhibitor (IC-50 = 7.25 μM), but was more resistant to simvastatin (IC-50 >20 μM). OAW-42 cells instead required 2–3-fold higher doses of AY9944 (IC-50 = 12 μM) compared to the three *TP53* mutant lines. While carboplatin treatment alone resulted in variable responses consistent with the expected high levels of chemoresistance, we observed significant drug synergy in all cell lines except OAW-42 (Fig. 7C; Appendix Table S1). Notably, in accordance with the DHCR7 responses, drug synergy was also strongest in OVCAR-8 and EFO-21, the models most resistant to carboplatin. Together, our patient and in vitro data support the notion that blocking the cholesterol biosynthesis pathway, possibly through dehydrocholesterol-reductase inhibition, could be a promising and previously hidden avenue for ovarian cancer treatment.

# Discussion

Despite the common view that STICs represent the precursor lesion for the majority of HGSOCs, their global proteomic landscape and relationship with NFTE and concurrent invasive tumors have remained largely unaddressed. Building on our previously developed frameworks for deep spatial proteomics of ultra-low-input tissue (Makhmut et al, 2023) and deep visual proteomics (Mund et al, 2022), we performed a comprehensive proteomic analysis of HGSOC precursor lesions, quantifying over 10,000 proteins. Such deep proteome coverage was previously obtained from hundreds of micrograms to milligrams of freshly frozen tissue (Zhang et al, 2016; Qian et al, 2024). Closely guided by histopathology and whole-slide imaging, this approach enabled us to dissect early changes in the disease-related proteome and to unravel the proteomic heterogeneity of STICs. We quantitatively compared the proteome of STICs and adjacent NFTE, the assumed cell of origin for HGSOC, and revealed hundreds of disease-related protein level changes. Compared to traditional bulk approaches using tumor-adjacent control tissue for comparison, mostly comprised of stromal cells and ECM, our cell-type resolved approach is more accurate in identifying disease-specific alterations and functional drivers. We provide strong support for the secretory fallopian tube epithelial cell as the cell-of-origin for our analyzed tumors. Our paired proteome analysis of STICs and adjacent healthy fallopian tube epithelial cells revealed that STICs and PAX8+ epithelial cells were not only globally highly related, compared to PAX8- epithelial cells, but also co-expressed well-established histological HGSOC and secretory cell markers (e.g., PAX8 and STMN1), as well as several newly identified ones (e.g., DHCR24, BCAM, and SERPINH1). These findings are in line with recent single-cell sequencing data (Dinh et al, 2021; Hu et al, 2020) as well as HGSOC mouse models (Labidi-Galy et al, 2017; Kim et al, 2012), supporting the view that the majority of HGSOC originates from secretory cells of the distal fallopian tube.

Surprisingly, we uncovered four atypical tissue regions that were missed by histopathological examination, likely due to p53 staining negativity (p53 null phenotype) and ambiguous Ki67 status. However, our deep proteome data clearly marked these regions as atypical with a strong upregulation of DNA replication, DNA damage response, and cell cycle proteins, including CDKN2A (p16), which is an additional marker used to diagnose STILs and STICs (Novak et al, 2015). Morphological and immunohistochemical reevaluation by p16 confirmed our suspicion and allowed us to reclassify them as precancerous lesions. In accordance with the current pathological nomenclature, these regions were STILs. Unbiased spatial proteomics data hence show great potential for companion diagnostics, especially in cases difficult to diagnose by morphology and IHC alone, or when lacking disease-specific histological markers

Many of the identified proteins upregulated in STICs and invasive carcinoma have known roles in cell cycle regulation, DNA metabolism, and oncogenic signaling, characteristic of this highly proliferative and genetically unstable cancer. We found that metabolic reprogramming is a common and likely early event in HGSOC pathogenesis, identifying several onco-metabolic changes

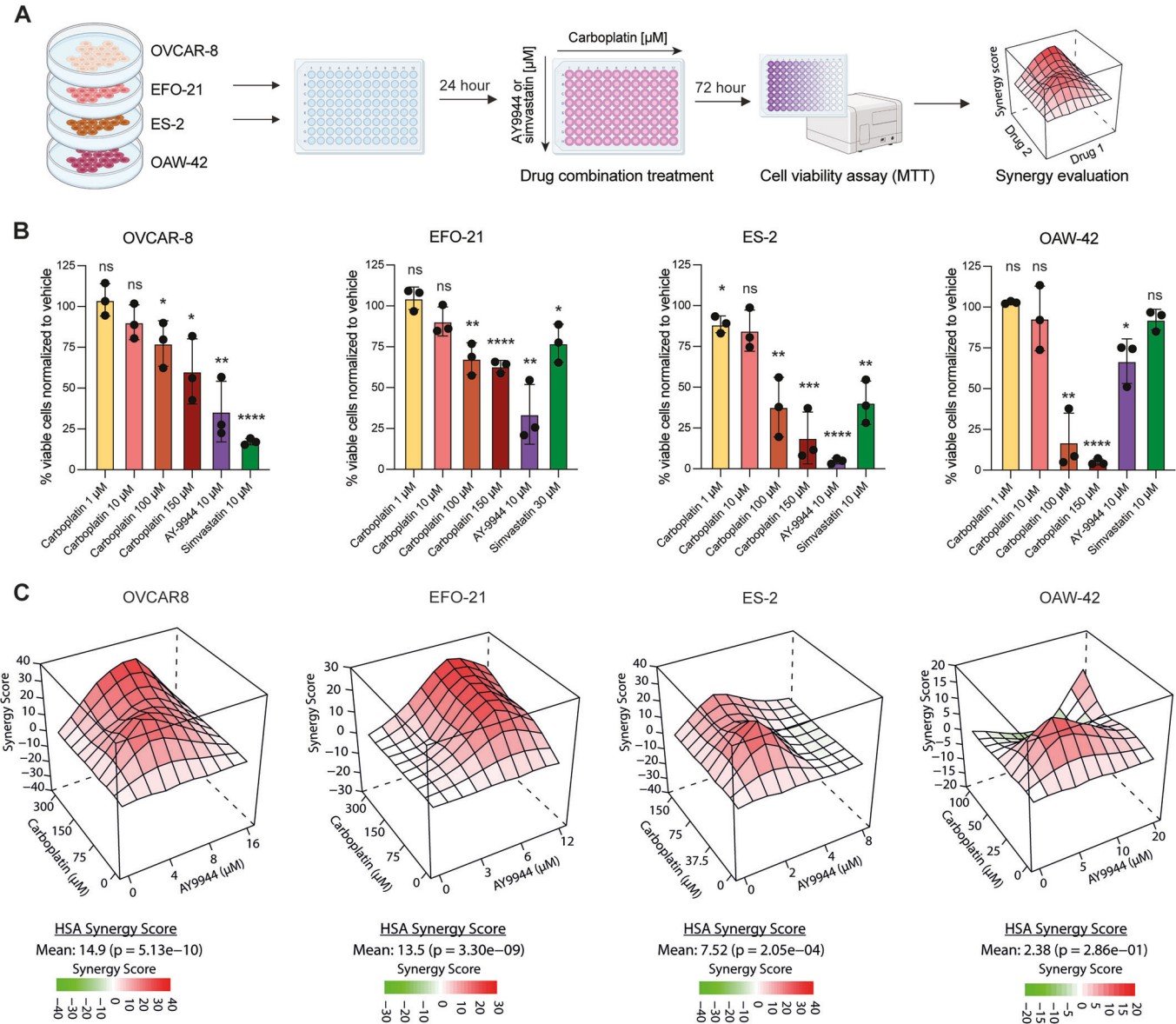

**Figure 7. Cholesterol biosynthesis inhibition sensitizes ovarian cancer cells to carboplatin.**

(A) Experimental workflow describing cholesterol biosynthesis inhibitor (AY9944 or simvastatin) treatment of ovarian cancer lines in combination with carboplatin. Cancer cells were treated for 72 h with AY9944 or simvastatin in combination with carboplatin. Cell viability was measured by MTT assay and normalized to untreated controls. Drug synergy was evaluated by Highest Single Agent (HSA) method, where synergy value >10 indicates synergistic effect, −10 to 10 indicates additive or no interaction, <−10 indicates antagonistic effect. (B) Cell viability assay of ovarian cancer cell lines (OAW-42, OVCAR-8, ES-2, and EFO-21) treated with different doses of carboplatin, the DHCR7 inhibitor AY9944 and simvastatin. Treatment effect relative to matched vehicle. Experiments were performed in biological triplicates (with $n = 3$ technical replicates each). Mean ± SD; significance test: unpaired Student's $t$-test: ns (not significant), $*p < 0.05$, $**p < 0.01$, $***p < 0.001$, $****p < 0.0001$. Exact $p$ values are provided in the Appendix Table S5. (C) Synergy score map of OVCAR-8, EFO-21, ES-2, and OAW-42 cells treated with different concentrations of carboplatin and the DHCR7 inhibitor AY9944. The mean HSA (Highest Single Agent) score for OVCAR-8, EFO-21, ES-2, and OAW-42 was 14.9 ($p$ value $= 5.13e^{-10}$), 13.5 ($p$ value $= 3.30e-09$), 7.52 ($p$ value $= 2.05e-3$), 2.38 ($p$ value $= 2.86e-01$) synergy value >10 indicates a synergistic effect, −10 to 10 indicates an additive or no interaction, and <−10 indicates an antagonistic effect.

for therapeutic intervention and possibly early detection. For example, our data revealed that oxidative glucose metabolism gradually decreases from healthy fallopian tube epithelial cells, over STICs, to invasive tumors, accompanied by upregulated glycolysis and lipid metabolism (cholesterol biosynthesis/mevalonate pathways). While the metabolic switch towards a glycolytic state is a well-known characteristic of many cancers (Hanahan and

Weinberg, 2011), upregulation of cholesterol biosynthesis was highly intriguing. Studies in breast and liver cancer showed a link between *TP53* mutant tumors and increased cholesterol biosynthesis (Freed-Pastor *et al*, 2012; Moon et al, 2018), likely explaining this signature's dominance in our *TP53* mutant cohort. Importantly, this metabolic phenotype was independent of molecular subtype, supporting the view that blocking cholesterol biosynthesis

(He et al, 2021), possibly via the upregulated enzymes DHCR7 or DHCR24, could be a promising therapeutic strategy for high-grade serous ovarian cancer. Our in vitro data based on several HGSOC models established that inhibiting the terminal steps of cholesterol biosynthesis via DHCR7 inhibition not only shows dose-dependent tumor cell killing, but also synergizes with platinum-based chemotherapy. Our data suggest a possible metabolic dependency of HGSOC on altered lipid and cholesterol metabolism with prospects for combination treatments to improve therapeutic outcomes and possibly overcome frequent chemoresistance in recurrent tumors. Notably, a previous study linked the inhibition of the cholesterol synthesis pathway by statins to decreased STIC formation in mouse models of ovarian cancer (Kobayashi et al, 2015), further emphasizing the translational relevance of our findings. Lastly, these data could also have implications to diagnose HGSOC at an early stage. For example, altered cholesterol/lipid profiles could be measured in pelvic fluid (Tang et al, 2024). Alternatively, the detection of endogenous plasma peptides, such as reported for DHCR24 (Dufresne et al, 2018), or tumor-derived extracellular vesicles (Hinestrosa et al, 2022; Trinidad et al, 2023) via HGSOC-specific cell surface proteins, as discovered here, could offer a sensitive approach for early HGSOC detection. However, more preclinical research is needed to assess the diagnostic relevance of our findings.

We further demonstrated that STICs and invasive carcinomas show high proteomic similarity, lacking a distinct STIC proteotype distinguishable from advanced carcinomas. Our results showed that STICs mirror the phenotypic and molecular subtype heterogeneity of ICs, implying they exhibit the full oncogenic potential of invasive HGSOC. Two recent smaller-scale studies (Wisztorski et al, 2023), including our own (Eckert et al, 2019), support this view. One explanation is that at HGSOC diagnosis, STICs have likely undergone additional molecular aberrations, making them indistinguishable from invasive tumors. The estimated time between STIC development and HGSOC progression is seven years (Labidi-Galy et al, 2017), possibly leading to further molecular adaptations. Another explanation for the high proteomic similarity between STICs and ICs is that STICs do not represent precursor lesions but advanced tumors that retrogradely metastasize to the fallopian tube. However, given the lower frequency of this scenario (Eckert et al, 2016; Labidi-Galy et al, 2017), this hypothesis doesn't explain the strong coherence of our proteomic data for the 32 matching STIC-invasive tumor pairs.

To further explore the proteomic heterogeneity of STICs, we integrated previously identified molecular subtypes from bulk transcriptomics and proteomics data (Bell et al, 2011; Zhang et al, 2016), assigning nearly half of the epithelial samples to one predominant subtype. The inability to unambiguously assign all samples to one subtype might reflect HGSOC's polyclonal nature and the co-existence of diverse tumor subpopulations with distinct gene expression programs in different spatial niches (Izar et al, 2020; Geistlinger et al, 2021; Denisenko et al, 2024). Among the clearly assigned samples, we classified half as the proliferative subtype and half as immune-enriched, with a likely underlying differentiated tumor cell phenotype (Geistlinger et al, 2021). While the clinical significance of HGSOC molecular subtypes remains debatable, our global proteomics data suggest a simple binary classification into two distinct epithelial proteotypes with therapeutic implications. Cluster 1 tumors showed an inflamed,

immune-enriched signature with elevated interferon and TNF signaling, aligning with a recent spatial omics report (Kader et al, 2024) and high lymphocyte infiltration. Our findings generally agree with Kader et al's conclusion that interferon signaling from epithelial cells is an early occurrence in HGSOC development, but indicate this inflammatory characteristic is present in approximately half of all STICs. Proliferative cluster 2 samples showed low interferon-related signaling, were immune-deserted, and linked to higher patient age and likely more unfavorable clinical outcomes. This dichotomy is supported by long-known phenotypic differences in tumor-infiltrating CD3 + T cells, a favorable prognostic factor for roughly half of HGSOC patients (Zhang et al, 2003). Our precise sampling through laser microdissection enabled us to disentangle distant from intraepithelial immune cells, where only the latter mark tumors likely to respond to immune checkpoint inhibitors (ICI) (Ghisoni et al, 2024). Notably, a comparable IFN-high immunophenotype (similar to cluster 1 tumors) associated with an upregulation of antigen presentation and intraepithelial immune cells was recently identified as a characteristic of metastatic deficient mismatch repair colorectal cancers responding to ICI treatment (Kim et al, 2012). Therefore, our deep proteomics data could serve as a valuable resource to build new classifiers for improved patient stratification beyond PD-L1 assessment. This is clinically relevant since ICI treatment in ovarian cancer has yielded disappointing results (Ghisoni et al, 2024), likely due to the absence of clear guidelines to tailor treatment to patients with the highest likelihood of therapeutic benefit.

The absence of an epithelial STIC proteotype distinct from concurrent invasive tumors contrasts with our stromal findings, which revealed a disease gradient of stromal remodeling from NFT stroma over STIC to invasive stroma. Stromal remodeling of STIC precursors featured both NFT and invasive phenotypes, possibly reflecting different stages of progressive tumorigenic ECM remodeling. This suggests some STICs, despite high epithelial proteome similarity to invasive tumors, have not undergone complete stromal transformation. As all STICs in our cohort were accompanied by adjacent invasive tumors, it raises questions about the timescales of TME remodeling, clonal escape, and tissue invasion. While the presence or absence of stromal immune cell infiltration (e.g., CD8 + T cells) was strongly associated with the epithelial tumor proteotype (i.e., immune-enriched/differentiated [cluster 1] versus proliferative [cluster 2]), oncogenic ECM remodeling was a more uniform feature in our cohort. We found that one-third of the stromal proteome undergoes quantitative remodeling towards an immunosuppressive TME, likely driven by myofibroblasts and M2-like macrophages. These changes include many known ECM degraders (cathepsins and MMPs), pro-inflammatory cytokines (TGFβ1, TGFβ2, IL16, and IL18), and many structural core matrisome proteins. On this basis, we extracted a 100-protein stromal signature of commonly upregulated and downregulated proteins. This signature features known disease drivers and oncogenic ECM proteins associated with ovarian cancer progression and metastasis (e.g., NNMT, CD163, and FN1) (Eckert et al, 2019; Mitra et al, 2011; Lecker et al, 2021) and sheds light on proteins with high therapeutic potential. For example, thymidine phosphorylase (TYMP), a therapeutic target in metastatic colorectal cancer (Prager et al, 2023), showed a strong and gradual increase in the invasive stroma. Similarly, SYK, a non-receptor tyrosine kinase implicated in immune cell regulation (Liu and Mamorska-Dyga,

2017), is targeted by Fostamatinib, which was approved by the FDA for the treatment of chronic immune thrombocytopenia (Newland et al, 2018). Notably, SYK also represents a promising strategy to target TAMs in pancreatic cancer (Rohila et al, 2023), cells we found to be strongly enriched in the invasive stroma. Together, these data underscore the central role of the stromal compartment in HGSOC pathogenesis, the importance of tumor-stromal co-evolution, and the potential of the TME as a therapeutic target.

In summary, our study highlights the power of spatially and cell-type resolved proteomics to dissect the molecular underpinnings of early carcinogenesis and provides a rich proteomic resource for biomarker and drug target discovery.

# Methods

### Reagents and tools table

| Reagent/resource | Reference or source | Identifier or catalog number |
|---|---|---|
| **Antibodies** | | |
| p53 (IHC) | Dako/Agilent | M7001 RRID:AB_2206626 |
| PAX8 (IHC) | Roche/Ventana | 760-4618 |
| Ki67 (IHC) | Dako/Agilent | M7240 RRID:AB_2142367 |
| p16 (IHC) | Roche/Ventana | 805-4713 RRID:AB_3675558 |
| DHCR7 (IHC) | Atlas Antibodies | HPA044280 RRID:AB_10794893 |
| DHCR24 (IHC) | Cell Signaling | 2033 RRID:AB_2091448 |
| Ki67 | Cell Signaling | 11882 RRID:AB_2687824 |
| anti-mouse | Invitrogen | A32794 RRID:AB_2536180 |
| CD4 | Cell Signaling | 40568S RRID: AB_3492108 |
| CD8 | Thermo Fisher | 53-0008-82 RRID: AB_2574413 |
| FOXP3 | BioLegend | 320107 RRID: AB_492986 |
| CD11c | BioTechne | NBP2-54432AF750 RRID: AB_3083691 |
| CD163 | Abcam | ab218293 RRID: AB_2889155 |
| panCK | Thermo Fisher | 41-9003-82 RRID: AB_11218704 |
| PAX8 | Proteintech | CL647-10336 RRID: AB_2920213 |
| COL1A1 | BioTechne | NB600-408AF750 RRID: AB_10000511 |
| **Chemicals, Enzymes and other reagents** | | |
| n-Dodecyl-beta-Maltoside (DDM) | Sigma-Aldrich | D4641-500MG |
| Endoproteinase LysC | Promega | VA1170 |

| Reagent/resource | Reference or source | Identifier or catalog number |
|---|---|---|
| Proteomics grade modified trypsin | Promega | V5117 |
| Tris(2-carboxyethyl) phosphine hydrochloride | Sigma-Aldrich | C4706-2G |
| Acetonitrile (ACN) HPLC-grade | VWR | 83640.290 |
| Isopropanol (ISO) | Sigma-Aldrich | 1070222511 |
| Formic acid | Merck | F0507-100ML |
| 2-chloroacetamide | Sigma-Aldrich | C0267-100G |
| DMEM | Gibco | 21885-025 |
| Fetal bovine serum (FBS) | Capricorn | FBS-16A |
| PCR mycoplasma kit | Biontex | M030/050 |
| AY-9944 | MedChemExpress | HY-107420 |
| Simvastatin | Biomol | Cay10010344-5 |
| Carboplatin | Merck | C2538 |
| MTT assay reagent | Merck | M2128 |
| Sodium hydroxide solution | Merck | 72068 |
| 4.5% hydrogen peroxide solution | Merck | 31642 |
| TEAB | Merck | T7408-100ML |
| Trifluoroacetic acid | Sigma-Aldrich | 96924-250ML-F |
| Microscopy Neo-Clear | Merck | 1.09843.5000 |
| BSA | Serva | 11948.01 |
| LC-MS grade water | Avantor | 9825.2500GL |
| Acetonitrile with 0.1% formic acid | Sigma-Aldrich | 1590021000 |
| Water with 0.1% Formic Acid | Fisher Scientific | 10188164 |
| 10% fetal bovine serum | Capricorn | FBS-16A |
| PCR mycoplasma kit | Biontex | M030/050 |
| DMSO cell culture grade | Genaxxon | M6323.0250 |
| EnVision FLEX Target Retrieval Solution High pH (50X) | Agilent Dako | K8004 |
| Hoechst 33342 staining reagent | Thermo Fsher | 62249 |
| **Software** | | |
| QuPath | https://qupath.github.io | Version 0.4.3 |
| Zeiss ZEN | Carl Zeiss AG | Version 3.5 |
| DIA-NN | Demichev et al https://github.com/vdemichev/DiaNN | Version 1.8.1 |
| Perseus | Tyanova et al https://maxquant.net/perseus/ | Version 1.6.15.0 |

| Reagent/resource | Reference or source | Identifier or catalog number |
|---|---|---|
| Leica Laser Microdissection software | Leica Microsystems | Version 8.3.0.08259 |
| R | The R Project for Statistical Computing. https://posit.co/download/rstudio-desktop/ | Version 4.3.2 |
| SEQUENCE Pilot | JSI Medical Systems GmbH | Version 5.4.0 |
| Qupath_to_LMD function: Contour export from Qupath to the Leica LMD7 | https://github.com/CosciaLab/Qupath_to_LMD. | https://doi.org/10.5281/zenodo.8414787 |
| Bruker Compass Data Analysis Software version 6.0 | Bruker Daltonik GmbH | https://www.bruker.com/en/products-and-solutions/mass-spectrometry/ms-software.html |
| **Other** | | |
| C18 Evotips (Evotip Pure, Evosep) | Evosep Biosystems | EV2013 |
| 96-well plate | Thermo Fisher Scientific | AB1300 |
| 384-well low-binding plate | Eppendorf | 0030129547 |
| Super PAP-pen liquid blocker mini | Science Services | N71312-N |
| Cover glass | Corning | CLS2980223 |
| PPS frame slides | Leica | 11600294 |
| High-volume diaphragm chips for MANTIS Liquid Dispenser | Formulatrix | 233128 |
| Super PAP-pen liquid blocker mini | Science Services | N71312-N |
| Evaporative concentrator: Eppendorf Vacuum Concentrator Plus with 96-well plate rotor | https://www.eppendorf.com/de-de/ | N/A |
| MANTIS Liquid Dispenser (Formulatrix, V3.3 ACC RFID, software version 4.7.5) | https://formulatrix.com/liquid-handling-systems/mantis-liquid-dispenser/ | N/A |
| PCR ComfortLid (Hamilton) | https://www.hamiltoncompany.com/ | N/A |
| Evosep One | Evosep | EV-1000 |
| timsTOF Ultra | Bruker Daltonik GmbH | N/A |
| Evosep Performance column | Evosep | EV1137 |
| BenchMark XT immunostainer | Ventana Medical Systems | N/A |
| PANNORAMIC 1000 digital slide scanner | 3DHISTECH | N/A |

## Sample collection and patient cohort

We retrieved all samples containing the term "STIC" and/or "serous tubal intraepithelial carcinoma" in the pathology report from the pathology archive of the Institute of Pathology, Charité, between 2013 and 2022. Two pathologists (M.P.D. and E.T.T.) performed two independent rounds of reclassification of the precursor lesions according to the criteria proposed by Vang et al (Vang et al, 2012), using three whole slides stained with H&E, p53, and Ki67. Briefly, STIC was defined by a lesion morphologically suspicious or unequivocal for STIC and an immunohistochemical p53 aberrant expression and a proliferation rate (Ki67) >10%. Only samples in which full agreement was reached were included in the proteomic analysis. In addition, we excluded all samples that received chemotherapy prior to the resection of tubal lesions. Clinical data were obtained from the Tumor Bank Ovarian Cancer Network (www.toc-network.de) or the Charité Comprehensive Cancer Center (https://cccc.charite.de). This study was approved by the local ethics committee (EA1/110/22).

## Cell line models

The human ovarian carcinoma cell line OAW-42 was obtained from the European Collection of Animal Cell Cultures (Salisbury, United Kingdom). OVCAR-8 was obtained from the laboratory of Ernst Lengyel (Department of Obstetrics and Gynecology/Section of Gynecologic Oncology, University of Chicago, Chicago, IL, USA). ES-2 was obtained from the American Type Culture Collection, and EFO-21 was obtained from Dr. Fritz Hölzel (Department of Gynecology, University Hospital Eppendorf, Hamburg, Germany). The cells were cultured in DMEM (Gibco, #21885-025), all supplemented with 10% fetal bovine serum (Capricorn, #FBS-16A), no added antibiotics, at 37 °C with 5% $CO_2$ and 95% humidity. Prior to the study, the cytogenetic analysis and cell authentication of the cells were performed at the DNA-Fingerprinting Facility at Charité Berlin using short tandem repeat DNA. All cell lines were tested for mycoplasma contamination using the PCR mycoplasma kit (Biontex, #M030/050).

## Cell viability assay

Cell viability assay was performed in 96-well plates. For OAW-42, OVCAR-8, and ES-2, we plated 4000 cells/well, and for EFO-21, we plated 6000 cells/well and we cultured the cells in full growth medium for 24 h. The medium was then removed and replaced with new full medium containing different concentrations of AY-9944 (MedChemExpress, #HY-107420), carboplatin (Merck, #C2538) or combinations of the two drugs. The plates were incubated at 37 °C, 5% $CO_2$ for 72 h. After the treatment, 11 μL of MTT assay reagent (Merck, #M2128) was added to the medium, and incubated at 37 °C, 5% $CO_2$ for 4 h. Following MTT incubation, the media and MTT were removed, 100 μL of DMSO (Genaxxon, M6323.0250) was added to all wells, including controls and the spectrophotometric absorbance of the samples was detected by using a microplate spectrophotometer (BioTek, Synergy 2) at 540 nm wavelength.

**Table 1.** Antibodies used for immunohistochemistry (IHC) imaging.

| Antibody | Company | Catalog # | Clone | Species | Dilution | Pre-preparation |
|---|---|---|---|---|---|---|
| p53 | Dako/Agilent | M7001 | DO-7 | Mouse | 1:50 | CC1 standard |
| PAX8 | Roche/Ventana | 760-4618 | MRQ-50 | Mouse | ready to use | CC1 mild |
| Ki67 | Dako/Agilent | M7240 | MIB-1 | Mouse | 1:50 | CC1 mild |
| p16 | Roche/Ventana | 805-4713 | E6H4 | Mouse | 1:2 | CC1 mild |
| DHCR7 | Atlas Antibodies | HPA044280 | - | Rabbit | 1:100 | CC1 standard |
| DHCR24 | Cell Signaling | 2033 | C59D8 | Rabbit | 1:100 | CC1 standard |
| Goat Anti-Rabbit IgG Antibody (H + L) | Vector | BA-1000-1.5 | - | Goat | 1:200 | - |

## Homologous repair deficiency analysis

HRD analysis was performed using the (Northeastern German Society for Gynecologic Oncology) NOGGO GIS v1 Assay, as previously described (Willing et al, 2023). Briefly, from tumor rich regions of invasive carcinoma from ten to twenty sequential 5 μm thick FFPE slides DNA was extracted. 50 to 100 ng of tumor DNA was used for library preparation using hybrid capture XT HS2 chemistry (Agilent Technologies) targeting all exonic bases as well as a minimum of 10 bp flanking region of 57 genes, including 35 HRR genes, as well as selected driver genes and more than 20,000 genome-wide evenly distributed single nucleotide polymorphism (SNP) loci that enable the detection of allele-specific copy number alterations (CNA). Next, the libraries were subjected to sequencing with a minimum of 20 million reads using 100 bp read length on a NextSeq 2000 instrument (Illumina, San Diego, CA, USA). The genomic instability score (GIS) was calculated using an allele-specific copy number profile and three measures of HRD based on the PureCN output: percent loss of heterozygosity (PLOH), percent copy number alteration (PCNA), and percent telomeric copy number alteration (PTCNA). A GIS cutoff of "83" or the presence of a pathogenic mutation in either *BRCA1* or *BRCA2* was used to define HRD-positive cases. Mutation calling was conducted using SEQUENCE Pilot Software, Version 5.4.0 (JSI Medical Systems GmbH, Ettenheim, Germany).

## Immunohistochemistry (IHC)

Immunohistochemical staining was performed on a BenchMark XT immunostainer (Ventana Medical Systems, Tucson, AZ, USA). For antigen retrieval, sections were incubated in CC1 mild/standard buffer (Ventana Medical Systems, Tucson, AZ, USA) for 30 min at 100 °C. The sections were stained with anti-Ki67 antibody (M7240, Dako, 1:50, CC1 mild buffer), anti-p53 (M7001, Dako, 1:50, CC1 standard buffer), anti-PAX8 (760-4618, Roche/Ventana, ready to use, CC1 mild buffer), anti-p16 (805-4713, Roche/Ventana, 1:2, CC1 mild buffer), anti-DHCR7 (HPA044280, Atlas Antibodies, 1:100, CC1 standard buffer), and anti-DHCR24 (2033, Cell Signaling, 1:100, CC1 standard buffer) for 60 min at room temperature, and visualized using the avidin–biotin complex method and DAB (Table 1). We stained the cell nuclei by additionally incubating for 12 min with hematoxylin and bluing reagent (Ventana Medical Systems, Tucson, AZ, USA). Histological images were acquired with a PANNORAMIC 1000 digital slide scanner (3DHISTECH).

For DHCR7 and DHCR24 scoring four intensity levels were used: 0 (no positivity), 1+ (weak positivity), 2+ (moderate

positivity), and 3+ (strong positivity). For each category, the percentage of cells exhibiting that staining intensity was determined. Then we applied the H-score formula $= (1 \times \%$ of cells with weak positivity $(1+)) + (2 \times \%$ of cells with moderate positivity $(2+)) + (3 \times \%$ of cells with strong positivity $(3+))$. The H-score ranged from 0 to 300, with a score of 0 indicating no marker expression and a score of 300 indicating strong staining in all cells.

## Cyclic immunofluorescence (CyCIF) staining and imaging

Prior to immunostaining, tissue sections were incubated at 60 °C for 30 min, followed by deparaffinization and sequential rehydration as follows: two 5-min immersions in neo-clear buffer, two 2-min washes in 99% ethanol, and sequential 2-min washes in 80%, 70% ethanol, and 1x PBS (twice for 1 min each). Heat-mediated antigen retrieval was performed in Tris-EDTA (pH 9) using a steamer for 25 min, followed by cooling to room temperature in the retrieval solution. Slides were then washed three times in 1x PBS. To reduce tissue autofluorescence, sections were pre-bleached for 30 min under direct white light in 4.5% $H_2O_2$ and 24 mM NaOH diluted in 1x PBS. Following three additional PBS washes, tissue sections were outlined with a PAP-pen (Science Services, N71312-N) to minimize the reaction volume and blocked with 3% BSA (Serva, 11948.01) in 1x PBS for 30 min at room temperature. Antibody incubation was performed overnight at 4 °C in a blocking buffer in a humidified staining chamber. Immunofluorescence staining was conducted over several cycles. Except for p53, all antibodies were directly conjugated (Table 2). For p53 staining, sections were washed in 1x PBS after primary antibody incubation and then incubated with a secondary fluorescently conjugated antibody (A555 donkey anti-mouse, 1:500) at room temperature for 1 h. Slides were subsequently washed in 1x PBS, counterstained with Hoechst 33342 (1 μg/mL) for 5 min at room temperature, washed again, mounted with 10% glycerol in 1x PBS and imaged. Following imaging, coverslips were removed by soaking the slides in 1x PBS within a vertical Coplin jar on a platform shaker. Detached slides were washed in 1x PBS, bleached for 30 min under direct white light in 4.5% $H_2O_2$ and 24 mM NaOH diluted in 1x PBS, and washed again in 1x PBS before proceeding to the next round of primary antibody staining. This process was repeated twice to achieve staining for ten markers. After the final cycle, coverslips were removed by soaking the slides in 1x PBS within a vertical Coplin jar on a platform shaker. Once detached, the slides were rinsed with Milli-Q water, air-dried, and stored at 4 °C. Imaging was conducted using a Zeiss Axioscan 7 slide scanner equipped with the Colibri 7 LED light source and an EC Plan-Neofluar 20x/0.50 M27 objective at 2 × 2 binning. Stitching of the raw images

**Table 2. Antibodies used for immunofluorescence (IF) imaging.**

| Target | Dilution | Fluorophore | Source | Catalog no. | Clone | Research resource identifier (RRID) |
|---|---|---|---|---|---|---|
| Ki67 | 1:100 | Alexa Fluor 488 | Cell Signaling Technology | 11882 | D3B5 | AB_2687824 |
| p53 | 1:25 | - | Agilent Dako | M7001 | - | AB_2206626 |
| anti-mouse | 1:500 | Alexa Fluor 555 | Invitrogen | A31570 | - | AB_2536180 |
| CD4 | 1:50 | Alexa Fluor 647 | Cell Signaling Technology | 40568S | MSVA-004R | AB_3492108 |
| CD8 | 1:100 | Alexa Fluor 488 | Thermo Fisher | 53-0008-82 | AMC908 | AB_2574413 |
| FOXP3 | 1:50 | PE | Biolegend | 320107 | 206D | AB_492986 |
| CD11c | 1:50 | Alexa Fluor 750 | BioTechne | NBP2-54432AF750 | ITGAX/1243 | AB_3083691 |
| CD163 | 1:50 | Alexa Fluor 488 | Abcam | ab218293 | EPR14643-36 | AB_2889155 |
| panCK | 1:100 | eFluor™ 570 | Thermo Fisher | 41-9003-82 | AE1/AE3 | AB_11218704 |
| PAX8 | 1:50 | CoraLite 647 | Proteintech | CL647-10336 | - | AB_2920213 |
| COL1A1 | 1:50 | Alexa Fluor 750 | BioTechne | NB600-408AF750 | - | AB_10000511 |

was performed using ZEN software (version 3.5, Blue Edition), with the DAPI channel set as the reference for all channels and the following parameters: minimal overlap of 5%, maximal shift of 15%, Comparer set to Optimized, and Global Optimizer set to Best.

## Image analysis and contour export for laser microdissection

QuPath (version 0.4.3) was utilized for conducting image analysis. Regions of interest were manually annotated in QuPath following the image analysis process. To ensure accurate contour transfer between the screening and laser microdissection microscopes, three tissue reference points (x-y coordinates) were selected. The contours and reference points were then exported in GeoJSON format and converted into.XML format, which is compatible with Leica LMD7 software. The shape processing code can be accessed at github.com/CosciaLab/Qupath_to_LMD, and it employs geopandas (Version 0.12.2) and py-lmd (Makhmut et al, 2023) (Version 1.0.0).

## Laser microdissection

For laser microdissection, three slides were prepared for each sample and mounted on PPS-Membrane Frame Slides (Leica). The slides were then stained immunohistochemically with p53 (5-µm thick slide), PAX8 (2.5-µm thick slide), and Ki67 (2.5 µm thick slide). Whenever feasible, the specimen was collected from the p53-stained slide, while the other slides served mainly for orientation. Sample annotation was conducted under the direct supervision of at least one of the study pathologists (M.P.D./E.T.T.).

For laser microdissection-based tissue collection, we employed the Leica LMD7 system along with Leica Laser Microdissection V 8.3.0.08259 software. The tissue was cut using a 20x objective in either brightfield or fluorescence mode. We applied the following laser parameters for the 20x objective (HC PL FL L 20x/0.40 CORR): 56 power, 1 aperture, 15 speed, 1 middle pulse count, −1 final pulse, 37 – 45% head current (adjusted based on tissue type and section thickness), 801 pulse frequency, and 101 offset. The cut contours were then collected into a low-binding 384-well plate (Eppendorf 0030129547), which was set up using the 'universal holder' function with a single empty well separating each sample.

## Sample preparation for MS-based proteomic analysis

Automated cutting was employed to gather tissue specimens following contour import into 384-well plates (Eppendorf, 0030129547) with low-binding properties. To ensure tissue settled at the well bottoms post-LMD collection, each well received 15 µL of acetonitrile, underwent brief vortexing, and was vacuum dried (15 min at 60 °C). A subsequent well inspection was conducted prior to proteomics sample preparation to verify successful collection. The DDM-based protocol utilized a lysis buffer comprising 0.025% DDM, 5 mM TCEP, 20 mM CAA, and 0.1 M TEAB in water. A MANTIS Liquid Dispenser with high-volume diaphragm chips was used to add 4 µl of lysis buffer to each sample well. The plate was sealed with PCR ComfortLid and heated at 95 °C for 60 min. After brief cooling, 1 µL of LysC (10 ng/ µL in 0.1 M TEAB [pH 8.5] and 30% ACN in milli-Q water) was introduced, followed by digestion for at least 2 h at 37 °C in a thermal cycler (50 °C lid temperature). Next, 1 µL of trypsin (10 ng/µl containing 10% ACN and 0.1 M TEAB [pH 8.5] in milli-Q water) was added, with overnight incubation at 37 °C in the thermal cycler. The following day, digestion was stopped by the addition of trifluoroacetic acid (TFA, final concentration 1% v/v), and samples underwent vacuum drying before peptide clean-up.

## Peptide clean-up with C18 tips

Peptide purification was conducted using Evotips (Evosep, Odense, Denmark) following the manufacturer's instructions. The tips were prepared by adding 20 µl of buffer B (99.9% ACN, 0.1% FA) to each C18 tip (EV2013, Evotip Pure, Evosep) followed by centrifugation at 700 rpm for 1 min. Subsequently, 20 µl of buffer A (99.9% water, 0.1% FA) was added to the top of each C18 tip, which was then activated in isopropanol for 20 s and centrifuged again at 700 rpm for 1 min. Peptides were then applied to the Evotips, washed once with 20 µL of buffer A, and finally loaded with 200 µL of buffer A. The tips were also submerged at the bottom in buffer A before initiating the LC-MS analysis.

## MS-based proteomic analysis

Liquid chromatography was performed using the Evosep One LC system (Evosep, Odense, Denmark) connected to a trapped ion

mobility spectrometer with quadrupole time-of-flight mass spectrometer (timsTOF Ultra, Bruker Daltonik, Bremen, Germany) with a nano-electrospray ion source (CaptiveSpray, Bruker Daltonik, Bremen, Germany). Digested peptides were loaded on the Evosep Performance column (EV1137, 150 μm inner diameter, packed with 1.5 μm C18 beads) at 40 °C. Chromatographic separation was performed using an Evosep 30SPD gradient. The solvents utilized were LC-MS-grade water containing 0.1% formic acid (buffer A) and acetonitrile with 0.1% formic acid (buffer B). For the dia-PASEF analysis, we employed a method comprising eight dia-PASEF scans divided into three ion mobility windows per scan. This covered a mass-to-charge ratio range of 400–1000 m/z using 25 Th windows and an ion mobility range from 0.64 to 1.37 Vs cm$^{-2}$. The mass spectrometer was run in high-sensitivity mode, with accumulation and ramp time set to 100 ms, and capillary voltage at 1750 V. The collision energy was configured as a linear ramp, starting at 20 eV at $1/k_0 = 0.6$ Vs cm$^{-2}$ and increasing to 59 eV at $1/k_0 = 1.6$ Vs cm$^{-2}$. This collision energy ramp was applied linearly as a function of ion mobility, decreasing from 59 eV at $1/k_0 = 1.6$ Vs cm$^{-2}$ to 20 eV at $1/k_0 = 0.6$ Vs cm$^{-2}$.

## Mass spectrometry raw file analysis

For the analysis of dia-PASEF raw files and generation of spectral libraries, we employed DIA-NN (Demichev et al, 2020) (version 1.8.1). The human FASTA file was obtained from the UniProt database (2022 release, UP000005640_9606, downloaded on April 8th 2022). To generate in silico predicted libraries, we provided the human FASTA file along with commonly encountered contaminants (Frankenfield et al, 2022). We enabled deep learning-based predictions for spectra, RTs, and IMs within the 300-1200 m/z mass range. Fixed modifications included N-terminal M excision and cysteine carbamidomethylation. We allowed up to two missed cleavages and set the precursor charge to 2–4. DIA-NN was run in default mode with slight modifications. We configured MS1 and MS2 accuracies to 15.0, set scan windows to 0 (for DIA-NN assignment), enabled isotopologues, activated match-between-runs, applied heuristic protein inference, and disallowed shared spectra. Protein inference was based on genes. We set the neural network classifier to single-pass mode and selected "Robust LC (high precision)" as the quantification strategy. Cross-run normalization was configured as "RT-dependent," library generation as "smart profiling," and speed and RAM usage as "optimal results."

## Data analysis and statistics

Proteomic data analysis was performed within the R environment (https://r-project.org/version 4.3.2) with the following packages: tidyverse (version 2.0.0), Rstatix (version 0.7.2), UpSetR package (version 1.4.0), FactoMineR (version 2.11), factoextra (version 1.0.7), ggpubr (version 0.6.0), corrplot (version 0.92), Complex-Heatmap (version 2.18.0), RColorBrewer (version 1.1.3), circlize (version 0.4.16), dendsort (version 0.3.4), ggcorrplot (version 0.1.4.1), ggplot2 (version 3.5.1), ggrepel (version 0.9.5), viridis (0.6.5) and networkD3 (version 0.4). Perseus (Tyanova et al, 2016) (version 1.6.15.0) and Instant Clue (version 0.12.2) (Nolte et al, 2018) were used for additional exploratory data analysis. Prior to statistical testing, data were first filtered to keep only proteins with 70% non-missing values in at least one group, or more stringently,

with 70% in all groups. Missing values were imputed based on a normal distribution (width = 0.3, downshift = 1.8). Pairwise comparisons were computed using a two-sided Student's $t$-test. For multiple group comparisons, analysis of variance (ANOVA) was used. For both tests, a permutation-based FDR of 5% was applied to correct the multiple hypothesis testing. Pathway enrichment analysis was performed in Perseus based on Fisher's exact test (categorical data) or 1D pathway enrichment analysis (Cox and Mann, 2012). Hallmark gene sets, WikiPathways, and the Reactome pathways were enriched terms filtered using a Benjamini–Hochberg FDR cut-off of 0.05. The minimum category size was set to five. The annotation matrix algorithm of Perseus (version 1.5.0.3) was additionally employed to globally compare pathway-level changes across multiple samples (Fig. EV3B). This algorithm tests the difference in any protein annotation from the overall intensity distribution of the sample. The resulting pathway scores were z-scored for relative comparison.

For subtype classification of high-grade serous ovarian cancers based on TCGA subtypes, the consensusOV package (version 1.26.0, get.consensus.subtypes function) was used. The package calculates the probability for each subtype and assigns each sample to the subtype with the highest probability. To characterize the previously identified fallopian tube epithelial cell types in the dataset, we used the signature matrix from Dinh et al (Dinh et al, 2021). For the systematic analysis of ECM changes, the matrisome database described by Renner et al (Renner et al, 2022) was used. To prioritize proteins upregulated in the invasive stroma during HGSOC progression (Fig. 5), we employed support vector machine (SVM) classification using the Perseus software. Before classification, data were first filtered (70% in at least one group), log2-transformed, and imputed based on a normal distribution (width = 0.3, downshift = 1.8). The SVM was then trained on the stromal proteomes to determine the optimal classification parameters. To achieve the lowest classification error, we systematically tested kernels and the corresponding parameters. We used a Radial basis function (RBF) kernel, parameter sigma (σ) = 5 and parameter C = 10. For cross-validation, the "leave-one-out" function was used, and ANOVA was used as the feature ranking method with an $s_0$ value (Goss Tusher et al) of 0.1. The optimal number of discriminating protein features was then determined from the error percentage curve, which resulted in an error rate of 5% for the top 100 features.

For single-cell RNAseq analysis of HGSOC stromal cell populations, data were retrieved from two public datasets (Xu et al, 2022; Olbrecht et al, 2021), including seven malignant and five non-malignant samples in one dataset, and two malignant and one non-malignant sample in the other. Data were analyzed and integrated using Seurat (v5). Cell clusters were annotated using both SingleR and known marker genes. Expression across ovarian cell types was visualized using Seurat's DotPlot function.

The list of FDA-approved drug targets was retrieved from the Human Protein Atlas database (https://www.proteinatlas.org/search/protein_class:FDA+approved+drug+targets).

For drug-response curve and drug synergy calculations, we used the SynergyFinder R package and web application (Zheng et al, 2022). We used the highest single agent (HSA) score to assess drug synergy.

## Declaration of generative AI and AI-assisted technologies in the writing process

During the preparation of this manuscript, the authors used Paperpal in order to improve the readability and language of the text. After using this tool, the authors reviewed and edited the content as needed and take full responsibility for the content of the publication.

## Data availability

The mass spectrometry proteomics data have been deposited to the ProteomeXchange Consortium (http://proteomecentral.proteomexchange.org) via the PRIDE partner (Perez-Riverol et al, 2019) with the dataset identifier PXD069630. CycIF images (Fig. 4) have been deposited at BioImage Archive under accession number S-BIAD2315 (https://www.ebi.ac.uk/biostudies/bioimages/studies/S-BIAD2315). All data reported in this paper will be shared by the lead contact upon reasonable request. Any additional information required to reanalyze the data reported in this paper is available from the lead contact upon request.

The source data of this paper are collected in the following database record: biostudies:S-SCDT-10_1038-S44320-025-00168-4.

## Peer review information

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

## Acknowledgements

We thank our colleagues at the Max Delbrück Center (MDC) and Charité for their support and fruitful discussion. In particular, Janett König supported multiplex imaging experiments. We thank Tobias Janik and Ines Koch for their help with slide preparation and cell line experiments. We thank Oliver Klein for tests on MALDI mass spectrometry imaging and Sabrina Geisberger and Stefan Kempa for support with targeted metabolomics experiments. We also thank Loren Méar and Cecilia Lindskog for their great support on RNAseq data analysis of HGSOC stromal cell populations. Furthermore, we acknowledge the MDC technology platform 'Proteomics' and 'Advanced light microscopy' for their great support. Gaetano Gargiulo (MDC) and Elena Ioana Braicu (Charité), we thank them for their critical feedback on the manuscript and Gina Dörpholz and Pia Larsen for administrative support. A.M, S.F., and F.C. acknowledge funding support by the Federal Ministry of Education and Research (BMBF), as part of the National Research Initiatives for Mass Spectrometry in Systems Medicine, under grant agreement No. 161L0222. This project received funding from the European Research Council (ERC) under the European Union's Horizon 2020 research and innovation program (grant agreement No. 101115681) and support by the ERC (ERC starting grant). M.P.D. is a Clinician Scientist as part of the Berlin Institute of Health Clinician Scientist Program. The work of M.P.D. is supported by a DKTK Berlin Young Investigator Grant 2022 and Berliner Krebsgesellschaft Grant (DRFF202204).

## Author contributions

**Anuar Makhmut**: Conceptualization; Data curation; Formal analysis; Investigation; Visualization; Methodology; Writing—review and editing.
**Mihnea P Dragomir**: Conceptualization; Resources; Data curation; Formal analysis; Funding acquisition; Investigation; Visualization; Methodology. **Sonja Fritzsche**: Methodology. **Markus Moebs**: Data curation. **Wolfgang D Schmitt**: Data curation. **Eliane T Taube**: Conceptualization; Supervision; Funding acquisition; Project administration. **Fabian Coscia**: Conceptualization; Formal analysis; Supervision; Funding acquisition; Investigation; Visualization; Writing—original draft; Project administration; Writing—review and editing.

Source data underlying figure panels in this paper may have individual authorship assigned. Where available, figure panel/source data authorship is listed in the following database record: biostudies:S-SCDT-10_1038-S44320-025-00168-4.

## Funding

## Disclosure and competing interests statement

The authors declare no competing interests.

# Expanded View Figures

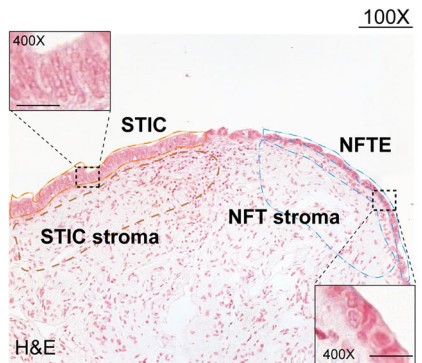 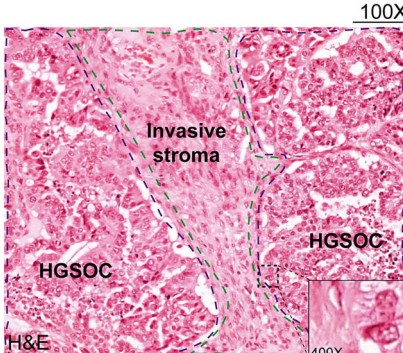

**Figure EV1. Pathology-guided ultra-low input proteomics of HGSOC precursor lesions.**

Representative H&E stains of normal fallopian tube epithelium (NFTE), serous tubal intraepithelial carcinoma (STIC), and invasive carcinoma (IC) and corresponding epithelial and stromal compartments. Scale bars: 100X: 100 μm, 400X: 20 μm.

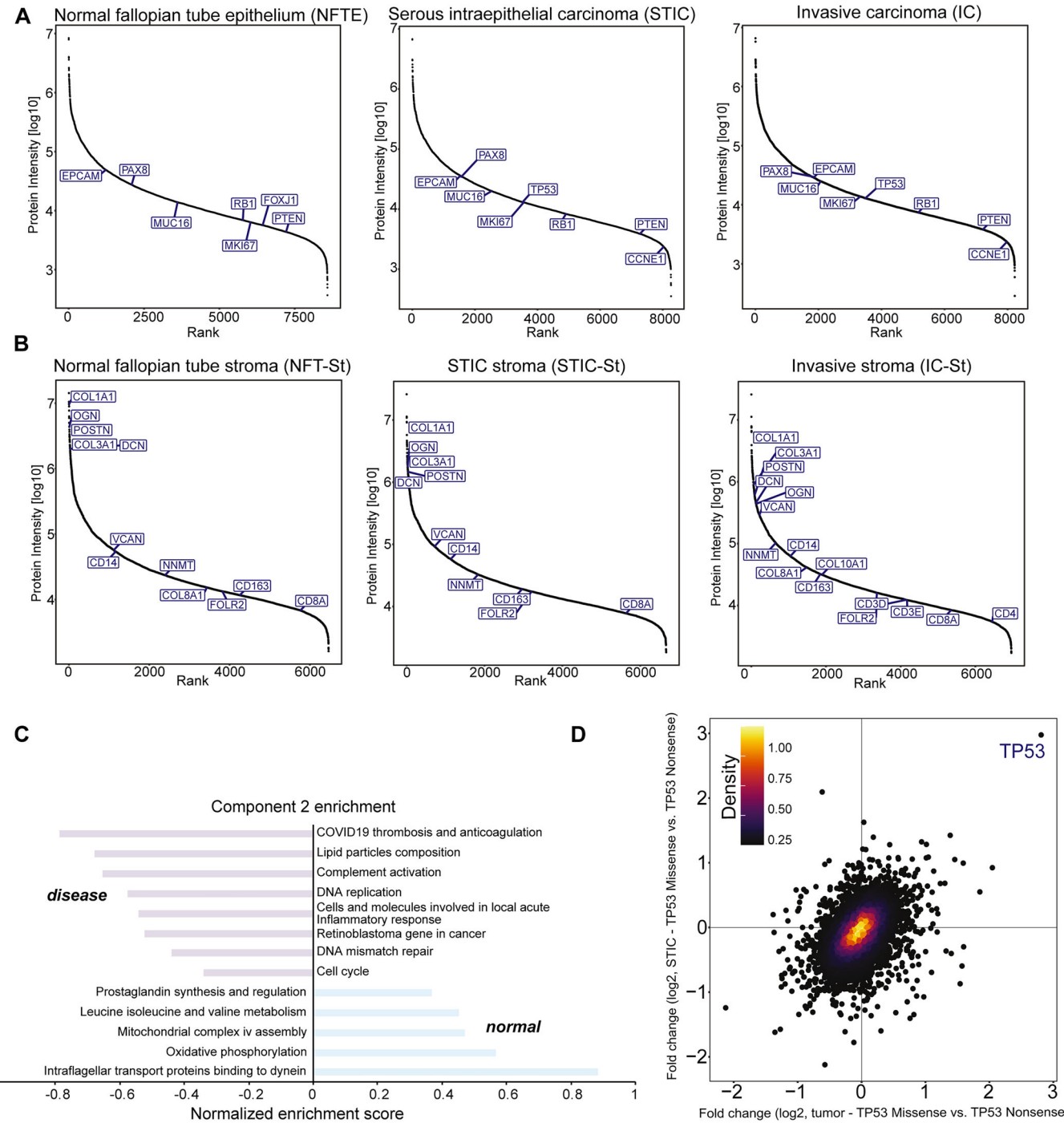

**Figure EV2. Spatially resolved proteomes reflect disease-specific alterations at the bulk level depth.**

(A, B) Dynamic range of median protein abundance for epithelial (A) and stromal (B) compartment samples. Known ovarian cancer-, cell type-, and stromal markers are highlighted. A minimum of 18 quantified values were required for each marker to be displayed. Proteins with 50% valid values for each group are shown. (C) Pathway enrichment analysis (Hallmarks) based on PC2 reveals differences between normal and disease compartments. Selected terms with a Benjamini–Hochberg FDR <0.05 are shown. (D) Scatter plot of *t*-test results between p53 overexpression and p53 null mutation patients, indicating that p53 is the only differentially abundant protein between the two groups.

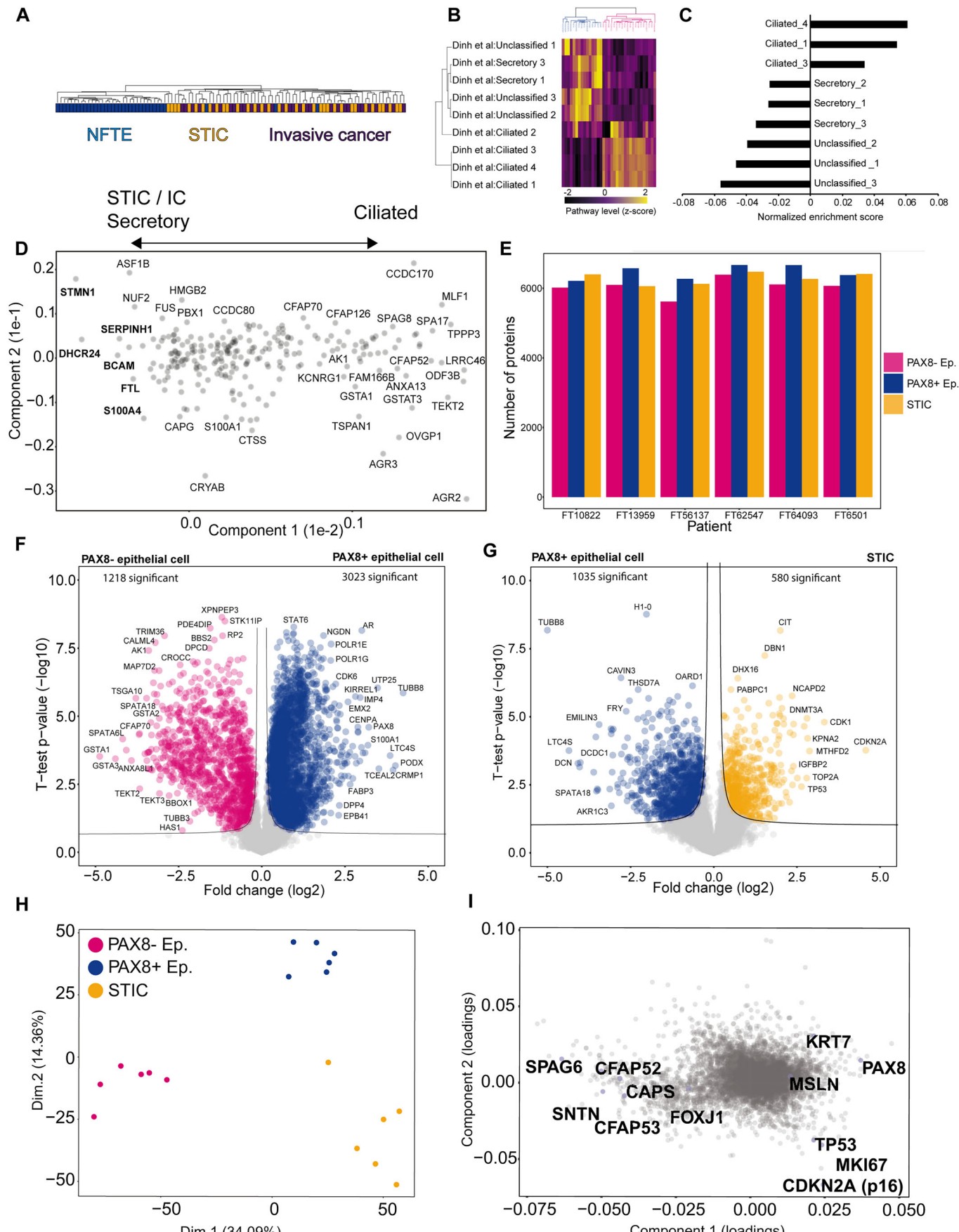

◀

**Figure EV3.   Precursor lesions feature histological markers and cell-of-origin signatures.**

(A) Hierarchical tree showing the clustering pattern across regions (normal fallopian tube epithelium, STIC, and invasive fallopian tube carcinoma), revealing three epithelial samples that are atypical lesions based on proteome profiling. (B) Unsupervised hierarchical clustering of all normal fallopian tube epithelial samples based on cell type abundance scores. Cell type abundance scores were calculated based on the annotation matrix algorithm (Perseus) for all fallopian tube epithelial cell types identified by Dinh et al Two main clusters of secretory and ciliated cell-enriched normal fallopian tube epithelium samples were identified. (C) Enriched cell type signatures along PC1, related to Fig. 3F. (D) PCA loadings related to Fig. 3F. Note that proteins on the left are higher in STIC, invasive carcinoma, and secretory-like epithelial cells, whereas proteins on the right are higher in ciliated epithelial cells. (E) Barplots showing the number of proteins identified in the PAX8 negative (dark pink), PAX8 positive (dark blue), and STIC (yellow) compartments across six patients. (F) Volcano plot of the pairwise proteomic comparison between PAX8 positive (dark blue, 6 samples) and PAX8 negative (dark pink, 6 samples) epithelial samples. Cell and functional markers with the highest abundance change and significance are highlighted (two-sided Student's *t*-test, FDR <0.05). (G) Volcano plot of pairwise proteomic comparison between STIC (yellow, 6 samples) and PAX8 positive (dark blue, six samples) samples. Cell and functional markers with the highest abundance change and significance are highlighted (two-sided Student's *t*-test, FDR <0.05). (H) PCA of all quantified proteins comparing STICs (yellow), secretory (dark blue), and ciliated cell (dark pink) samples. (I) PCA loadings show known secretory and ciliated markers. Related to panel (H).

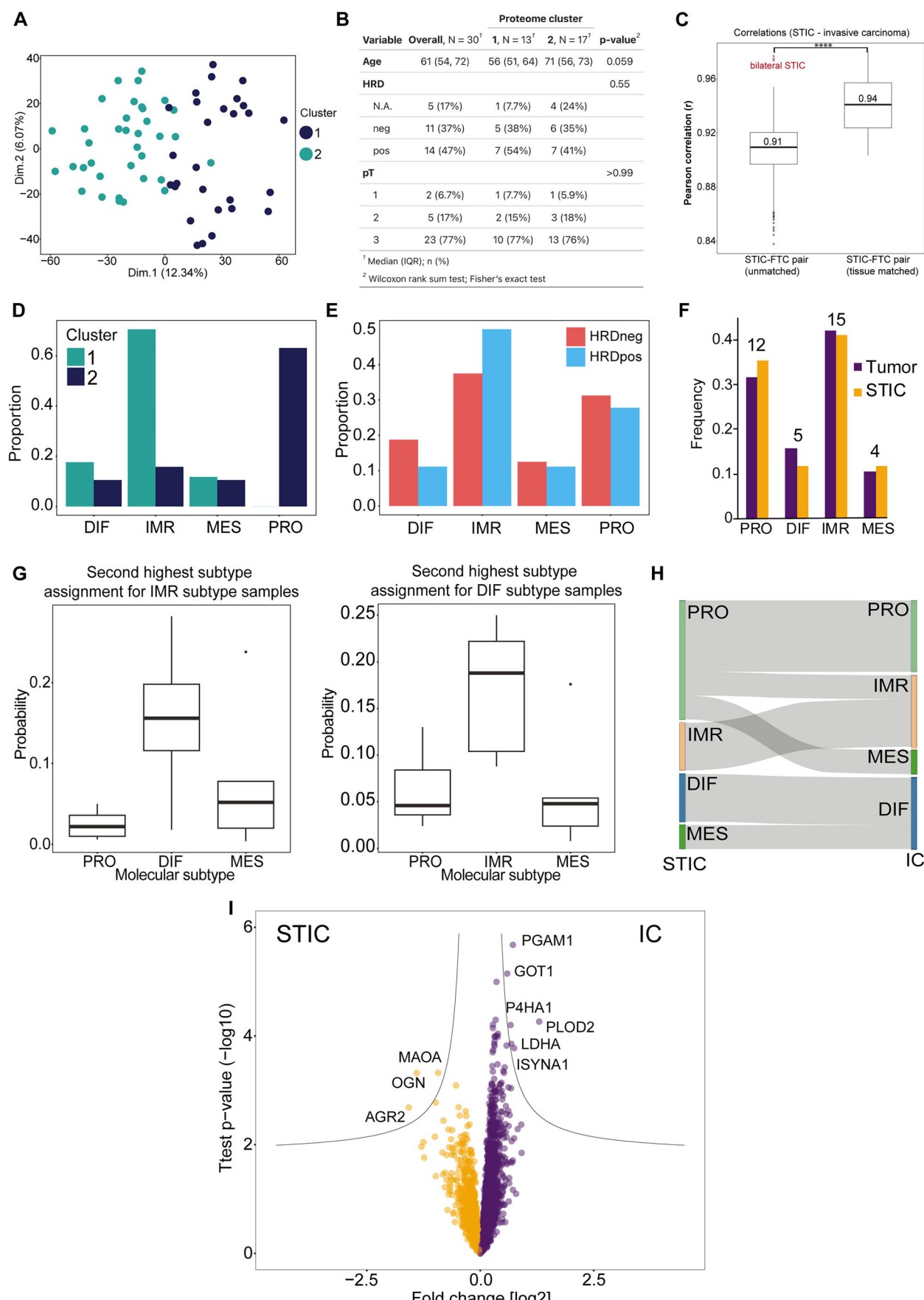

◀  **Figure EV4.   Refined molecular subtyping from spatial proteomics data.**

(**A**) Principal component analysis of STIC and invasive carcinoma samples corresponding to Fig. 3A. (**B**) Multivariate statistical analysis of patients in clusters 1 and 2. Note that cluster 2 samples showed a trend towards higher patient age at diagnosis ($p = 0.059$). (**C**) Boxplots showing proteomic correlations between STIC and invasive carcinoma samples between matched and unmatched samples, corresponding to intrapatient and interpatient correlations. Boxplots define the range of the data (whiskers), 25th and 75th percentiles (box), and medians (solid line). (**D**) Molecular subtype assignment of cluster 1 and 2 samples based on the consensusOV algorithm(Chen et al, 2018). Only samples with a margin score of >0.2 were included for high classification confidence. (**E**) Molecular subtype assignment of HRD-positive and HRD-negative invasive cancer samples based on the consensusOV algorithm(Chen et al, 2018). (**F**) Frequency of molecular subtypes of STIC and IC proteomes. (**G**) Boxplots indicating the second-highest molecular subtype classification probabilities for samples with immunoreactive (left) and differentiated (right) subtype calls. Boxplots define the range of the data (whiskers), 25th and 75th percentiles (box), and medians (solid line). Number of samples per molecular subtype with margin score >0.2: DIF – 5, PRO – 8, IMR – 5, MES – 2. (**H**) Sankey plot showing convergence or divergence of molecular subtypes for patient-matched STIC and invasive carcinoma pairs. (**I**) Volcano plot of pairwise proteomic comparison between STIC (yellow) and invasive fallopian tube carcinoma (violet) samples. Markers with the highest abundance change and significance are highlighted (two-sided Student's *t*-test, FDR <0.05). Number of samples per group: STIC – 35, IC – 31.

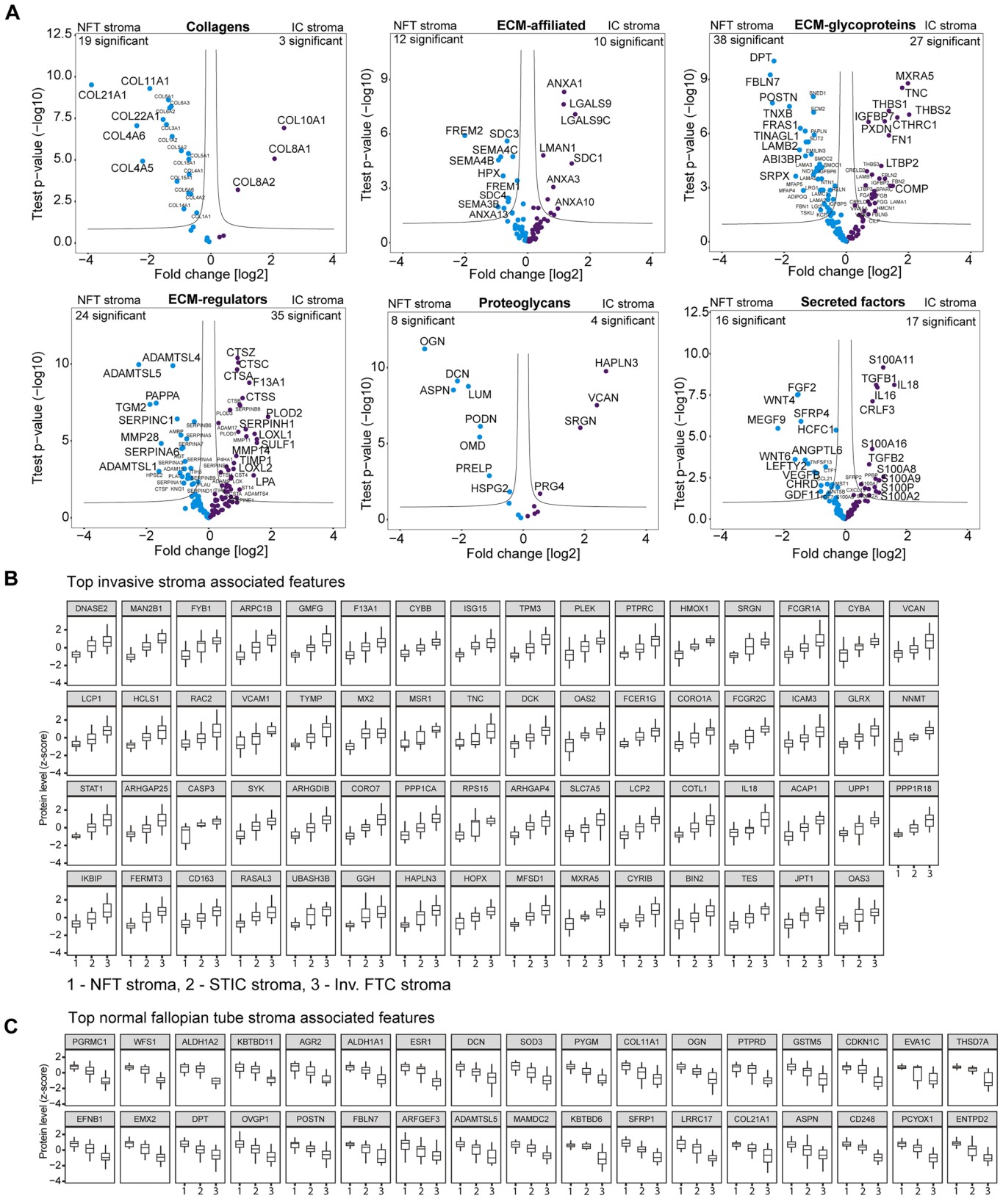

**A**

Collagens — NFT stroma (19 significant) / IC stroma (3 significant)

ECM-affiliated — NFT stroma (12 significant) / IC stroma (10 significant)

ECM-glycoproteins — NFT stroma (38 significant) / IC stroma (27 significant)

ECM-regulators — NFT stroma (24 significant) / IC stroma (35 significant)

Proteoglycans — NFT stroma (8 significant) / IC stroma (4 significant)

Secreted factors — NFT stroma (16 significant) / IC stroma (17 significant)

**B** Top invasive stroma associated features

1 - NFT stroma, 2 - STIC stroma, 3 - Inv. FTC stroma

**C** Top normal fallopian tube stroma associated features

1 - NFT stroma, 2 - STIC stroma, 3 - Inv. FTC stroma

◄ **Figure EV5.  Mapping progressive ECM remodeling reveals stromal drug targets.**

(**A**) Volcano plot of pairwise proteomic comparison between IC stroma (purple, 29 samples) and NFT stroma (light blue, 31 samples) samples across different matrisome categories (collagens, ECM-affiliated, ECM glycoproteins, ECM regulators, proteoglycans, secreted factors). Markers with the highest abundance change and significance are highlighted (two-sided Student's *t*-test, FDR <0.05). (**B, C**) Boxplots of relative protein levels (group average, z-scored) for IC stroma (29 samples) (**B**) and NFT stroma (31 samples) (**C**)-associated features. Top-ranked protein features were identified by support vector machine classification, as shown in Fig. 5E. Boxplots define the range of the data (whiskers), 25th and 75th percentiles (box), and medians (solid line).

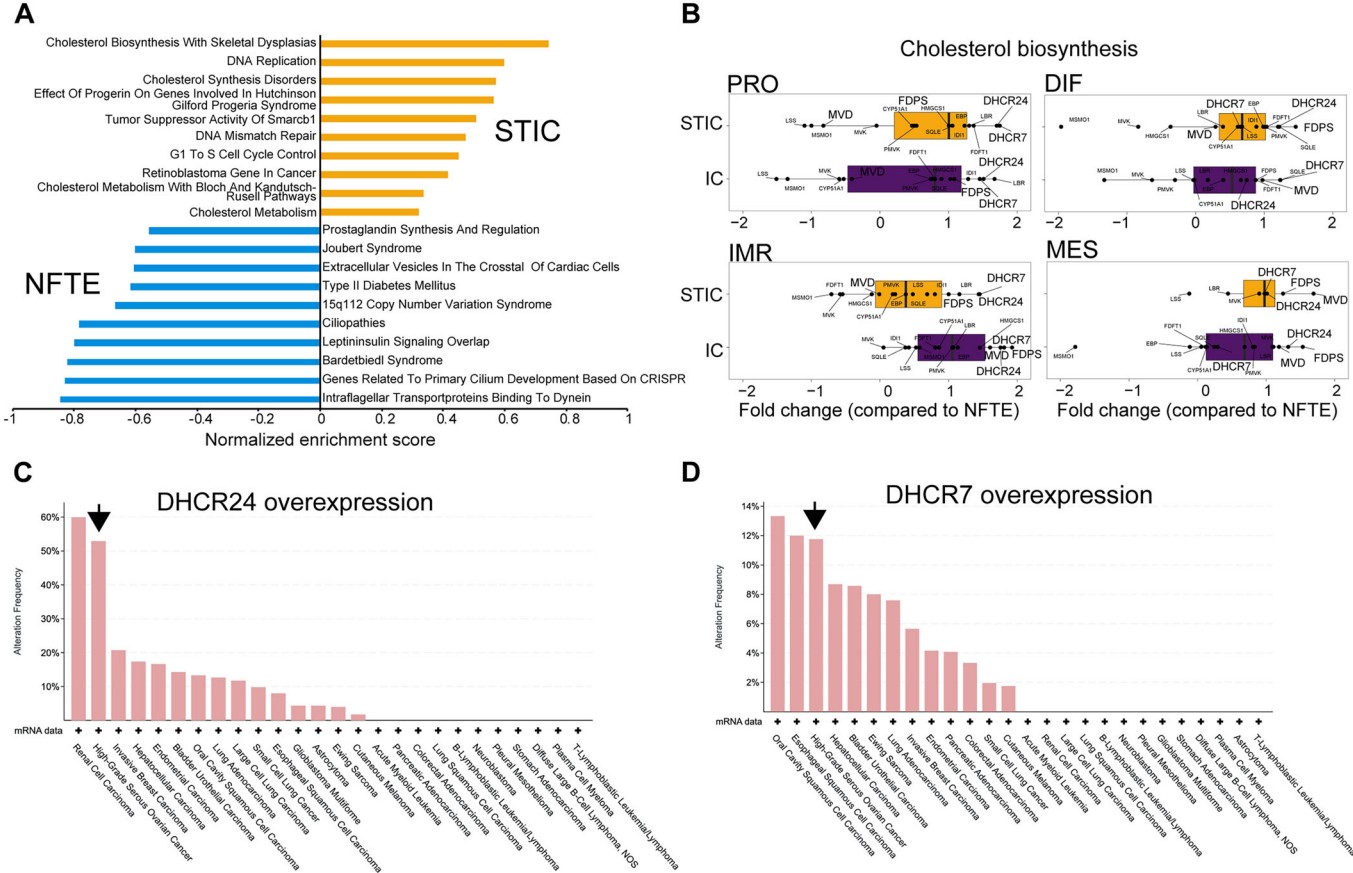

**Figure EV6. Identification of commonly dysregulated pathways of therapeutic relevance.**

(A) Pathway enrichment analysis (Hallmarks) based on t-test difference between STIC (yellow) and normal fallopian tube epithelium (light blue) samples. Selected pathways with a Benjamin–Hochberg FDR <0.05 are shown. (B) Protein fold changes (STIC vs. NFTE and IC vs NFTE, FDR <0.05) of all cholesterol biosynthesis pathway proteins across four molecular subtypes of HGSOC. Proteins with significant fold change are highlighted in bold. (C, D) Frequency of DHCR24 and DHCR7 genetic alteration (overexpression), respectively, across different cancer types analyzed by cBioPortal.

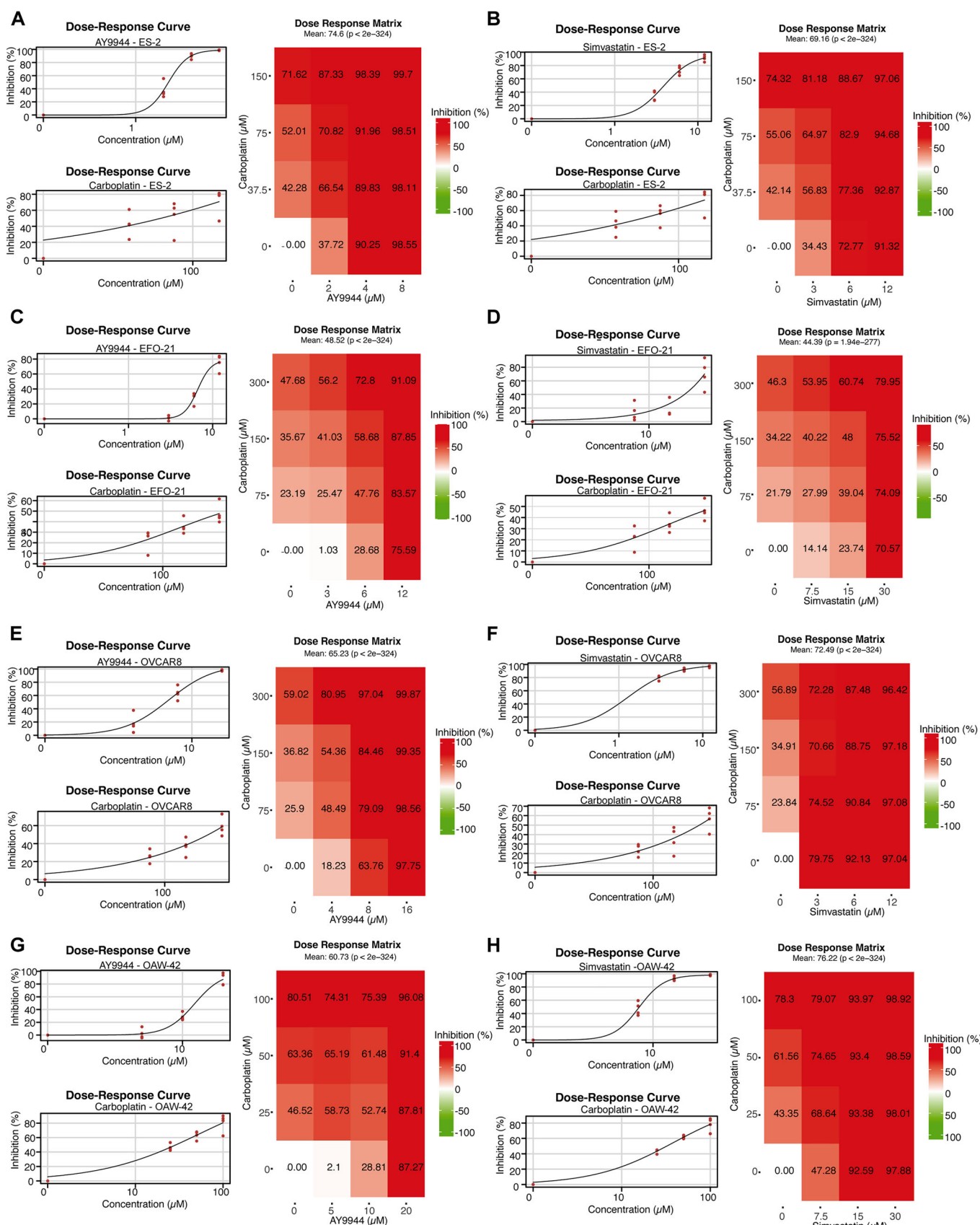

**Figure EV7. Cholesterol biosynthesis inhibition sensitizes ovarian cancer cells to carboplatin.**

(**A–H**) Dose–response curves (top left: AY9944 or simvastatin; bottom left: carboplatin) and dose–response matrices (right) for ES-2 (**A**, **B**), EFO-21 (**C**, **D**), OVCAR-8 (**E**, **F**), and OAW-42 (**G**, **H**) cells treated with AY9944 and carboplatin or simvastatin and carboplatin, respectively, alone or in combination. Experiments were performed in biological sextuplicate, each with $n = 3$ technical replicates.

