## [Peer Review File · Molecular Systems Biology]

Spatial proteomics of ovarian cancer precursors delineates early disease changes and drug targets

Anuar Makhmut, Mihnea Dragomir, Sonja Fritzsche, Markus Möbs, Wolfgang Schmitt, Eliane Taube, and Fabian Coscia

Corresponding author(s): Fabian Coscia (fabian.coscia@mdc-berlin.de)

Review Timeline:

Submission Date:	15th Apr 25
Editorial Decision:	18th May 25
Revision Received:	15th Aug 25
Editorial Decision:	12th Sep 25
Revision Received:	17th Oct 25
Accepted:	22nd Oct 25

Editor: Jingyi Hou

Transaction Report:

18th May 2025

Manuscript Number: MSB-2025-13047

Title: Spatial proteomics of ovarian cancer precursors delineates early disease changes and drug targets

Author: Anuar Makhmut

Mihnea Dragomir

Sonja Fritzsche

Markus Möbs

Wolfgang Schmitt

Eliane Taube

Fabian Coscia

Dear Fabian,

Thank you for submitting your work to Molecular Systems Biology. We have now heard back from the three reviewers who agreed to evaluate your manuscript. As you will see from the reports below, the reviewers are overall positive about the study. They raise, however, a series of concerns, which we would ask you to address in a major revision.

I think that the recommendations of the reviewers are rather clear so there is no need to repeat the points listed below. All issues raised by the reviewers need to be satisfactorily addressed. As you may already know, our editorial policy allows in principle a single round of major revision, so it is essential to provide responses to the reviewers' comments that are as complete as possible. Please feel free to contact me in case you would like to discuss in further detail any of the issues raised by the reviewers.

On a more editorial level, we would ask you to address the following issues:

- Please provide a .docx formatted version of the manuscript text (including legends for main figures, EV figures and tables). Please make sure that the changes are highlighted to be clearly visible.
- Please provide individual production quality figure files as .eps, .tif, .jpg (one file per figure).
- Please provide a .docx formatted letter INCLUDING the reviewers' reports and your detailed point-by-point responses to their comments. As part of the EMBO Press transparent editorial process, the point-by-point response is part of the Review Process File (RPF), which will be published alongside your paper.
- Please note that all corresponding authors are required to supply an ORCID ID for their name upon submission of a revised manuscript.
- We replaced Supplementary Information with Expanded View (EV) Figures and Tables that are collapsible/expandable online (see examples in <http://msb.embopress.org/content/11/6/812>). A maximum of 5 EV Figures can be typeset. EV Figures should be cited as 'Figure EV1, Figure EV2' etc... in the text and their respective legends should be included in the main text after the legends of regular figures.

Additional Tables/Datasets should be labeled and referred to as Table EV1, Dataset EV1, etc. Legends have to be provided in a separate tab in case of .xls files. Alternatively, the legend can be supplied as a separate text file (README) and zipped together with the Table/Dataset file.

For the figures and tables that you do NOT wish to display as Expanded View figures, they should be bundled together with their legends in a single PDF file called *Appendix*, which should start with a short Table of Content. Each legend should be below the corresponding Figure/Table in the Appendix. Appendix figures and tables should be referred to in the main text as: "Appendix Figure S1, Appendix Figure S2, Appendix Table S1" etc. See detailed instructions regarding expanded view here: <https://www.embopress.org/page/journal/17444292/authorguide#expandedview>.

- Before submitting your revision, primary datasets (and computer code, where appropriate) produced in this study need to be deposited in an appropriate public database (see <http://msb.embopress.org/authorguide-dataavailability> <https://www.embopress.org/page/journal/17444292/authorguide#dataavailability>). Please remember to provide a reviewer password if the datasets are not yet public. The accession numbers and database should be listed in a formal "Data Availability" section (placed after Materials & Method) that follows the model below (see also <https://www.embopress.org/page/journal/17444292/authorguide#dataavailability>). Please note that the Data Availability Section is restricted to new primary data that are part of this study.
Data availability

-At EMBO Press we ask authors to provide source data for the main figures. Our source data coordinator will contact you to discuss which figure panels we would need source data for and will also provide you with helpful tips on how to upload and organize the files.

- Our journal encourages inclusion of *data citations in the reference list* to directly cite datasets that were re-used and obtained from public databases. Data citations in the article text are distinct from normal bibliographical citations and should directly link to the database records from which the data can be accessed. In the main text, data citations are formatted as follows: "Data ref: Smith et al, 2001". In the Reference list, data citations must be labeled with "[DATASET]". A data reference must provide the database name, accession number/identifiers and a resolvable link to the landing page from which the data can be accessed at the end of the reference. Further instructions are available at .

- We updated our journal's competing interests policy in January 2022 and request authors to consider both actual and perceived competing interests. Please review the policy <https://www.embopress.org/competing-interests> and update your competing interests if necessary.

Please use the heading "Disclosure statement and competing interests".

- All Materials and Methods need to be described in the main text using our 'Structured Methods' format. According to this format, the Methods section includes a Reagents and Tools Table (listing key reagents, experimental models, software and relevant equipment and including their sources and relevant identifiers) followed by a Methods and Protocols section describing the methods, ideally using a step-by-step protocol format. The aim is to facilitate adoption of the methodologies across labs.

Please download and fill our Reagents and Tools Table template (.docx), which you can find in our author guidelines: <https://www.embopress.org/page/journal/17444292/authorguide#structuredmethods>.

-Regarding data quantification:

Please ensure to specify the name of the statistical test used to generate error bars and P values, the number (n) of independent experiments (please specify technical or biological replicates) underlying each data point and the test used to calculate p-values in each figure legend. Discussion of statistical methodology can be reported in the materials and methods section, but figure legends should contain a basic description of n, P and the test applied.

Graphs must include a description of the bars and the error bars (s.d., s.e.m.).

- Please provide a "standfirst text" summarizing the study in one or two sentences (approximately 250 characters, including space), three to four "bullet points" highlighting the main findings and a "synopsis image" (550px width and 400-600 px height, PNG format) to highlight the paper on our homepage.

Here are a couple of examples:

<https://www.embopress.org/doi/10.15252/msb.20199356>

<https://www.embopress.org/doi/10.15252/msb.20209475>

<https://www.embopress.org/doi/10.15252/msb.209495>

When you resubmit your manuscript, please download our CHECKLIST (<https://www.embopress.org/pb-assets/embo-site/EMBO%20Press%20Author%20Checklist-1642513524327.xlsx>) and include the completed form in your submission.

Please note that the Author Checklist will be published alongside the paper as part of the transparent process (<https://www.embopress.org/page/journal/17444292/authorguide#transparentprocess>).

If you feel you can satisfactorily deal with these points and those listed by the referees, you may wish to submit a revised version of your manuscript. Please attach a covering letter giving details of the way in which you have handled each of the points raised by the referees. A revised manuscript will be once again subject to review and you probably understand that we can give you no guarantee at this stage that the eventual outcome will be favorable.

I look forward to receiving the revised manuscript soon.

Kind regards,

Jingyi

Sincerely,
Jingyi

Jingyi Hou, PhD
Senior Editor
Molecular Systems Biology

We realize that it is difficult to revise to a specific deadline. In the interest of protecting the conceptual advance provided by the work, we recommend a revision within 3 months (16th Aug 2025). Please discuss the revision progress ahead of this time with the editor if you require more time to complete the revisions. Use the link below to submit your revision:

IMPORTANT: When you send your revision, we will require the following items:

1. the manuscript text in LaTeX, RTF or MS Word format
2. a letter with a detailed description of the changes made in response to the referees. Please specify clearly the exact places in the text (pages and paragraphs) where each change has been made in response to each specific comment given
3. three to four 'bullet points' highlighting the main findings of your study
4. a short 'blurb' text summarizing in two sentences the study (max. 250 characters)
5. a 'thumbnail image' (550px width and max 400px height, Illustrator, PowerPoint or jpeg format), which can be used as 'visual title' for the synopsis section of your paper.
6. Please include an author contributions statement after the Acknowledgements section (see <https://www.embopress.org/page/journal/17444292/authorguide>)
7. Please complete the CHECKLIST available at (<https://bit.ly/EMBOPressAuthorChecklist>). Please note that the Author Checklist will be published alongside the paper as part of the transparent process (<https://www.embopress.org/page/journal/17444292/authorguide#transparentprocess>).
8. When assembling figures, please refer to our figure preparation guideline in order to ensure proper formatting and readability in print as well as on screen:
<https://bit.ly/EMBOPressFigurePreparationGuideline>
See also figure legend guidelines: <https://www.embopress.org/page/journal/17444292/authorguide#figureformat>
9. Please note that corresponding authors are required to supply an ORCID ID for their name upon submission of a revised manuscript (EMBO Press signed a joint statement to encourage ORCID adoption). (<https://www.embopress.org/page/journal/17444292/authorguide#editorialprocess>)
Currently, our records indicate that the ORCID for your account is 0000-0002-2244-5081.

Please click the link below to modify this ORCID:
Link Not Available

10. At EMBO Press we ask authors to provide source data for the main manuscript figures. You will receive a separate email with instructions for providing source data with your revised manuscript, including how to upload and organize the files.
11. Include a Reagents and Tools Table, which can be downloaded from our author guidelines (<https://www.embopress.org/page/journal/17444292/authorguide#structuredmethods>)

*** PLEASE NOTE *** As part of the EMBO Press transparent editorial process initiative (see our Editorial at <https://dx.doi.org/10.1038/msb.2010.72>), Molecular Systems Biology publishes online a Review Process File with each accepted manuscripts. This file will be published in conjunction with your paper and will include the anonymous referee reports, your point-by-point response and all pertinent correspondence relating to the manuscript. If you do NOT want this File to be published, please inform the editorial office at contact@molsystbiol.org within 14 days upon receipt of the present letter.

Reviewer #1:

General comments

The manuscript presents a well-designed study utilizing spatial proteomics to map ovarian cancer precursors. The study is timely, showing the utility of this still relatively novel technology from a clinical perspective, and the advantages of spatial proteomics. Overall, the manuscript is clearly written, and the conclusions drawn are adequate. I only have a few minor comments and suggestions.

Introduction

- Very engaging and well-written.
- Line 3: Not entirely sure about the use of the word "variant" here perhaps consider rephrasing? But I'm not confident either...

Results

- Line 60 / Fig. 1C: This figure might be a bit hard to interpret at first glance, consider improving readability.
- Line 63 / EV1A: What about P3? is there no data available?
- Line 85: Consider adding a reference to Fig. EV1A for clarity.
- Line 88: should be "EV1A" instead of "S1A."
- Line 98: There's a font issue here, or it might just be due to my printed version?
- Lines 170-171 / Fig. 3B and 3D: The pathway enrichment analysis and protein markers shown don't strongly support the presence of a ciliated signature, could this be clarified or strengthened?
- Line 188: "Many more" feels a bit vague, could you specify the number or provide an estimate?
- Line 193 / Fig. 3J: Text is quite small and hard to read, maybe increase the font size? Fig. 3K: The visualization works, but there might be a clearer way to present it.
- Line 234 / Fig. 4E legend: Consider changing cluster "II" to "2" for consistency with the rest of the figure panels.

Figure-specific suggestions

- Fig. 2D: Increase the font size of the source labels.
- Fig. 2F: The numbers in the dark purple areas are hard to read.
- Fig. 2I: Do the colors carry meaning or are they just aesthetic? Consider clarifying in the legend.

Discussion _ Line 400: Font issue with "metabolomic reprogramming", though again, this could be due to the printed version.

Reviewer #2:

The authors, Makhmut, Dragomir, and colleagues, have submitted the manuscript entitled "Spatial proteomics of ovarian cancer precursors delineates early disease changes and drug targets." In this study, an ultra-low input spatial proteomics workflow was employed to profile normal fallopian tube tissue in comparison with precancerous lesions and invasive carcinoma, using a 36-patient FFPE cohort. The authors aim to identify changes in protein abundance associated with tumour progression and its microenvironment, with the goal of highlighting therapeutically relevant alterations.

Overall, the manuscript is well written; however, the figures require improvement to effectively communicate the research findings to the reader. The study presents novel datasets that will serve as a valuable resource and foundation for future investigations into therapeutic strategies for ovarian cancer. As such, the manuscript holds significant relevance for the ovarian cancer and mass spectrometry-based spatial proteomics research communities. Nevertheless, the study remains largely descriptive and requires both minor and major revisions, as outlined below, before it can be recommended for publication.

Major points

1. Throughout the whole manuscript a comprehensive overview of sample types is missing which leads to confusion for the reader. Extending Figure 1 by what is shown in Figure EV1 would be helpful, as well as a simple illustration of what is shown in Figure 2H. In addition, the authors should stick to the same colouring for each sample type throughout all figures - which is ideally introduced in the overview illustration.

a. Clarify the tissue class "Pre-malignant" in Figure 3B (pink)

2. The manuscript reaches its most compelling point in Figure 6, where it transitions from a descriptive study to one with translational and therapeutic relevance. In comparing NFTE and STIC tissues, the authors identify an upregulation of the "Cholesterol Biosynthesis pathway" in STIC, proposing this pathway as a potential therapeutic target in ovarian cancer. To support this hypothesis, they treat four ovarian cancer cell lines with AY-9944, simvastatin, and carboplatin, and subsequently report a synergistic effect between AY-9944 and chemotherapy. However, as shown in Figure EV6, this conclusion is based solely on drug response experiments in a single cell line (OVCAR-8). While the result is promising, a more comprehensive analysis involving additional cell lines or models would significantly enhance the impact and robustness of the manuscript's translational claims.

a. Each of the three treatments should be tested in a dose-dependent manner alone as well as in combination on all four cell lines.

b. The cell lines respond differently to the treatments. A genetic and proteomic characterisation of the cell lines should be

conducted to potentially elucidate the underlying molecular cause.

c. Figure 6G should be replaced. The 'synergy score' axis is skewed with the numbers partly unreadable, and scaling of the synergy score to include negative values is unnecessary as no negative values appear in the plot. It should be replaced with something similar to EV6G, in line with further experiments suggested above.

Minor points

1. Red/green should not be used as a colour scheme in plots and should be replaced (e.g. Figure 3E).
2. The 'Top 100 stromal proteins' are mentioned in the text but should also be shown in an additional figure or at least marked in Figure 2.
3. Readability (font size) of figure text needs to be improved throughout the manuscript, e.g. Figure 2D; EV3B.
4. Axis labelling needs to be improved to make it easier for the reader to understand (e.g. Figure 4B, i.e. "Relative protein level (log2)")
5. Figure formatting is not consistent and should be reworked throughout the paper, e.g. removing background lines (e.g. Figure 3F).
6. Clarify whether all upregulated proteins or the proteins above the significance threshold of Figure 3B contribute to the enriched pathways in Figure 3D. If a threshold was used (which is highly recommended), only the proteins above the cutoff line in 3B should be coloured in pink. In addition, at least the proteins highlighted in the text should be linked to the respective dot in the volcano plot. All volcano plots require a rework accordingly.
7. More comprehensive and consistent naming is highly recommended - e.g. "normal" vs NFTE
8. The titles of the Figures and subchapters should be adjusted to reflect the main message of the paragraph and not the overall story line.

Reviewer #3:

The article from Makhmut and Dragomir et al. utilizes a cutting-edge spatial proteomic approach for a comprehensive characterization of the epithelial and stromal compartment of tissues from 36 HGSOE patients. Specifically, the authors characterized the proteome of normal fallopian tubes (NFTE), serous tubal intraepithelial lesions (STICs or early lesions), and invasive carcinoma (IC or advanced disease), providing well-resolved epithelial and stromal proteomes that represent a valuable resource for the scientific community. The authors also use DVP to characterize the proteome of PAX8+ (secretory cells) and PAX8- (ciliated-like) cells, demonstrating that PAX8+ secretory cell proteomes display overlapping signatures with STICs, thus supporting the view of the ovarian cancer community that HGSOE originates from secretory cells. Furthermore, the authors compare both epithelial and stromal proteomes of STICs (for the epithelium) or normal-like (for the stroma) and IC revealing a high degree of similarity in the epithelial compartment, and well-defined differences in proteomic composition of the stromal compartment. Finally, the authors compared the epithelial proteomes of NFTE and STICs, demonstrating pronounced differences, particularly as it relates to metabolism. They identify the cholesterol biosynthesis pathway as a promising drug target in combination with chemotherapy and demonstrated the efficacy of such combination in vitro.

Overall, this study generates a comprehensive clinical dataset of HGSOE of great value for the scientific community. The authors also conduct elegant comparisons with publicly available proteomics and gene expression data in equivalent settings, making interesting parallels between RNA and protein alterations in ovarian cancer. The design, approach, results and analysis strategies employed in the study are impressive. I also commend the authors for putting together such organized source data file. Overall, I have a few suggestions that are listed below that I believe would enhance the impact and translational value of the study.

Major points:

1. The authors observe a high similarity between the STIC and IC epithelial proteomes, with only 9 proteins being significantly regulated. However, the samples did segregate into an immunoreactive cluster 1 and a proliferative cluster 2, with well-defined proteomic signatures (Figure 4). In spite of the clustering being independent of TP53 mutation type, histological stage, and HRD status, it would be interesting for the authors to conduct a correlation analysis between clusters 1 and 2 with patient survival, which is an important clinical parameter that has not been explored in this study. This has the potential to shed light into the implications of these epithelial signatures in clinical outcome. Furthermore, a link between the epithelial and stromal compartments has not been addressed. Considering that the stroma can influence the behavior of cancer cells, it is important to evaluate if there are unique features of the stroma associated with these clusters.

2. Using SVM, the authors identify a 100-protein stromal signature that can distinguish the normal from the malignant stroma. Importantly, 9 of these proteins (Figure 5H) have FDA-approved drugs for specific targeting. This is a very promising finding that could be explored further. I would suggest that the authors use relevant stromal cells to test whether these FDA-approved drugs reduce their pro-tumorigenic traits and their ability to support invasion of cancer cells. I believe that this would certainly increase the impact and translational relevance of this study. Furthermore, the discussion addressing these 9 targets with FDA-approved

drugs is incomplete. Please expand for each target on which stromal cell population these proteins are expressed (e.g. authors can use publicly available single cell data) and the relevance of their targeting.

3. In Figure 5, when displaying the data from the normal stroma vs the invasive stroma clusters, the authors should include a Volcano Plot containing all the proteins identified (as they did in Figure 4C for the epithelial compartment), alongside with the source data table. The Volcano plot in Figure 5C only shows the proteins related to the Matrisome. Please include the full information in the revised manuscript.

4. The authors nicely compare the differences in the epithelial compartment between NFTE and STICs but do not explore such differences in the stromal compartment. Even though STIC stroma segregated half with NFTE and half with IC stroma, there might still be distinct differences between STIC and NFTE stroma that were not captured in the PC analysis. I would strongly recommend that at least a supplementary figure comparing the NFT and STIC stroma is included in the revised manuscript. This information will certainly add to their findings and will be a valuable resource for the scientific community studying the ovarian cancer stroma.

5. The authors identify DHCR7 and DHCR24 as significantly upregulated in epithelial STICs vs NFTE and used DHCR7 inhibitor to show the potential of cholesterol synthesis pathway in HGSOC in vitro. To validate their findings, the authors should use mass spec imaging to show increased cholesterol synthesis in the STICs compared to normal tissue.

6. It would be interesting to use available clinical data of cholesterol levels in HGSOC patients and investigate its correlation with patient survival and even therapy response.

7. In the discussion session, I recommend that the authors discuss how their findings can be explored for biomarker discovery aimed at early detection of HGSOC.

Minor points:

1. Page 10, line 318: correct figure is 5H, not 5F as indicated in the text.

2. Figures: Please use colorblind safe colors, in particular avoid green and red in the same plot when possible, for example Fig. 3e,f.

3. Fig. 5G. It is difficult to read what's written in black over violet.

4. The authors have uploaded on PRIDE the results of the search engine, but not the raw MS files. The authors should upload their raw MS files as it is common practice to enable access to the community.

Point-by-point response to the reviewers' comments

We thank all reviewers for their thorough and positive evaluation of our manuscript entitled "*Spatial proteomics of ovarian cancer precursors delineates early disease changes and drug targets*".

In the revised manuscript, we have included additional experiments and analyses, and edited the main text and figures in accordance to the reviewers' suggestions. We believe that these additions have further strengthened our study and further validated many of our results. Below, we provide a detailed point-by-point response to each of the reviewers' comments. We tracked text related changes in the provided manuscript file.

Reviewer #1:

General comments

The manuscript presents a well-designed study utilizing spatial proteomics to map ovarian cancer precursors. The study is timely, showing the utility of this still relatively novel technology from a clinical perspective, and the advantages of spatial proteomics. Overall, the manuscript is clearly written, and the conclusions drawn are adequate. I only have a few minor comments and suggestions.

We thank the reviewer for the very positive feedback of our work and the minor comments helping us to further improve our study.

Introduction

- Very engaging and well-written.

We are very pleased to hear this.

- Line 3: Not entirely sure about the use of the word "variant" here perhaps consider rephrasing? But I'm not confident either...

We agree that the word "variant" does not fit in this context and changed it to "subtype" instead.

Results

- Line 60 / Fig. 1C: This figure might be a bit hard to interpret at first glance, consider improving readability.

We have adjusted the opacity for better readability and added the age of each patient to the respective box. Please note that, in response to reviewer 2, we have replaced this figure with the former Fig. EV1A.

- Line 63 / EV1A: What about P3? is there no data available?

Initially, our dataset consisted of 36 patients. As one tissue specimen did not pass QC filtering, the final cohort consisted of 35 patients. We have clarified this in the main text and also highlighted in Table EV1 for which samples proteomics data was acquired.

- Line 85: Consider adding a reference to Fig. EV1A for clarity.

We have added this figure reference.

- Line 88: should be "EV1A" instead of "S1A."

This has been corrected now.

- Line 98: There's a font issue here, or it might just be due to my printed version?

We have re-evaluated our document to make sure that there are no font related issues.

- Lines 170-171 / Fig. 3B and 3D: The pathway enrichment analysis and protein markers shown don't strongly support the presence of a ciliated signature, could this be clarified or strengthened?

We agree that this was not entirely clear from the previous version. In the revised Fig. 3B and 3D (Revision Fig. 1, below), we now show the results of the pathway enrichment analysis for both sides (higher in atypical lesions or in NFTE samples). Ciliated signatures are among the top enriched pathways higher in normal fallopian tube epithelial (NFTE) samples.

Revision Figure 1: Left: Volcano plot of pairwise proteomic comparison between atypical (light-pink) and NFTE samples (light blue). Markers with the highest fold change are highlighted (two-sided t-test, permutation-based FDR < 0.05). Right: Pathway enrichment analysis based on the t-test difference between atypical and normal fallopian tube epithelial samples. Selected significantly enriched pathways are shown for pathways higher in pre-malignant samples (Benjamin-Hochberg FDR < 0.05).

- Line 188: "Many more" feels a bit vague, could you specify the number or provide an estimate?

We changed this statement to '27 other proteins with similar abundance profiles'.

- Line 193 / Fig. 3J: Text is quite small and hard to read, maybe increase the font size? Fig. 3K: The visualization works, but there might be a clearer way to present it.

We have increased the font size to 6 or higher and highlighted key enrichment terms in the Figure 3J.

- Line 234 / Fig. 4E legend: Consider changing cluster "II" to "2" for consistency with the rest of the figure panels.

We have changed this accordingly throughout the manuscript.

Figure-specific suggestions

- Fig. 2D: Increase the font size of the source labels.

We have increased the font size of the source labels for better readability.

- Fig. 2F: The numbers in the dark purple areas are hard to read.

We have revised this figure and placed the numbers next to the heatmap.

- Fig. 2I: Do the colors carry meaning or are they just aesthetic? Consider clarifying in the legend.

We clarified this in the figure legend as follows: "Point colors reflect protein levels and are for visual guidance, where brighter colors indicate higher protein levels and darker colors indicate lower protein levels."

Discussion _ Line 400: Font issue with "metabolomic reprogramming", though again, this could be due to the printed version.

We confirmed that there are no font related problems in the electronic and printed versions.

Reviewer #2:

The authors, Makhmut, Dragomir, and colleagues, have submitted the manuscript entitled "Spatial proteomics of ovarian cancer precursors delineates early disease changes and drug targets." In this study, an ultra-low input spatial proteomics workflow was employed to profile normal fallopian tube tissue in comparison with precancerous lesions and invasive carcinoma, using a 36-patient FFPE cohort. The authors aim to identify changes in protein abundance associated with tumour progression and its microenvironment, with the goal of highlighting therapeutically relevant alterations.

Overall, the manuscript is well written; however, the figures require improvement to effectively communicate the research findings to the reader. The study presents novel datasets that will serve as a valuable resource and foundation for future investigations into therapeutic strategies for ovarian cancer. As such, the manuscript holds significant relevance for the ovarian cancer and mass spectrometry-based spatial proteomics research communities. Nevertheless, the study remains largely descriptive and requires both minor and major revisions, as outlined below, before it can be recommended for publication.

We thank the reviewer for the thorough and positive feedback on our work and for acknowledging the importance as resource and foundation for future studies.

Major points

1. Throughout the whole manuscript a comprehensive overview of sample types is missing which leads to confusion for the reader. Extending Figure 1 by what is shown in Figure EV1 would be helpful, as well as a simple illustration of what is shown in Figure 2H. In addition, the authors should stick to the same colouring for each sample type throughout all figures - which is ideally introduced in the overview illustration.

We have now extended Fig. 1 and moved the information from former Fig. EV1A to the revised Fig. 1C. Our sampling strategy of different epithelial and stromal regions is exemplarily shown in the main Fig. 1B and also in the revised Fig. EV1. In addition, we provided a detailed table EV1, which contains a comprehensive overview of all collected and measured samples for each of the 35 patients. For more

consistency, we have also revised and edited the colors throughout the manuscript. We hope that these changes have clarified the previous confusion.

Revision Figure 2: Extended sample collection information, as shown in panel C. This figure is now the revised Fig.1.

a. Clarify the tissue class "Pre-malignant" in Figure 3B (pink)

By pre-malignant, we refer to samples that were initially annotated as normal (healthy) fallopian tube epithelium (NFTE), but which clearly deviated from the majority of NFTE proteomes. Additional histological analyses revealed that these samples likely represented pre-cancerous serous tubal intraepithelial lesions (STILs) and STICs. Such lesions are characterized by an atypical morphology and an aberrant expression of p16, the surrogate marker for STICs. We have now changed the term “pre-malignant” to “atypical” in the manuscript as we think it describes them more accurately.

2. The manuscript reaches its most compelling point in Figure 6, where it transitions from a descriptive study to one with translational and therapeutic relevance. In comparing NFTE and STIC tissues, the authors identify an upregulation of the "Cholesterol Biosynthesis pathway" in STIC, proposing this pathway as a potential therapeutic target in ovarian cancer. To support this

hypothesis, they treat four ovarian cancer cell lines with AY-9944, simvastatin, and carboplatin, and subsequently report a synergistic effect between AY-9944 and chemotherapy. However, as shown in Figure EV6, this conclusion is based solely on drug response experiments in a single cell line (OVCAR-8). While the result is promising, a more comprehensive analysis involving additional cell lines or models would significantly enhance the impact and robustness of the manuscript's translational claims.

a. Each of the three treatments should be tested in a dose-dependent manner alone as well as in combination on all four cell lines.

We fully agree that additional cell line experiments will further substantiate our findings. We now provide new experimental data to show single treatments, as well as combination treatments for all of the four ovarian cancer cell lines (OVCAR-8, ES-2, EFO-21, and OAW-42). For all cell lines, we tested the combination of carboplatin and the DHCR7 inhibitor AY9944, as well as carboplatin with simvastatin. Each drug was tested with three different concentrations, as shown in the Revision Fig. 3 below.

Revision Figure 3: Dose-response curves and dose-response matrices for drug treatments in the four ovarian cancer cell lines ES-2, EFO-2, OVCAR-8, and OAW-42. This figure is included in the revised manuscript as Fig. EV7.

In summary, three out of four cell lines (OVCAR-8, EFO-21, and ES-2) responded well to the DHCR7 inhibitor AY9944 (IC-50 < 10 μM), and in a dose-dependent manner. We used Synergyfinder+ (<https://synergyfinderplus.org/>) to calculate synergistic drug effects, which revealed significant synergy between carboplatin and AY9944 for those three cell lines. Synergy was strongest (HSA score > 10, high synergy) in the cell lines OVCAR-8 and EFO-21 and moderate for ES-2 (HSA score = 7.52). For the combination carboplatin – simvastatin, we likewise found synergistic drug effects, but generally at lower synergy scores (See Revision Table 1, below). The negative synergy score for OVCAR-8 was an

exception, which can be explained by the fact that even the lowest tested simvastatin dose (3 μ M) already resulted in 70-80% inhibition, thus not allowing us to reliably assess synergistic effects with carboplatin at these concentrations. We conclude that inhibition of cholesterol biosynthesis resulted in strong dose-dependent growth inhibition in all four cell lines and that targeting this metabolic pathway via DHCR7 could represent a novel therapeutic strategy, especially in cases of platinum resistance. We have included these new data in Fig. 7, Fig. EV7 and Appendix Table 1 in the revised manuscript (also attached below) and edited the main text accordingly.

Cell line	Drug combination	HSA Synergy Score (Mean) *** p-value < 0.001
ES-2	Carboplatin - AY9944	7.52 ***
EFO-21	Carboplatin - AY9944	13.5 ***
OVCAR-8	Carboplatin - AY9944	14.9 ***
OAW-42	Carboplatin - AY9944	2.38 n.s.
ES-2	Carboplatin - Simvastatin	7.9 ***
EFO-21	Carboplatin - Simvastatin	9.36 ***
OVCAR-8	Carboplatin - Simvastatin	-3.52 ***
OAW-42	Carboplatin - Simvastatin	4.64 ***

Revision Table 1: Synergy scores for all tested drug combinations. An HSA score of greater than 10 indicates high synergy.

Revision Figure 4: Drug treatment assays of single and combination treatments for all four cell lines. Cell lines are ordered by synergy score from highest to lowest. This figure is included in the revised manuscript as new Fig. 7.

b. The cell lines respond differently to the treatments. A genetic and proteomic characterization of the cell lines should be conducted to potentially elucidate the underlying molecular cause.

We thank the reviewer for bringing up this important point. Using publicly available genomics and proteomics data of all four cell lines, we now provide a detailed genomic and proteomic characterization to compare subtype annotation, mutation status, copy number alterations, DHCR7 protein levels and protein abundance levels of the cholesterol biosynthesis pathway (Revision Table 2). These data are also included in the revised manuscript as Suppl. Table EV24. Genomic information was retrieved from Cell Modell Passports (<https://cellmodellpassports.sanger.ac.uk/>, Wellcome Sanger Institute) and proteomics data from a recent pan-cancer proteomic study by Goncalves et al., Cancer Cell (PMID: 35839778).

Overall, we did not observe a clear association between DHCR7 protein abundance, or more generally cholesterol pathway abundance, and the response to the DHCR7 inhibitor AY9944 or Simvastatin. However, we noted that the least responsive cell line OAW-42 showed the lowest DHCR7 protein abundance. Interestingly, previous studies have identified a causal link between *TP53* mutational status and the expression of cholesterol pathway genes. *TP53* mutant cells up-regulate cholesterol related genes, for example as shown in breast (PMID: 22265415) and liver cancer (PMID: 30580964).

Notably, all three cell lines that responded to AY9944 were *TP53* mutant lines, supporting the notion that *TP53* status is an important component determining treatment responses to cholesterol inhibitors.

Moreover, a previous study identified that inhibiting cholesterol synthesis or uptake (e.g., using statins or other inhibitors) can selectively impair the growth of *ARID1A*-mutant ovarian clear-cell cancer cells (PMID: 36963401). Our genetic comparison revealed that OAW-42 carried an *ARID1A* mutation, but this cell line was generally less responsive to compared to the other models. We therefore conclude that *TP53* status could be a more critical factor in determining sensitivity to cholesterol inhibition than *ARID1A* status.

We have revised the main text of our manuscript to contextualize the observed cell line drug responses based on our proteogenomic comparison.

Characteristic	OVCAR-8	EFO-21	OAW-42	ES-2
Subtype annotation	High Grade Ovarian Serous Adenocarcinoma	Ovarian Clear Cell Adenocarcinoma	Ovarian Serous Cystadenocarcinoma	Ovarian Clear Cell Adenocarcinoma
TP53 mutation status	Yes (splice site mutation)	Yes (missense)	No	Yes (missense)
BRCA status	WT	BRCA2 mut	BRCA1 mut	BRCA2 mut
MSI stats	MSS	MSS	MSS	MSS
Copy number alterations	MYC , FAM135B , CCNE1 amplification, MECOM loss	MAP3K13 , MB21D2 , EIF4A2 amplification, NF1 , RSPH10B2 loss	NFATC2 , PTK6 , SALL4 amplification	NSD2 , BIRC3 , FGRF3 amplification
Mutations	ERBB2 GOF mutation, EP300 , B2M , AMER1 LOF mutation	LRP1B LOF mutation	ARID1A LOF mutation, PIK3CA GOD mutation	BRAF , CDK4 , MAP21K (GOF mutations), TCIRG1 LOF mutation

Revision Table 2: Genomic characterization of the four tested ovarian cancer cell lines.

Revision Figure 5: A) Relative protein levels of all detected cholesterol biosynthesis proteins in OVCAR8, ES2, EFO21 and OAW42 cell lines. B) Relative protein levels of DHCR7 in OVCAR8, ES2, EFO21 and OAW42 cell lines. Data was obtained from Gonçalves et al (PMID: 35839778).

c. Figure 6G should be replaced. The 'synergy score' axis is skewed with the numbers partly unreadable, and scaling of the synergy score to include negative values is unnecessary as no negative values appear in the plot. It should be replaced with something similar to EV6G, in line with further experiments suggested above.

We have now revised this figure accordingly including synergy scores for all four cell lines and both drug combinations (carboplatin-AY994, carboplatin-simvastatin). These results are included in Fig. 7 and Fig. EV7.

Minor points

1. Red/green should not be used as a colour scheme in plots and should be replaced (e.g. Figure 3E).

We have revised all figures that included a green/red color code.

2. The 'Top 100 stromal proteins' are mentioned in the text but should also be shown in an additional figure or at least marked in Figure 2.

We now provide these data as additional expanded view tables (EV20 and EV21) that include the top abundant proteins and their abundance levels.

3. Readability (font size) of figure text needs to be improved throughout the manuscript, e.g. Figure 2D; EV3B.

For better readability, we have carefully checked and adjusted all font sizes or edited the labeling when we felt it was necessary. We hope these adjustments have made our figures more readable.

4. Axis labelling needs to be improved to make it easier for the reader to understand (e.g. Figure 4B, i.e. "Relative protein level (log2)")

We have adjusted the axis labelling throughout the manuscript and changed "Relative protein level (log2)" to "Fold change (log2)".

5. Figure formatting is not consistent and should be reworked throughout the paper, e.g. removing background lines (e.g. Figure 3F).

We have now revised all figures for consistency.

6. Clarify whether all upregulated proteins or the proteins above the significance threshold of Figure 3B contribute to the enriched pathways in Figure 3D. If a threshold was used (which is highly recommended), only the proteins above the cutoff line in 3B should be coloured in pink. In addition, at least the proteins highlighted in the text should be linked to the respective dot in the volcano plot. All volcano plots require a rework accordingly.

For the pathway enrichment analysis shown in Fig. 3D, we employed one-dimensional annotation enrichment analysis as described by Cox et al. (PMID: 23176165). In contrast to categorical enrichment (e.g., comparing a subset of significant proteins versus all proteins), this approach is based on numerical data. We used the protein fold change (X-axis) of the volcano plot shown in Fig. 3B as input for enrichment analysis. We prefer this numerical enrichment approach as it is generally more sensitive in detecting global pathway level changes. Importantly, this strategy also encompasses a strict (5%) false-discovery rate. For the volcano plot in Fig. 3B, we have now linked the respective dots to each gene symbol label, and highlighted the most differentially abundant proteins in the figure.

7. More comprehensive and consistent naming is highly recommended - e.g. "normal" vs NFTE –

We have re-labeled all “normal” epithelial samples to “NFTE” in the text and figures for consistency.

8. The titles of the Figures and subchapters should be adjusted to reflect the main message of the paragraph and not the overall story line.

We have revised all figure and subchapter titles to more accurately reflect the main message of each section.

Reviewer #3:

The article from Makhmut and Dragomir et al. utilizes a cutting-edge spatial proteomic approach for a comprehensive characterization of the epithelial and stromal compartment of tissues from 36 HGSOc patients. Specifically, the authors characterized the proteome of normal fallopian tubes (NFTE), serous tubal intraepithelial lesions (STICs or early lesions), and invasive carcinoma (IC or advanced disease), providing well-resolved epithelial and stromal proteomes that represent a valuable resource for the scientific community. The authors also use DVP to characterize the proteome of PAX8+ (secretory cells) and PAX8- (ciliated-like) cells, demonstrating that PAX8+ secretory cell proteomes display overlapping signatures with STICs, thus supporting the view of the ovarian cancer community that HGSOc originates from secretory cells. Furthermore, the authors compare both epithelial and stromal proteomes of STICs (for the epithelium) or normal-like (for the stroma) and IC revealing a high degree of similarity in the epithelial compartment, and well-defined differences in proteomic composition of the stromal compartment. Finally, the authors compared the epithelial proteomes of NFTE and STICs, demonstrating pronounced differences, particularly as it relates to metabolism. They identify the cholesterol biosynthesis pathway as a promising drug target in combination with chemotherapy and demonstrated the efficacy of such combination in vitro.

Overall, this study generates a comprehensive clinical dataset of HGSOc of great value for the scientific community. The authors also conduct elegant comparisons with publicly available proteomics and gene expression data in equivalent settings, making interesting parallels between RNA and protein alterations in ovarian cancer. The design, approach, results and analysis strategies employed in the study are impressive. I also commend the authors for putting together such organized source data file. Overall, I have a few suggestions that are listed below that I believe would enhance the impact and translational value of the study.

We are very pleased about the very positive feedback on our work and thank the reviewer for the additional comments and suggestions to further improve our study.

Major points:

1. The authors observe a high similarity between the STIC and IC epithelial proteomes, with only 9 proteins being significantly regulated. However, the samples did segregate into an immunoreactive cluster 1 and a proliferative cluster 2, with well-defined proteomic signatures (Figure 4). In spite of the clustering being independent of TP53 mutation type, histological stage, and HRD status, it would be interesting for the authors to conduct a correlation analysis between clusters 1 and 2 with patient survival, which is an important clinical parameter that has not been explored in this study. This has the potential to shed light into the implications of these epithelial signatures in clinical outcome. Furthermore, a link between the epithelial and stromal compartments has not been addressed. Considering that the stroma can influence the behavior of cancer cells, it is important to evaluate if there are unique features of the stroma associated with these clusters.

We fully agree that the additional analyses to compare patient survival of cluster 1 and 2 tumors and to assess the stromal proteome composition are an important addition to our results.

Survival analysis between cluster 1 and cluster 2 tumors:

Initially, we did not include a survival analysis of cluster 1 and 2 tumors due to the rather limited size of our patient cohort with only 35 patients after filtering. However, we have now conducted an additional univariate survival analysis to complement the previous multivariate analysis shown in Fig. EV4B. As expected, univariate Kaplan-Meier analysis (Revision Fig. 6) showed a clear trend (p-value = 0.082) towards longer overall survival of cluster 1 tumors (immune-enriched) compared to cluster 2 tumors (proliferative subtype). However, our multivariate survival analysis in Fig. EV4B further revealed that patients with cluster 2 tumors were generally older, in good agreement with recent a study (Geistlinger et al, PMID: 32747365), showing that proliferative tumors are associated with higher patient age and more advanced disease stages. We have included this analysis as Appendix Fig. 1.

Overall survival (OS) based on consensus clustering

Revision Figure 6: Kaplan Meier plot comparing overall survival between patients with cluster 1 tumors (turquoise) and cluster 2 tumors (dark blue). The x-axis indicates time in days, and the y-axis shows the estimated probability of survival. Significance levels were determined using the log-rank test.

Stromal proteome comparison of cluster 1 and 2 tumors:

To analyze the stromal proteomic differences between cluster 1 and 2 tumors, we included additional analyses. First, we performed a pairwise (two-sided t-test) proteomic comparison of all stromal samples that we grouped by the corresponding tumor cluster id (cluster 1 or 2). Overall, this revealed only a 17 differentially regulated proteins at an FDR of 5% (Revision Fig. 7A).

To globally compare and integrate the tumor and stromal results, we additionally performed a pathway enrichment analysis based on the protein fold change calculated between the cluster 1 and 2 stromal proteomes (Revision Fig. 7B). Cluster 1 stroma samples were strongly enriched for immune-associated pathways and macrophage signatures, consistent with the association of cluster 1 STIC/tumor samples with the 'immunoreactive' (IMR) subtype assignment by consensusOV. The higher macrophage signature in cluster 1 tumor is consistent with a previous study that linked to IMR subtype to higher intratumoral macrophage infiltration (PMID:32747365). In contrast, up-regulated proteins in cluster 2 stromal samples were enriched for ECM-associated processes.

In the revised manuscript, the tumor cluster information is provided in the stromal proteomics heatmap in Fig. 5B. We also included a statement in the main text to describe the association of the stromal proteomes with the two tumor clusters: "We found that stromal clustering could generally not

be explained by the two tumor clusters (Fig. 5B), emphasizing the importance of compartment and cell-type resolved analyses to study spatially-defined changes during disease progression. Nevertheless, immune-related and inflammatory pathways were clearly higher in the stroma of immune-enriched cluster 1 tumors.”. We have included this analysis as Appendix Fig. 2.

Revision Figure 7: A) Volcano plot of pairwise comparison between cluster 1 (turquoise) and cluster 2 (dark blue) STIC/tumor stroma samples. Proteins with the highest fold changes are highlighted (two-sided Student’s t-test, false discovery rate [FDR] < 0.05). B) Pathway enrichment analysis (WikiPathways, Hallmarks, Reactome) based on the t-test difference between cluster 1 and cluster 2 epithelial samples. Selected pathways with a Benjamin-Hochberg FDR < 0.05 are shown.

2. Using SVM, the authors identify a 100-protein stromal signature that can distinguish the normal from the malignant stroma. Importantly, 9 of these proteins (Figure 5H) have FDA-approved drugs for specific targeting. This is a very promising finding that could be explored further. I would suggest that the authors use relevant stromal cells to test whether these FDA-approved drugs reduce their pro-tumorigenic traits and their ability to support invasion of cancer cells. I believe that this would certainly increase the impact and translational relevance of this study. Furthermore, the discussion addressing these 9 targets with FDA-approved drugs is incomplete. Please expand for each target on which stromal cell population these proteins are expressed (e.g. authors can use publicly available single cell data) and the relevance of their targeting.

We thank the reviewer for this great suggestion. We agree that the functional validation of the identified stromal drug targets using stromal cell lines would further improve the translational significance of our findings. This was one motivation behind our study to provide a rich proteomic resource that stimulates new research assessing novel protein-based drug targets for ovarian cancer treatment. Unfortunately, we did not have access to stromal lines to conduct the proposed co-culture experiments within the rather short time of revision. We believe the suggested assays using several drugs and co-culture models would have required significantly more time, particularly as optimal drug concentrations and treatment timepoints would need to be identified first. Nevertheless, we have performed a number of additional *in-silico* analyses based on available HGSOc scRNA seq datasets to investigate the expression patterns of the identified drug targets across ovarian cancer cell types of the tumor microenvironment.

We first re-analyzed data from Izar et al (PMID:32572264), in which the authors used single-cell RNAseq to characterize ascites samples from patients with HGSOc. This dataset includes six stromal cell types (B-cell, T-cell, dendritic cell, fibroblast, macrophage, and erythrocyte) and one cancer cell type (Malignant). Comparing the expression levels of our eight prioritized stromal drug targets, we found that most (TYMP, FCER1G, FCGR1A, SYK, and F13A1) were highly expressed in tumor-associated macrophages / myeloid cells (Revision Fig. 8). NNMT was mainly expressed in cancer-associated fibroblasts, consistent with our previous study, which identified a novel function of this protein as

druggable CAF regulator (PMID: 31043742) and a more recent follow-up study of this work (PMID: 40702186). DCK and CASP3 showed a more heterogenous expression pattern, which did not allow us to derived a clear cell type assignment. We have added these new data to Suppl. Table EV23, Appendix Fig. 3, and revised the main text accordingly.

Revision Figure 8: Heatmap of average gene expression levels of eight FDA-approved drug targets across seven cell types. Rows represent genes, and columns represent distinct clusters annotated by cell type. Expression levels are indicated by color intensity (red – higher expression, yellow – lower expression). Data was obtained from Izar et al. (PMID: 32572264).

We additionally analyzed the single-cell RNA-seq data from Xu et al. and Olbrecht et al. to compare the expression of the identified FDA-approved drug targets between malignant (HGSOC) and non-malignant samples. These data revealed that most genes were up-regulated in ovarian cancer (Revision Figs 9 and 10). We believe these data are important additions to our manuscript. In addition to the newly provided Table EV23, we included Revision Fig. 10H as new Fig. 5J in the revised manuscript.

Revision Figure 9: A) A UMAP plot depicting the distribution of single cells across different cell types. B) Split UMAP plot depicting the distribution of single cells across different cell types in HGSOc (left) and non-malignant samples (right). C) Feature plot showing the expression of DCK (C), F13A1 (D), FCGR1A (E), NNMT (F), SYK (G), and TYMP (H).

Revision Figure 10: Violin plots displaying the expression levels of CASP3 (A), DCK (B), F13A1 (C), FCGR1A (D), NNMT (E), SYK (F), and TYMP (G) in HGSOC and non-malignant samples. The width of the violin depicts the density of cells at each expression level. (H) Dotplot illustrating the expression of NNMT, FCGR1A, TYMP, F13A1, DCK, CASP3 and SYK across different cell types in HGSOC and non-malignant samples.

In terms of targeting these proteins therapeutically, there is supporting literature for a of these proteins. Our own previous work (PMID: 31043742) identified NNMT as a master metabolic regulator of cancer-associated fibroblasts (CAFs). Building on this, Heide et al. (PMID: 40702186) further investigated NNMT's role in HGSOc and developed a novel NNMT inhibitor, which reduces tumor metastasis and burden in several mouse models and reversed immunosuppressive microenvironment. This study illustrates the beneficial impact of targeting TME components. In our study, we observed that NNMT is upregulated at the STIC level, indicating that its therapeutic targeting may impact precursor lesions and thereby intervene in HGSOc progression at an early stage.

3. In Figure 5, when displaying the data from the normal stroma vs the invasive stroma clusters, the authors should include a Volcano Plot containing all the proteins identified (as they did in Figure 4C for the epithelial compartment), alongside with the source data table. The Volcano plot in Figure 5C only shows the proteins related to the Matrisome. Please include the full information in the revised manuscript.

We thank reviewer for this suggestion and we agree that presenting the complete comparison between invasive stroma and normal (NFT) stroma beyond matrisome proteins provides a more comprehensive view of the stroma remodeling that accompanies HGSOc progression. In the revised manuscript, we have included a global pairwise proteomic comparison displaying all quantified proteins in the IC related stroma vs. NFT stroma in Fig. 5C (Revision Fig. 11). This analysis identified 1904 significantly upregulated proteins in the IC stroma and 1238 in the NFT stroma (p -value < 0.05).

Revision Figure 11: Left: Volcano plot of pairwise comparison between IC stroma (purple) and NFTE stroma (light blue) samples. Proteins with the highest fold changes are highlighted (two-sided Student's t -test, false discovery rate [FDR] < 0.05). Right: Same comparison but only showing all matrisome associated proteins.

4. The authors nicely compare the differences in the epithelial compartment between NFTE and STICs but do not explore such differences in the stromal compartment. Even though STIC stroma segregated half with NFTE and half with IC stroma, there might still be distinct differences between STIC and NFTE stroma that were not captured in the PC analysis. I would strongly recommend that at least a supplementary figure comparing the NFT and STIC stroma is included in the revised manuscript. This information will certainly add to their findings and will be a valuable resource for the scientific community studying the ovarian cancer stroma.

Please see comment above. We have now revised Fig. 5C and included the comparison of all normal stroma and STIC stroma samples. We also provide these data in Table EV13 to make them easily accessible for the scientific community.

5. The authors identify DHCR7 and DHCR24 as significantly upregulated in epithelial STICs vs NFTE and used DHCR7 inhibitor to show the potential of cholesterol synthesis pathway in HGSOC in vitro. To validate their findings, the authors should use mass spec imaging to show increased cholesterol synthesis in the STICs compared to normal tissue.

We fully agree that this would be a great validation of our findings. However, this experiment turned out to be impossible in practice due to the limitation of FFPE material for lipid/cholesterol profiling. The biggest limitation is that the analysis of frozen STIC samples is impossible as STICs are exclusively diagnosed and studied from archival FFPE tissue specimens. We consulted two large pathology departments (Charité Pathology as Europe's largest University hospital and the University of Chicago). Both confirmed that the access to frozen fallopian cancer tissue samples including precancerous STICs is not possible. Due to harsh organic solvent based dewaxing steps (xylene and ethanol), FFPE tissues are generally considered incompatible with lipid analyses such as cholesterol and its derivatives. However, in response to this suggestion, we have still performed a number of experiments to assess whether MALDI imaging of lipids could still be done.

We modified the protocol described by Alice Ly et al., Nature Protocols (2016, PMID: 27414759), which includes FFPE tissue deparaffinization by heating to 60°C for one hour, followed by three-minute washes in xylene two times. This is done to reduce the loss of small molecules, metabolites, and other analytes during the washing procedure. While cholesterol is readily detectable in standard solutions (red dot, Revision Fig. 11), it is generally not detectable in tissue samples under the same conditions.

Revision Figure 12: (Top) MALDI mass spectrometry imaging of cholesterol distribution in ten FFPE tissue sections corresponding to ten patients, acquired using Bruker Rapiflex instrument. Matrix: 7 g/L α -cyano-4-hydroxycinnamic acid. Mass range: 150–900 Da, positive ion mode. Spatial resolution: 50 μ m pixel size with 500 shots per pixel. Imaging targeted the cholesterol at m/z 369.35 m/z. (Bottom) Mass spectrum of cholesterol ion at m/z 369.35 m/z.

However, to provide additional orthogonal validation, we have now performed IHC stainings on our NFTE, STIC and IC samples to validate high DHCR7 and DHCR24 protein levels in STICs and invasive tumors versus NFTE. These results are now included in the revised Fig. 6F-G (Revision Fig. 13, below). Note, both proteins are strongly upregulated in STICs compared to normal fallopian tube epithelium, as discovered by our MS-based spatial proteomics approach.

Revision Figure 13: IHC validation of DHCR7 and DHCR24 in STICs and invasive carcinomas.

6. It would be interesting to use available clinical data of cholesterol levels in HGSOC patients and investigate its correlation with patient survival and even therapy response.

Unfortunately, we did not have access to the clinical cholesterol measurements for our patient cohort. We believe for such analysis a much large number of patients (i.e., >1000) would be required due to strong confounding factors from diet, age, co-morbidities etc. We fully agree that this is an interesting direction for the future.

7. In the discussion section, I recommend that the authors discuss how their findings can be explored for biomarker discovery aimed at early detection of HGSOC.

We have expanded the discussion section to elaborate on how our findings could be applied to detect HGSOC at an early stage. For example, we discuss the possibility to use metabolite/lipid assays of pelvic fluid as a potential early detection strategy, similar to a recent study (PMID: 39032862). Additionally, highly sensitive assays focusing on tumor-derived extracellular vesicles (EVs) could target specific HGSOC cell surface proteins that could be shed by early cancer lesions. Our deep proteomic data identified numerous candidate proteins that may serve as important antibody targets for EV detection. Interestingly, a previous study detected endogenous DHCR24 peptides in blood of ovarian cancer patients using mass spectrometry (PMID: 30598658).

Minor points:

1. Page 10, line 318: correct figure is 5H, not 5F as indicated in the text.

This is corrected now.

2. Figures: Please use colorblind safe colors, in particular avoid green and red in the same plot when possible, for example Fig. 3e,f.

This is corrected now.

3. Fig. 5G. It is difficult to read what's written in black over violet.

We changed this figure accordingly.

4. The authors have uploaded on PRIDE the results of the search engine, but not the raw MS files. The authors should upload their raw MS files as it is common practice to enable access to the community.

Now all raw files are provided through PRIDE.

12th Sep 2025

Manuscript Number: MSB-2025-13047R

Title: Spatial proteomics of ovarian cancer precursors delineates early disease changes and drug targets

Author: Anuar Makhmut

Mihnea Dragomir

Sonja Fritzsche

Markus Möbs

Wolfgang Schmitt

Eliane Taube

Fabian Coscia

Dear Fabian,

Thank you for sending us your revised manuscript. We have now heard back from the two reviewers who were asked to evaluate your revised study. As you will see below, the reviewers are satisfied with the performed revisions and support publication. Before we can formally accept the manuscript for publication, we would ask you to address some remaining issues listed below.

1. The remaining minor issues raised by Reviewer #3.

On a more editorial level:

2. Please provide up to five keywords in the manuscript file.

3. Data availability: please remove the reviewer token and provide a specific URL for dataset PXD060547. Please make sure the datasets will be made publicly available upon the acceptance of the manuscript.

4. Remove the Author Contributions section from the manuscript file.

5. Funding Information: The funding details provided in the Comments box (Berliner Krebsgesellschaft Grant [DRFF202204]) could not be extracted by our production team. Please ensure that all funding sources are added to the "More Funders" section in your next submission.

6. Make sure all callouts are listed sequentially. Add missing callouts for Fig. 3G, 5I, 7A, EV5, Table EV17, EV20-EV21, EV24.

7. Synopsis Image: Please resize the image to 550 pixels in width and 400-600 pixels in height, and provide it in PNG format. Ensure that all text remains clear and not blurry at the adjusted size.

8. Appendix: The title page should include the heading "Appendix for [Manuscript Title]" along with a Table of Contents that lists all items with their corresponding page numbers. Throughout the manuscript and the Appendix PDF, please use the following nomenclature: Appendix Figure Sx and Appendix Table Sx.

9. Tables in the manuscript should be labeled as Table 1 and Table 2, with their legends placed above each corresponding table. The tables should be positioned between the main figure legends and the EV figure legends.

10. EV tables and datasets: Tables EV1, EV4, EV14, and EV23 should remain as EV Tables, but must be renumbered sequentially as Table EV1-EV4. Their legends should be removed from the manuscript file and instead placed above the corresponding tables in each Excel file. Please ensure that all callouts in the manuscript are updated accordingly.

For all other EV Tables (Tables EV2-EV3, EV5-EV13, EV15-EV22, and EV24), these should be renamed and treated as Dataset EV1-EV#. The source file names, table titles, legends, and manuscript callouts must all be updated accordingly. Additionally, legends for these datasets should be removed from the manuscript and uploaded as a separate tab/sheet within each corresponding Excel file.

11. Figure 4G shows pixelation when image filters are applied, despite being a high-resolution figure. Please provide the source data for this figure.

12. Please address the following issues in figure legends:

- Please note that the exact p values are not provided in the legends of figures 2I, 6C, G; 7B

- Please note that the box plots need to be defined in terms of minima, maxima, centre, bounds of box and whiskers, and percentile in the legends of figures 5G, I

- Please note that information related to n is missing in the legends of figures 2A, I; 3B, E, G; 4C, 5C, D, G, I; 6C, G; EV3 F, G; EV4 G, I, EV5 A, B

13. The section heading "Abstract" should not be bolded, and "Methods and Protocols" should be renamed to "Methods".

14. Please revise the section order as follows: Title page - Abstract - Keywords - Introduction - Results - Discussion - Methods - Data Availability - Acknowledgements - Disclosure and Competing Interests Statement - References - Figure Legends - Table(s) - Expanded View Figure Legends.

Click on the link below to submit your revised paper.

Kind regards,
Jingyi

Jingyi Hou, PhD
Senior Editor
Molecular Systems Biology

*** PLEASE NOTE *** As part of the EMBO Press transparent editorial process initiative (see our Editorial at <https://dx.doi.org/10.1038/msb.2010.72>), Molecular Systems Biology will publish online a Review Process File to accompany accepted manuscripts. When preparing your letter of response, please be aware that in the event of acceptance, your cover letter/point-by-point document will be included as part of this File, which will be available to the scientific community. More information about this initiative is available in our Instructions to Authors. If you have any questions about this initiative, please contact the editorial office (msb@embo.org).

Reviewer #2:

The points raised during the initial review have been adequately addressed.

Reviewer #3:

We thank the authors for addressing our concerns. Overall, the study has greatly improved with the revisions. There are only a few minor points that remain to be addressed:

1. Raw files are still not uploaded in PRIDE
2. Please provide histology quantification for immune infiltration and markers assessed in images displayed in figure 4F-G.
3. Line 298: should this be Fig 5D rather than 5C?
4. Line 329: should this be Fig 5I rather than 5H?
5. Line 331: to accurately reflect the findings, I would change macrophage-derived to 'macrophage and monocyte associated with malignant features.'

Point-by-point response to the remaining reviewers' and editorial comments

We thank all reviewers again for their careful re-evaluation of our manuscript entitled "*Spatial proteomics of ovarian cancer precursors delineates early disease changes and drug targets*". In response, we performed additional minor revisions, which are explained in more detail below.

Reviewer #2:

The points raised during the initial review have been adequately addressed.

We again thank the reviewer for the positive assessment of our work and the very constructive feedback that allowed us to further improve our study.

Reviewer #3:

We thank the authors for addressing our concerns. Overall, the study has greatly improved with the revisions. There are only a few minor points that remain to be addressed:

We are very glad to read that our revised manuscript has addressed the reviewers' concerns and again thank the reviewers for their thorough and constructive evaluation of our work.

1. Raw files are still not uploaded in PRIDE.

We apologize that the raw data were still not available in the PRIDE database after the first round of revision. We have experienced significant problems with the submission system over a few weeks now. We confirm that all raw data will be made publicly available on PRIDE. We are in frequent exchange with Pride to solve the upload problems as soon as possible.

2. Please provide histology quantification for immune infiltration and markers assessed in images displayed in figure 4F-G.

We thank the reviewer for this valuable suggestion. We have now included the histology quantification of immune infiltration and the used markers shown in Fig. 4F-G. These results are provided in the Appendix file (Appendix figure 5, Appendix methods), together with an additional figure presenting the quantification data for both immune and functional markers. These data support the raw images and the drawn conclusions described in the main text.

Appendix figure 4: Quantification of marker-positive cells in histological tissue sections from patients W31V (cluster 1, IMR subtype) and W30W30D (cluster 2, PRO subtype). Immunofluorescence staining was performed to detect various markers across tissue sections (as in Figure 4F-H). The proportion of marker-positive cells was quantified for each marker using QuPath. Each value represents the ratio of cells positive for a given marker to the total number of DAPI-positive nuclei. Markers analyzed include TP53, Ki67, CD4, CD8, PAX8, FOXP3, CD163. Boxplot represent the proportion of marker-positive cells for each marker. Data represent a single experimental replicate per sample.

Appendix figure 5: Representative immunofluorescence microscopy image of cancer tissue sections from patients W31V (cluster 1, IMR subtype, upper panel) and W30W30D (cluster 2, PRO subtype, lower panel), showing TP53, Ki67, CD4, FOXP3, CD8, PAX8, CD163 and DAPI. Cell segmentation was performed using QuPath software to identify individual cells within the tissue. Segmented cells were subsequently analyzed to determine the proportion of marker-positive cells relative to the total number of cells. Scale bar: 50 μm.

3. Line 298: should this be Fig 5D rather than 5C?

This is corrected now.

4. Line 329: should this be Fig 5I rather than 5H?

This is corrected now.

5. Line 331: to accurately reflect the findings, I would change macrophage-derived to 'macrophage and monocyte associated with malignant features.'

We agree that 'macrophage and monocyte associated with malignant features' would better reflect findings and have now changed it in the manuscript.

Editorial requests

2. Please provide up to five keywords in the manuscript file.

We have now provided four keywords in the manuscript file: spatial tissue proteomics, high-grade serous ovarian cancer, serous tubal intraepithelial carcinoma, cancer proteomics

3. Data availability: please remove the reviewer token and provide a specific URL for dataset PXD060547. Please make sure the datasets will be made publicly available upon the acceptance of the manuscript.

The proteomics data is now publicly available.

4. Remove the Author Contributions section from the manuscript file.

The Author Contribution section is now removed from the manuscript file.

5. Funding Information: The funding details provided in the Comments box (Berliner Krebsgesellschaft Grant [DRFF202204]) could not be extracted by our production team. Please ensure that all funding sources are added to the "More Funders" section in your next submission.

We confirm.

6. Make sure all callouts are listed sequentially. Add missing callouts for Fig. 3G, 5I, 7A, EV5, Table EV17, EV20-EV21, EV24.

We have now provided missing callouts for the figures and tables mentioned and made sure they are listed sequentially.

7. Synopsis Image: Please resize the image to 550 pixels in width and 400-600 pixels in height, and provide it in PNG format. Ensure that all text remains clear and not blurry at the adjusted size.

The image is now adjusted to meet the requirements.

8. Appendix: The title page should include the heading "Appendix for [Manuscript Title]" along with a Table of Contents that lists all items with their corresponding page numbers. Throughout the manuscript and the Appendix PDF, please use the following nomenclature: Appendix Figure Sx and Appendix Table Sx.

The Appendix file is now modified and contains the required information, including table of contents, page numbers and name adjustments.

9. Tables in the manuscript should be labeled as Table 1 and Table 2, with their legends placed above each corresponding table. The tables should be positioned between the main figure legends and the EV figure legends.

10. EV tables and datasets: Tables EV1, EV4, EV14, and EV23 should remain as EV Tables, but must be renumbered sequentially as Table EV1-EV4. Their legends should be removed from the manuscript file and instead placed above the corresponding tables in each Excel file. Please ensure that all callouts in the manuscript are updated accordingly.

Provided accordingly.

For all other EV Tables (Tables EV2-EV3, EV5-EV13, EV15-EV22, and EV24), these should be renamed and treated as Dataset EV1-EV#. The source file names, table titles, legends, and manuscript callouts must all be updated accordingly. Additionally, legends for these datasets should be removed from the manuscript and uploaded as a separate tab/sheet within each corresponding Excel file.

We have now modified tables and separated them into EV tables and Dataset EV tables. All manuscripts callouts are updated and listed sequentially.

11. Figure 4G shows pixelation when image filters are applied, despite being a high-resolution figure. Please provide the source data for this figure.

We have now uploaded cyclF images from the Fig. 4 to the BioImage Archive database under accession number S-BIAD2315 (<https://www.ebi.ac.uk/biostudies/bioimages/studies/S-BIAD2315>).

12. Please address the following issues in figure legends:

- Please note that the exact p values are not provided in the legends of figures 2I, 6C, G; 7B
- Please note that the box plots need to be defined in terms of minima, maxima, centre, bounds of box and whiskers, and percentile in the legends of figures 5G, I
- Please note that information related to n is missing in the legends of figures 2A, I; 3B, E, G; 4C, 5C, D, G, I; 6C, G; EV3 F, G; EV4 G, I, EV5 A, B.

This information is now included in the revised manuscript. Due to the large number of exact p-values and groups, we provided this information as tables in the Appendix.

13. The section heading "Abstract" should not be bolded, and "Methods and Protocols" should be renamed to "Methods".

This is now modified in the revised version of the manuscript.

14. Please revise the section order as follows: Title page - Abstract - Keywords - Introduction - Results - Discussion - Methods - Data Availability - Acknowledgements - Disclosure and Competing Interests Statement - References - Figure Legends - Table(s) - Expanded View Figure Legends.

We confirm.

22nd Oct 2025

Manuscript number: MSB-2025-13047RR

Title: Spatial proteomics of ovarian cancer precursors delineates early disease changes and drug targets

Dear Dr. Coscia,

Thank you again for sending us your revised manuscript. We are now satisfied with the modifications made and I am pleased to inform you that your paper has been accepted for publication.

Yours sincerely,

Sincerely,
Jingyi

Jingyi Hou, PhD
Senior Editor
Molecular Systems Biology
